# On the Implicit Bias of Adam

## Abstract

In previous literature, backward error analysis was used to find ordinary differential equations (ODEs) approximating the gradient descent trajectory. It was found that finite step sizes implicitly regularize solutions because terms appearing in the ODEs penalize the two-norm of the loss gradients. We prove that the existence of similar implicit regularization in RMSProp and Adam depends on their hyperparameters and the training stage, but with a different "norm" involved: the corresponding ODE terms either penalize the (perturbed) one-norm of the loss gradients or, on the contrary, hinder its decrease (the latter case being typical). We also conduct numerical experiments and discuss how the proven facts can influence generalization.

## 1 Introduction

Gradient descent (GD) can be seen as a numerical method solving the ordinary differential equation (ODE) $\dot{\boldsymbol{\theta}} = -\nabla E(\boldsymbol{\theta})$, where $E(\cdot)$ is the loss function and $\nabla E(\boldsymbol{\theta})$ is its gradient. Starting at $\boldsymbol{\theta}^{(0)}$, it creates a sequence of guesses $\boldsymbol{\theta}^{(1)}, \boldsymbol{\theta}^{(2)}, \ldots$, which lie close to the solution trajectory $\boldsymbol{\theta}(t)$ governed by aforementioned ODE. Since the step size $h$ is finite, one could search for a modified differential equation $\dot{\tilde{\boldsymbol{\theta}}} = -\nabla \widetilde{E}(\tilde{\boldsymbol{\theta}})$ such that $\boldsymbol{\theta}^{(n)} - \tilde{\boldsymbol{\theta}}(nh)$ is exactly zero, or at least closer to zero than $\boldsymbol{\theta}^{(n)} - \boldsymbol{\theta}(nh)$, that is, all the guesses of the descent lie exactly on the new solution curve or closer compared to the original curve. This approach to analysing properties of a numerical method is called backward error analysis in the numerical integration literature (see Chapter IX in Ernst Hairer & Wanner (2006)).

Barrett & Dherin (2021) first used this idea for full-batch GD and found that the modified loss function $\widetilde{E}(\tilde{\boldsymbol{\theta}}) = E(\tilde{\boldsymbol{\theta}}) + (h/4)\|\nabla E(\tilde{\boldsymbol{\theta}})\|^2$ makes the trajectory of the solution to $\dot{\tilde{\boldsymbol{\theta}}} = -\nabla \widetilde{E}(\tilde{\boldsymbol{\theta}})$ approximate the sequence $\{\boldsymbol{\theta}^{(n)}\}_{n=0}^{\infty}$ one order of $h$ better than the original ODE, where $\|\cdot\|$ is the Euclidean norm. In related work, Miyagawa (2022) obtained the correction term for full-batch GD up to any chosen order, also studying the global error (uniform in the iteration number) as opposed to the local (one-step) error.

The analysis was later extended to mini-batch GD in Smith et al. (2021). Assume that the training set is split in batches of size $B$ and there are $m$ batches per epoch (so the training set size is $mB$), the cost function is rewritten as $E(\boldsymbol{\theta}) = (1/m)\sum_{k=0}^{m-1}\hat{E}_k(\boldsymbol{\theta})$ with mini-batch costs denoted $\hat{E}_k(\boldsymbol{\theta}) = (1/B)\sum_{j=kB+1}^{kB+B} E_j(\boldsymbol{\theta})$. It was obtained in that work that after one epoch, the mean iterate of the algorithm, averaged over all possible shuffles of the batch indices, is close to the solution to $\dot{\boldsymbol{\theta}} = -\nabla \widetilde{E}_{SGD}(\boldsymbol{\theta})$, where the modified loss is given by $\widetilde{E}_{SGD}(\boldsymbol{\theta}) = E(\boldsymbol{\theta}) + h/(4m) \cdot \sum_{k=0}^{m-1}\left\|\nabla \hat{E}(\boldsymbol{\theta})\right\|^2$.

More recently, Ghosh et al. (2023) studied GD with heavy-ball momentum $\boldsymbol{\theta}^{(n+1)} = \boldsymbol{\theta}^{(n)} - h\nabla E(\boldsymbol{\theta}^{(n)}) + \beta(\boldsymbol{\theta}^{(n)} - \boldsymbol{\theta}^{(n-1)})$, where $\beta$ is the momentum parameter. In the full-batch setting, they proved that for $n$ large enough it is close to the continuous trajectory solving

$$\dot{\boldsymbol{\theta}} = (1-\beta)^{-1}\nabla E(\boldsymbol{\theta}) + \underbrace{h(1+\beta)(1-\beta)^{-3}\nabla\|\nabla E(\boldsymbol{\theta})\|^2/4}_{\text{implicit regularization}}. \tag{1}$$

Their main theorem also provides the analysis for the general mini-batch case.

In another recent work, Zhao et al. (2022) introduce a regularization term $\lambda \cdot \|\nabla E(\boldsymbol{\theta})\|$ to the loss function as a way to ensure finding flatter minima, improving generalization. The only difference between their term and the first-order correction coming from backward error analysis (up to a coefficient) is that the norm is not squared and regularization is applied on a per-batch basis.

Using backward error analysis to approximate the discrete dynamics with a modified ODE for adaptive algorithms such as RMSProp (Tieleman et al., 2012) and Adam (Kingma & Ba, 2015) is currently missing in the literature. Barrett & Dherin (2021) note that "it would be interesting to use backward error analysis to calculate the modified loss and implicit regularization for other widely used optimizers such as momentum, Adam and RMSprop". Smith et al. (2021) reiterate that they "anticipate that backward error analysis could also be used to clarify the role of finite learning rates in adaptive optimizers like Adam". Ghosh et al. (2023) agree that "RMSProp ... and Adam ..., albeit being powerful alternatives to SGD with faster convergence rates, are far from well-understood in the aspect of implicit regularization". In a similar context, in Appendix G to Miyagawa (2022) it is mentioned that "its [Adam's] counter term and discretization error are open questions".

This work fills the gap by conducting backward error analysis for (mini-batch, and full-batch as a special case) Adam and RMSProp. Our main contributions are listed below.

- In Theorem 3.1, we provide a global second-order in $h$ continuous ODE approximation to Adam in the general mini-batch setting. (A similar result for RMSProp is moved to the supplemental appendix.) For the full-batch special case, it was shown in prior work Ma et al. (2022) that the continuous-time limit of both these algorithms is a (perturbed by the numerical stability parameter $\varepsilon$) signGD flow $\dot{\boldsymbol{\theta}} = -\nabla E(\boldsymbol{\theta})/(|\nabla E(\boldsymbol{\theta})|+\varepsilon)$ component-wise; we make this more precise by finding a linear in $h$ "bias" term on the right.

- We analyze the full-batch case in more detail: see the summary in Section 2. We find that the bias term does something different from penalizing the two-norm of the loss gradient as in the case of GD: it either penalizes the perturbed one-norm of the loss gradient, defined as $\|\mathbf{v}\|_{1,\varepsilon} = \sum_{i=1}^{p} \sqrt{v_i^2 + \varepsilon}$, or, on the contrary, hinders its decrease (depending on hyperparameters and the training stage). Example 2.1 provides a backward error analysis result for heavy-ball momentum GD (Ghosh et al., 2023) as a special case.

- We provide numerical evidence consistent with our results. In particular, we observe that often penalizing the perturbed one-norm appears to improve generalization, and hindering the norm's decrease does the opposite. The bias we identify typically acts as anti-regularization, which is a previously unidentified possible explanation for often reported poorer generalization of adaptive gradient algorithms compared to other methods.

RELATED WORK

**Backward error analysis of first-order methods.** We provide the history of finding ODEs approximating different algorithms above in the introduction. Recently, there have been other applications of backward error analysis related to machine learning. Kunin et al. (2020) show that the approximating continuous-time trajectories satisfy conservation laws that are broken in discrete time. França et al. (2021) use backward error analysis while studying how to discretize continuous-time dynamical systems preserving stability and convergence rates. Rosca et al. (2021) find continuous-time approximations of discrete two-player differential games.

**Approximating gradient methods by differential equation trajectories.** Ma et al. (2022) prove that the trajectories of Adam and RMSProp are close to signGD dynamics, and investigate different training regimes of these algorithms empirically. SGD is approximated by stochastic differential equations and novel adaptive parameter adjustment policies are devised in Li et al. (2017).

**Implicit bias of first-order methods.** Soudry et al. (2018) prove that GD trained to classify linearly separable data with logistic loss converges to the direction of the max-margin vector (the solution to the hard margin SVM). This result has been extended to different loss functions in Nacson et al. (2019b), to SGD in Nacson et al. (2019c) and more

generic optimization methods in Gunasekar et al. (2018a), to the nonseparable case in Ji & Telgarsky (2018b), Ji & Telgarsky (2019). This line of research has been generalized to studying implicit biases of linear networks (Ji & Telgarsky, 2018a; Gunasekar et al., 2018b), homogeneous neural networks (Ji & Telgarsky, 2020; Nacson et al., 2019a; Lyu & Li, 2019). Woodworth et al. (2020) study the gradient flow of a diagonal linear network with squared loss and show that large initializations lead to minimum two-norm solutions while small initializations lead to minimum one-norm solutions. Even et al. (2023) extend this work to the case of non-zero step sizes and mini-batch training. Wang et al. (2021) prove that Adam and RMSProp maximize the margin of homogeneous neural networks.

**Generalization of adaptive methods.** Cohen et al. (2022) investigate the edge-of-stability regime of adaptive gradient algorithms and the effect of sharpness (the largest eigenvalue of the hessian) on generalization; Granziol (2020); Chen et al. (2021) observe that adaptive methods find sharper minima than SGD and Zhou et al. (2020); Xie et al. (2022) argue theoretically that it is the case. Jiang et al. (2022) introduce a statistic that measures the uniformity of the hessian diagonal and argue that adaptive gradient algorithms are biased towards making this statistic smaller. Keskar & Socher (2017) propose to improve generalization of adaptive methods by switching to SGD in the middle of training.

NOTATION

We denote the loss of the $k$th minibatch as a function of the network parameters $\boldsymbol{\theta} \in \mathbb{R}^p$ by $E_k(\boldsymbol{\theta})$, and in the full-batch setting we omit the index and write $E(\boldsymbol{\theta})$. $\nabla E$ means the gradient of $E$, and $\nabla$ with indices denotes partial derivatives, e.g. $\nabla_{ijs}E$ is a shortcut for $\frac{\partial^3 E}{\partial \theta_i \partial \theta_j \partial \theta_s}$. The norm without indices $\|\cdot\|$ is the two-norm of a vector, $\|\cdot\|_1$ is the one-norm and $\|\cdot\|_{1,\varepsilon}$ is the perturbed one-norm defined as $\|\mathbf{v}\|_{1,\varepsilon} = \sum_{i=1}^p \sqrt{v_i^2 + \varepsilon}$. (Of course, if $\varepsilon > 0$ the perturbed one-norm is not a norm, but $\varepsilon = 0$ makes it the one-norm.)

To provide the names and notations for hyperparameters, we define the algorithm below.

**Definition 1.1.** The *Adam* algorithm is an optimization algorithm with numerical stability hyperparameter $\varepsilon > 0$, squared gradient momentum hyperparameter $\rho \in (0,1)$, gradient momentum hyperparameter $\beta \in (0,1)$, initialization $\boldsymbol{\theta}^{(0)} \in \mathbb{R}^p$, $\boldsymbol{\nu}^{(0)} = \mathbf{0} \in \mathbb{R}^p$, $\mathbf{m}^{(0)} = \mathbf{0} \in \mathbb{R}^p$ and the following update rule: for each $n \geq 0$, $j \in \{1,\dots,p\}$

$$
\begin{aligned}
\nu_j^{(n+1)} &= \rho\nu_j^{(n)} + (1-\rho)\big(\nabla_j E_n(\boldsymbol{\theta}^{(n)})\big)^2, \quad m_j^{(n+1)} = \beta m_j^{(n)} + (1-\beta)\nabla_j E_n(\boldsymbol{\theta}^{(n)}), \\
\theta_j^{(n+1)} &= \theta_j^{(n)} - h\big(\nu_j^{(n+1)}/(1-\rho^{n+1}) + \varepsilon\big)^{-1/2}\big[m_j^{(n+1)}/(1-\beta^{n+1})\big].
\end{aligned}
\tag{2}
$$

**Remark 1.2.** Note that the numerical stability hyperparameter $\varepsilon > 0$, which is introduced in these algorithms to avoid division by zero, is inside the square root in our definition. This way we avoid division by zero in the derivative too: the first derivative of $x \mapsto \big(\sqrt{x+\varepsilon}\big)^{-1}$ is bounded for $x \geq 0$. This is useful for our analysis. In Theorems SA-2.4 and SA-4.4 in the appendix, the original versions of RMSProp and Adam are also tackled, though with an additional assumption which requires that no component of the gradient can come very close to zero in the region of interest. This is true only for the initial period of learning (whereas Theorem 3.1 tackles the whole period). Practitioners do not seem to make a distinction between the version with $\varepsilon$ inside vs. outside the square root: tutorials with both versions abound on machine learning related websites. Moreover, the popular Tensorflow variant of RMSProp has $\varepsilon$ inside the square root[1] even though in the documentation[2] Kingma & Ba (2015) is cited, where $\varepsilon$ is outside. Empirically we also observed that moving $\varepsilon$ inside or outside the square root does not change the behavior of Adam or RMSProp qualitatively.

---

[1]`https://github.com/keras-team/keras/blob/f9336cc5114b4a9429a242deb264b707379646b7/keras/optimizers/rmsprop.py#L190`

[2]`https://www.tensorflow.org/api_docs/python/tf/keras/optimizers/experimental/RMSprop`

## 2 Implicit bias of full-batch Adam: an informal summary

We are ready to informally describe our theoretical result (in the full-batch special case). Assume $E(\boldsymbol{\theta})$ is the loss, whose partial derivatives up to the fourth order are bounded. Let $\{\boldsymbol{\theta}^{(n)}\}$ be iterations of Adam as defined in Definition 1.1. We find an ODE whose solution trajectory $\tilde{\boldsymbol{\theta}}(t)$ is $h^2$-close to $\{\boldsymbol{\theta}^{(n)}\}$, meaning that for any time horizon $T > 0$ there is a constant $C$ such that for any step size $h \in (0, T)$ we have $\|\tilde{\boldsymbol{\theta}}(nh) - \boldsymbol{\theta}^{(n)}\| \leq Ch^2$ (for $n$ between 0 and $\lfloor T/h \rfloor$). The ODE is written the following way (up to terms that rapidly go to zero as $n$ grows): for the component number $j \in \{1, \ldots, p\}$

$$\dot{\tilde{\theta}}_j(t) = -\big(|\nabla_j E(\tilde{\boldsymbol{\theta}}(t))|^2 + \varepsilon\big)^{-1/2}\big(\nabla_j E(\tilde{\boldsymbol{\theta}}(t)) + \text{bias}\big) \tag{3}$$

with initial conditions $\tilde{\boldsymbol{\theta}}_j(0) = \boldsymbol{\theta}_j^{(0)}$ for all $j$, where the bias term is

$$\text{bias} := \frac{h}{2}\left\{\frac{1+\beta}{1-\beta} - \frac{1+\rho}{1-\rho} + \frac{1+\rho}{1-\rho} \cdot \frac{\varepsilon}{|\nabla_j E(\tilde{\boldsymbol{\theta}}(t))|^2 + \varepsilon}\right\}\nabla_j\big\|\nabla E(\tilde{\boldsymbol{\theta}}(t))\big\|_{1,\varepsilon}. \tag{4}$$

Depending on hyperparameters and the training stage, the bias term can take two extreme forms listed below, and during most of the training the reality is usually in between.

- If $\sqrt{\varepsilon}$ is **small** compared to all components of $\nabla E(\tilde{\boldsymbol{\theta}}(t))$, i.e. $\min_j |\nabla_j E(\tilde{\boldsymbol{\theta}}(t))| \gg \sqrt{\varepsilon}$, which is the case during the initial learning stage, then

$$\text{bias} = (h/2)\big\{(1+\beta)/(1-\beta) - (1+\rho)/(1-\rho)\big\}\nabla_j\big\|\nabla E(\tilde{\boldsymbol{\theta}}(t))\big\|_{1,\varepsilon}. \tag{5}$$

For small $\varepsilon$, the perturbed one-norm is indistinguishable from the usual one-norm, and for $\beta > \rho$ it is penalized (in much the same way as the squared two-norm is implicitly penalized in the case of GD), but for $\rho > \beta$ its decrease is actually hindered by this term (so the bias is opposite to penalization). The ODE in (3) approximately becomes

$$\dot{\tilde{\theta}}_j(t) = -\frac{\nabla_j \widetilde{E}(\tilde{\boldsymbol{\theta}}(t))}{|\nabla_j E(\tilde{\boldsymbol{\theta}}(t))|}, \qquad \widetilde{E}(\boldsymbol{\theta}) = E(\boldsymbol{\theta}) + \frac{h}{2}\left\{\frac{1+\beta}{1-\beta} - \frac{1+\rho}{1-\rho}\right\}\big\|\nabla E(\boldsymbol{\theta})\big\|_1. \tag{6}$$

- If $\sqrt{\varepsilon}$ is **large** compared to all gradient components, i.e. $\max_j |\nabla_j E(\tilde{\boldsymbol{\theta}}(t))| \ll \sqrt{\varepsilon}$, which may happen during the later learning stage, the fraction in (4) with $\varepsilon$ in the numerator approaches one, the dependence on $\rho$ cancels out, and

$$\big\|\nabla E(\tilde{\boldsymbol{\theta}}(t))\big\|_{1,\varepsilon} \approx \sum_{i=1}^{p} \sqrt{\varepsilon}\big(1 + |\nabla_i E(\tilde{\boldsymbol{\theta}}(t))|^2/(2\varepsilon)\big) = p\sqrt{\varepsilon} + \frac{1}{2\sqrt{\varepsilon}}\big\|\nabla E(\tilde{\boldsymbol{\theta}}(t))\big\|^2. \tag{7}$$

In other words, $\|\cdot\|_{1,\varepsilon}$ becomes $\|\cdot\|^2/(2\sqrt{\varepsilon})$ up to an additive constant, giving

$$\text{bias} = \big(4\sqrt{\varepsilon}\big)^{-1}(1-\beta)^{-1}(1+\beta)\nabla_j\|\nabla E(\tilde{\boldsymbol{\theta}}(t))\|^2.$$

The form of the ODE in this case is

$$\dot{\tilde{\theta}}_j(t) = -\nabla_j \widetilde{E}(\tilde{\boldsymbol{\theta}}(t)), \qquad \widetilde{E}(\boldsymbol{\theta}) = \frac{1}{\sqrt{\varepsilon}}\left(E(\tilde{\boldsymbol{\theta}}(t)) + \frac{h}{4\sqrt{\varepsilon}}\frac{1+\beta}{1-\beta}\|\nabla E(\tilde{\boldsymbol{\theta}}(t))\|^2\right). \tag{8}$$

These two extreme cases are summarized in Table 1. In Figure 1, we use the one-dimensional ($p = 1$) case to illustrate what kind of term is being implicitly penalized.

Since in practice $\varepsilon$ is usually small, during most of the training Adam is better described by the first extreme case. It is clear from (6) that, if $\rho > \beta$, this bias term does not provide the same kind of implicit regularization as the correction term in (1) does. In fact, it provides the opposite of regularization. This phenomenon may partially explain why adaptive gradient methods have been reported to generalize worse than non-adaptive ones (Cohen et al. (2022) and references therein), and it may be a previously unknown perspective on why they are biased towards higher-curvature regions and find "sharper" minima. Moreover, (6) suggests that decreasing $\rho$ and increasing $\beta$ moves the trajectory towards regions with lower "norm",

| | $\varepsilon$ "small" | $\varepsilon$ "large" |
|---|---|---|
| $\beta \geq \rho$ | $\|\nabla E(\boldsymbol{\theta})\|_1$-penalized | $\|\nabla E(\boldsymbol{\theta})\|_2^2$-penalized |
| $\rho > \beta$ | $-\|\nabla E(\boldsymbol{\theta})\|_1$-penalized | $\|\nabla E(\boldsymbol{\theta})\|_2^2$-penalized |

Table 1: Implicit bias of Adam: special cases. "Small" and "large" are in relation to squared gradient components (Adam in the latter case is close to GD with momentum).

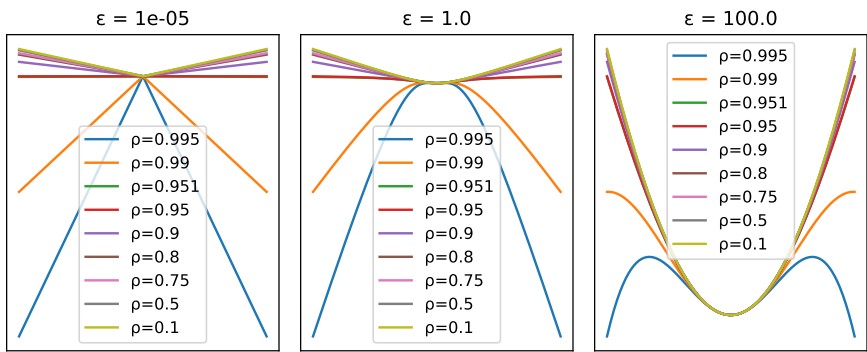

Figure 1: The graphs of $x \mapsto \int_0^x \left\{ \frac{1+\beta}{1-\beta} - \frac{1+\rho}{1-\rho} + \frac{1+\rho}{1-\rho} \cdot \frac{\varepsilon}{y^2+\varepsilon} \right\} \mathrm{d}\sqrt{\varepsilon + y^2}$ with $\beta = 0.95$.

which may improve the test error. We stress that the link between (anti-)penalizing the one-norm and sharpness of minima is speculative, and even the connection between sharpness and generalization is not clear-cut (Andriushchenko et al., 2023).

This overview also applies to RMSProp by setting $\beta = 0$. See Theorem SA-3.4 in the appendix for the formal result.

**Example 2.1** (Backward Error Analysis for GD with Heavy-ball Momentum). Assume $\varepsilon$ is large compared to all squared gradient components during the whole training process, so that the form of the ODE is approximated by (8). Since Adam with a large $\varepsilon$ and after a certain number of iterations approximates SGD with heavy-ball momentum with step size $h(1-\beta)/\sqrt{\varepsilon}$, a linear step size change (and corresponding time change) gives exactly the equations in Theorem 4.1 of Ghosh et al. (2023). Taking $\beta = 0$ (no momentum), we get the implicit regularization of GD from Barrett & Dherin (2021).

## 3 Main result: ODE approximating mini-batch Adam

We only make one assumption, which is standard in the literature: the loss $E_k$ for each mini-batch is 4 times continuously differentiable, and partial derivatives of $E_k$ up to order 4 are bounded, i.e. there is a positive constant $M$ such that for $\boldsymbol{\theta}$ in the region of interest

$$\sup_k \left\{ \sup_i |\nabla_i E_k(\boldsymbol{\theta})| \vee \sup_{i,j} |\nabla_{ij} E_k(\boldsymbol{\theta})| \vee \sup_{i,j,s} |\nabla_{ijs} E_k(\boldsymbol{\theta})| \vee \sup_{i,j,s,r} |\nabla_{ijsr} E_k(\boldsymbol{\theta})| \right\} \leq M. \quad (9)$$

**Theorem 3.1.** *Assume* (9) *holds. Let* $\{\boldsymbol{\theta}^{(n)}\}$ *be iterations of Adam as defined in Definition 1.1,* $\tilde{\boldsymbol{\theta}}(t)$ *be the continuous solution to the piecewise ODE*

$$
\begin{aligned}
\dot{\tilde{\theta}}_j(t) = & -\frac{M_j^{(n)}(\tilde{\boldsymbol{\theta}}(t))}{R_j^{(n)}(\tilde{\boldsymbol{\theta}}(t))} \\
& + h\left( \frac{M_j^{(n)}(\tilde{\boldsymbol{\theta}}(t))\left(2P_j^{(n)}(\tilde{\boldsymbol{\theta}}(t)) + \bar{P}_j^{(n)}(\tilde{\boldsymbol{\theta}}(t))\right)}{2R_j^{(n)}(\tilde{\boldsymbol{\theta}}(t))^3} - \frac{2L_j^{(n)}(\tilde{\boldsymbol{\theta}}(t)) + \bar{L}_j^{(n)}(\tilde{\boldsymbol{\theta}}(t))}{2R_j^{(n)}(\tilde{\boldsymbol{\theta}}(t))} \right)
\end{aligned}
\quad (10)
$$

*for $t \in [nh, (n+1)h]$ with the initial condition $\tilde{\boldsymbol{\theta}}(0) = \boldsymbol{\theta}^{(0)}$, where*

$$R_j^n(\boldsymbol{\theta}) := \left( \left(1 - \rho^{n+1}\right)^{-1} \sum_{k=0}^{n} \rho^{n-k}(1-\rho)(\nabla_j E_k(\boldsymbol{\theta}))^2 + \varepsilon \right)^{1/2},$$

$$M_j^{(n)}(\boldsymbol{\theta}) := \left(1 - \beta^{n+1}\right)^{-1} \sum_{k=0}^{n} \beta^{n-k}(1-\beta)\nabla_j E_k(\boldsymbol{\theta}),$$

$$L_j^{(n)}(\boldsymbol{\theta}) := \left(1 - \beta^{n+1}\right)^{-1} \sum_{k=0}^{n} \beta^{n-k}(1-\beta) \sum_{i=1}^{p} \nabla_{ij} E_k(\boldsymbol{\theta}) \sum_{l=k}^{n-1} M_i^{(l)}(\boldsymbol{\theta})/R_i^{(l)}(\boldsymbol{\theta}),$$

$$\bar{L}_j^{(n)}(\boldsymbol{\theta}) := \left(1 - \beta^{n+1}\right)^{-1} \sum_{k=0}^{n} \beta^{n-k}(1-\beta) \sum_{i=1}^{p} \nabla_{ij} E_k(\boldsymbol{\theta}) M_i^{(n)}(\boldsymbol{\theta})/R_i^{(n)}(\boldsymbol{\theta}),$$

$$P_j^{(n)}(\boldsymbol{\theta}) := \left(1 - \rho^{n+1}\right)^{-1} \sum_{k=0}^{n} \rho^{n-k}(1-\rho)\nabla_j E_k(\boldsymbol{\theta}) \sum_{i=1}^{p} \nabla_{ij} E_k(\boldsymbol{\theta}) \sum_{l=k}^{n-1} M_i^{(l)}(\boldsymbol{\theta})/R_i^{(l)}(\boldsymbol{\theta}),$$

$$\bar{P}_j^{(n)}(\boldsymbol{\theta}) := \left(1 - \rho^{n+1}\right)^{-1} \sum_{k=0}^{n} \rho^{n-k}(1-\rho)\nabla_j E_k(\boldsymbol{\theta}) \sum_{i=1}^{p} \nabla_{ij} E_k(\boldsymbol{\theta}) M_i^{(n)}(\boldsymbol{\theta})/R_i^{(n)}(\boldsymbol{\theta}).$$

*Then, for any fixed positive time horizon $T > 0$ there exists a constant $C$ such that for any step size $h \in (0, T)$ we have $\left\| \tilde{\boldsymbol{\theta}}(nh) - \boldsymbol{\theta}^{(n)} \right\| \le Ch^2$ for $n \in \{0, \dots, \lfloor T/h \rfloor\}$.*

The proof is in the appendix (this is Theorem SA-5.4; see SA-1 for the overview of the contents). To help the reader understand the argument, apart from the full proof, we include an informal derivation in Section SA-9 of the appendix, and we provide an even briefer sketch of this derivation here.

Our goal is to find such a sequence $\tilde{\boldsymbol{\theta}}(t_n)$, where $t_n := nh$, that $\tilde{\theta}_j(t_{n+1}) = \tilde{\theta}_j(t_n) - hT_\beta T_\rho^{-1/2} + O(h^3)$, denoting $T_\beta := (1 - \beta^{n+1})^{-1} \sum_{k=0}^{n} \beta^{n-k}(1-\beta)\nabla_j E_k(\tilde{\boldsymbol{\theta}}(t_k))$ and $T_\rho := (1 - \rho^{n+1})^{-1} \sum_{k=0}^{n} \rho^{n-k}(1-\rho)(\nabla_j E_k(\tilde{\boldsymbol{\theta}}(t_k)))^2 + \varepsilon$. Ignoring the terms of order higher than one, we can take a first-order approximation for granted: $\tilde{\theta}_j(t_{n+1}) = \tilde{\theta}_j(t_n) - hA(\tilde{\boldsymbol{\theta}}(t_n)) + O(h^2)$ with $A(\boldsymbol{\theta}) := M_j^{(n)}(\boldsymbol{\theta})/R_j^{(n)}(\boldsymbol{\theta})$. The challenge is to make this more precise by finding an equality of the form $\tilde{\theta}_j(t_{n+1}) = \tilde{\theta}_j(t_n) - hA(\tilde{\boldsymbol{\theta}}(t_n)) + h^2 B(\tilde{\boldsymbol{\theta}}(t_n)) + O(h^3)$, where $B(\cdot)$ is a known function, because this is a numerical iteration to which standard backward error analysis (Chapter IX in Ernst Hairer & Wanner (2006)) can be applied.

Using the Taylor series, we can write

$$\nabla_j E_k(\tilde{\boldsymbol{\theta}}(t_{n-1})) = \nabla_j E_k(\tilde{\boldsymbol{\theta}}(t_n)) + \sum_{i=1}^{p} \nabla_{ij} E_k(\tilde{\boldsymbol{\theta}}(t_n))\{\tilde{\theta}_i(t_{n-1}) - \tilde{\theta}_i(t_n)\} + O(h^2)$$

$$= \nabla_j E_k(\tilde{\boldsymbol{\theta}}(t_n)) + h\sum_{i=1}^{p} \nabla_{ij} E_k(\tilde{\boldsymbol{\theta}}(t_n))M_j^{(n-1)}(\tilde{\boldsymbol{\theta}}(t_n))/R_j^{(n-1)}(\tilde{\boldsymbol{\theta}}(t_n)) + O(h^2),$$

where in the last equality we just replaced $t_{n-1}$ with $t_n$ in the $h$-term since it only affects higher-order terms. Doing this again for steps $n-1$, $n-2$ and so on, and adding the resulting equations, will give for $k < n$

$$\nabla_j E_k(\tilde{\boldsymbol{\theta}}(t_k)) = \nabla_j E_k(\tilde{\boldsymbol{\theta}}(t_n)) + h\sum_{i=1}^{p} \nabla_{ij} E_k(\tilde{\boldsymbol{\theta}}(t_n)) \sum_{l=k}^{n-1} M_i^{(l)}(\tilde{\boldsymbol{\theta}}(t_n))/R_i^{(l)}(\tilde{\boldsymbol{\theta}}(t_n)) + O(h^2),$$

where we could ignore that $n - k$ is not bounded because of exponential averaging. Taking the square of this formal power series (in $h$), summing up over $k$, and using the expression for the inverse square root of a formal power series $\sum_{r=0}^{\infty} a_r h^r$, gives us an expansions of $T_\rho^{-1/2}$, and a similar process provides an expansion for $T_\beta$. Combining them leads to an expression for $B(\cdot)$.

**Remark 3.2.** In the *full-batch* setting $E_k \equiv E$, the terms in Theorem 3.1 simplify to

$$R_j^{(n)}(\boldsymbol{\theta}) = (|\nabla_j E(\boldsymbol{\theta})|^2 + \varepsilon)^{1/2}, \qquad\qquad M_j^{(n)}(\boldsymbol{\theta}) = \nabla_j E(\boldsymbol{\theta}),$$

$$L_j^{(n)}(\boldsymbol{\theta}) = \left[\frac{\beta}{1-\beta} - \frac{(n+1)\beta^{n+1}}{1-\beta^{n+1}}\right]\bar{L}_j^{(n)}(\boldsymbol{\theta}), \qquad \bar{L}_j^{(n)}(\boldsymbol{\theta}) = \nabla_j \|\nabla E(\boldsymbol{\theta})\|_{1,\varepsilon},$$

$$P_j^{(n)}(\boldsymbol{\theta}) = \left[\frac{\rho}{1-\rho} - \frac{(n+1)\rho^{n+1}}{1-\rho^{n+1}}\right]\bar{P}_j^{(n)}(\boldsymbol{\theta}), \qquad \bar{P}_j^{(n)}(\boldsymbol{\theta}) = \nabla_j E(\boldsymbol{\theta})\nabla_j\|\nabla E(\boldsymbol{\theta})\|_{1,\varepsilon}.$$

If the iteration number $n$ is large, (10) rapidly becomes as described in (3) and (4).

## 4 ILLUSTRATION: SIMPLE BILINEAR MODEL

We now analyze the effect of the first-order term for Adam in the same model as Barrett & Dherin (2021) and Ghosh et al. (2023) have studied. Namely, assume the parameter $\boldsymbol{\theta} = (\theta_1, \theta_2)$ is 2-dimensional, and the loss is given by $E(\boldsymbol{\theta}) := 1/2(3/2 - 2\theta_1\theta_2)^2$. The loss is minimized on the hyperbola $\theta_1\theta_2 = 3/4$. We graph the trajectories of Adam in this case: Figure 2 shows that increasing $\beta$ forces the trajectory to the region with smaller $\|\nabla E(\boldsymbol{\theta})\|_1$, and increasing $\rho$ does the opposite. Figure 3 shows that increasing the learning rate moves Adam towards the region with smaller $\|\nabla E(\boldsymbol{\theta})\|_1$ if $\beta > \rho$ (just like in the case of GD, except the norm is different if $\varepsilon$ is small compared to gradient components), and does the opposite if $\rho > \beta$. All these observations are exactly what Theorem 3.1 predicts.

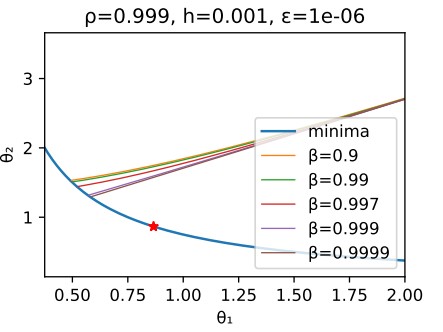 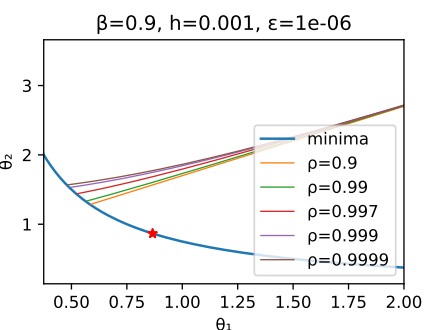

Figure 2: Increasing $\beta$ moves the trajectory of Adam towards the regions with smaller one-norm of the gradient (if $\varepsilon$ is sufficiently small); increasing $\rho$ does the opposite. The cross denotes the limit point of gradient one-norm minimizers on the level sets $4\theta_1\theta_2 - 3 = c$. All Adam trajectories start at $(2.8, 3.5)$.

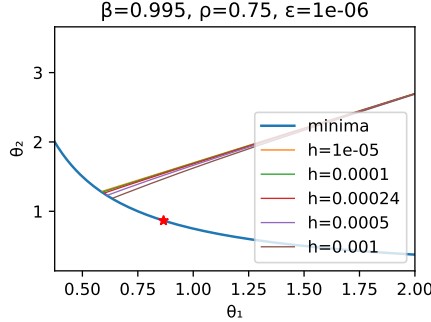 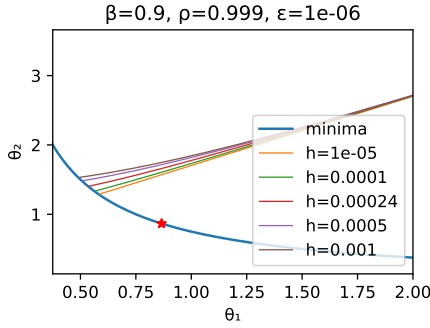

Figure 3: The setting is the same as in Figure 2. Increasing the learning rate moves the Adam trajectory towards the regions with smaller one-norm of the gradient if $\beta$ is significantly larger than $\rho$ and does the opposite if $\rho$ is larger than $\beta$.

## 5 Numerical experiments

We offer some preliminary empirical evidence of how the bias term shows up in deep neural networks.

Ma et al. (2022) divides training regimes of Adam into three categories: the spike regime when $\rho$ is much larger than $\beta$, in which the training loss curve contains very large spikes and the training is obviously unstable; the (stable) oscillation regime when $\rho$ is sufficiently close to $\beta$, in which the loss curve contains fast and small oscillations; the divergence regime when $\beta$ is much larger than $\rho$, in which Adam diverges. We exclude the last regime. In the spike regime, the loss spikes to large values at irregular intervals. This has been observed in the context of large transformers, and mitigation strategies have been proposed in Chowdhery et al. (2022) and Molybog et al. (2023). Since it is unlikely that an unstable Adam trajectory can be meaningfully approximated by a smooth ODE solution, we exclude the spike regime as well, and only consider the oscillation regime, which Ma et al. (2022) recommend to use in practice. We do this by making $\beta$ and $\rho$ not too far apart, because for clean experiments we do not do any explicit regularization, learning rate decay or stochastic batching, and decreasing $h$ increases training time and weakens the bias we identify.

We train Resnet-50 on the CIFAR-10 dataset with full-batch Adam. Figure 4 shows that in the stable oscillation regime increasing $\rho$ seems to increase the perturbed one-norm (consistent with our analysis: the smaller $\rho$, the more this "norm" is penalized) and decrease the test accuracy. The opposite to the latter was noticed in Cohen et al. (2022), which we think is the case in the spike regime (see above). Figure 5 shows that increasing $\beta$ seems to decrease the perturbed one-norm (consistent with our analysis: the larger $\beta$, the more this norm is penalized) and increase the test accuracy. The picture confirms the finding in Ghosh et al. (2023) (for the momentum parameter in momentum GD).

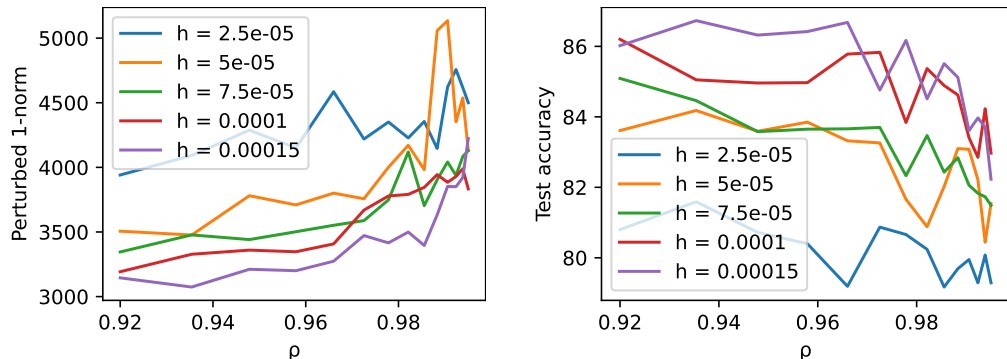

Figure 4: Resnet-50 on CIFAR-10 trained with full-batch Adam, $\varepsilon = 10^{-8}$, $\beta = 0.99$. As $\rho$ increases, the norm seems to rise and the test accuracy seems to fall (in the stable regime of training). The test accuracies plotted here are maximal after more than 3600 epochs. The perturbed norms are also maximal after excluding the initial training period (i.e., the plotted "norms" are at peaks of the "hills" described in Section 5). Additional evidence and more details are provided in Section SA-8 of the Appendix.

We obtain a more detailed picture of the perturbed norm's behavior by training Resnet-101 on CIFAR-10 and CIFAR-100 with full-batch Adam. Figure 6 shows the graphs of $\|\nabla E\|_{1,\varepsilon}$ as functions of the epoch number. The "norm" decreases, then rises again, and then decreases further until it flatlines. Throughout most of the training, the larger $\beta$ the smaller the "norm". The "hills" of the "norm" curves are higher with smaller $\beta$ and larger $\rho$. This is consistent with our analysis because the larger $\rho$ compared to $\beta$, the more $\|\nabla E\|_{1,\varepsilon}$ is prevented from falling by the bias term. Note that the perturbed one-norm cannot be near-zero at the end of training because it is bounded from below by $p\sqrt{\varepsilon}$.

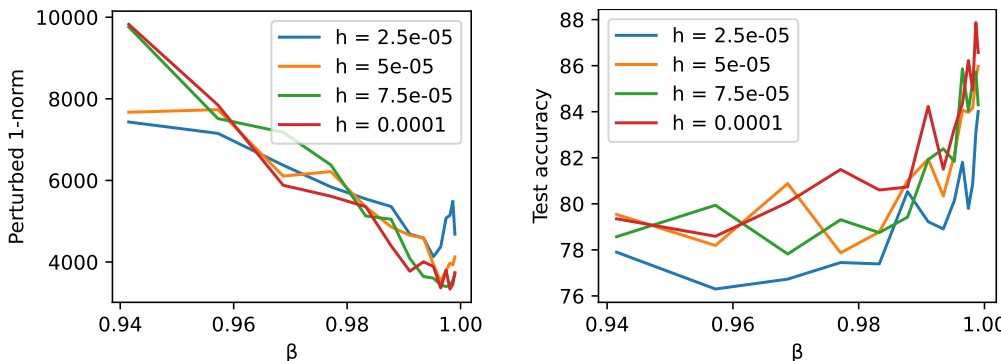

Figure 5: Resnet-50 on CIFAR-10 trained with full-batch Adam, $\rho = 0.999$, $\varepsilon = 10^{-8}$. The perturbed one-norm seems to fall as $\beta$ increases, and the test accuracy seems to rise. Both metrics are calculated as in Figure 4.

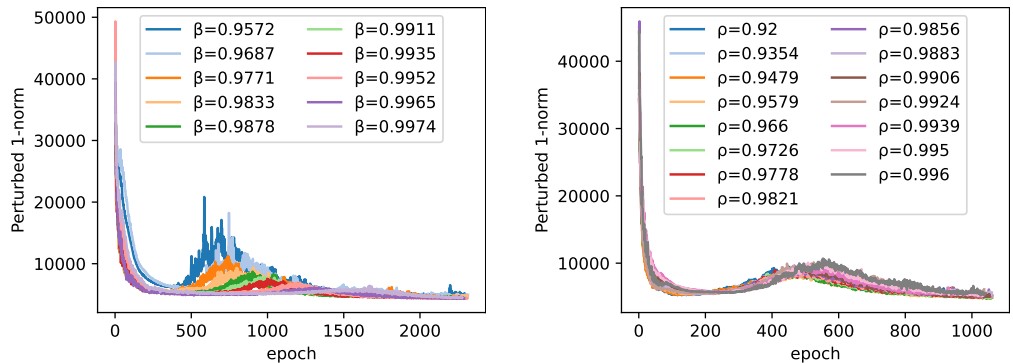

Figure 6: Plots of $\|\nabla E\|_{1,\varepsilon}$ after each epoch for a full-batch Adam, $h = 10^{-4}, \varepsilon = 10^{-8}$. Left: Resnet-101 on CIFAR-10, $\rho = 0.999$. Right: Resnet-101 on CIFAR-100, $\beta = 0.97$.

## 6  FUTURE DIRECTIONS

As far as we know, the assumption similar to (9) is explicitly or implicitly present in all previous work on backward error analysis of gradient-based machine learning algorithms. There is evidence that large-batch algorithms often operate at the edge of stability (Cohen et al., 2021; 2022), in which the largest eigenvalue of the hessian can be large, making it unclear whether the higher-order partial derivatives can safely be assumed bounded near optimality. However, as Smith et al. (2021) point out, in the mini-batch setting backward error analysis can be more accurate. We leave a qualitative analysis of the behavior of first-order terms in Theorem 3.1 in the mini-batch case as a future direction.

Relatedly, Adam is known to not always generalize worse than SGD: for transformers, Adam often outperforms (Zhang et al., 2020; Kumar et al., 2022). Moreover, for NLP tasks we may spend a long time training close to an interpolating solution. Though our analysis suggests that in the latter regime the anti-regularization effect disappears, more work is needed to connect the implicit bias to the training dynamics of transformers.

Also, the constant $C$ in Theorem 3.1 goes to infinity as $\varepsilon$ goes to zero. Theoretically, our proof does not exclude the case where for very small $\varepsilon$ the trajectory of the piecewise ODE is only close to the Adam trajectory for small, suboptimal learning rates, at least at later stages of learning. (For the initial learning period, this is not a problem.) It appears to also be true of Proposition 1 in Ma et al. (2022) (zeroth-order approximation by sign-GD). This is especially noticeable in the large-spike regime of training (see Section 5) which, despite being obviously unstable, can still lead to acceptable test errors. It would be interesting to investigate this regime in detail.

## Reproducibility Statement

Detailed proofs to the theoretical claims in the paper are available in the appendix and are referenced in the main text. All hyperparameters used for experiments are given either in figure captions or the pictures illustrating our empirical results, and the details about our model architectures and training approaches are available in the appendix (Section SA-8).

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
