# Supplementary Material for the Manuscript "On the Implicit Bias of Adam"

November 22, 2023

## Contents

## SA-1  Overview

**SA-1.1.**  This appendix provides some omitted details and proofs.

We consider two algorithms: RMSProp and Adam, and two versions of each algorithm (with the numerical stability $\varepsilon$ parameter inside and outside of the square root in the denominator). This means there are four main theorems: Theorem SA-2.4, Theorem SA-3.4, Theorem SA-4.4 and Theorem SA-5.4, each residing in the section completely devoted to one algorithm. The simple induction argument taken from [1], essentially the same for each of these theorems, is based on an auxiliary result whose corresponding versions are Theorem SA-2.3, Theorem SA-3.3, Theorem SA-4.3 and Theorem SA-5.3. The proof of this result is also elementary but long, and it is done by a series of lemmas in Section SA-6 and Section SA-7, culminating in Section SA-7.6. Out of these four, we only prove Theorem SA-2.3 since the other three results are proven in the same way with obvious changes.

Section SA-8 contains some details about the numerical experiments.

**SA-1.2 Notation.**  We denote the loss of the $k$th minibatch as a function of the network parameters $\boldsymbol{\theta} \in \mathbb{R}^p$ by $E_k(\boldsymbol{\theta})$, and in the full-batch setting we omit the index and write $E(\boldsymbol{\theta})$. As usual, $\nabla E$ means the gradient of $E$, and nabla with indices means partial derivatives, e. g. $\nabla_{ijs} E$ is a shortcut for $\frac{\partial^3 E}{\partial \theta_i \partial \theta_j \partial \theta_s}$.

The letter $T > 0$ will always denote a finite time horizon of the ODEs, $h$ will always denote the training step size, and we will replace $nh$ with $t_n$ when convenient, where $n \in \{0, 1, \ldots\}$ is the step number. *We will use the same notation* for the iteration of the discrete algorithm $\left\{ \boldsymbol{\theta}^{(k)} \right\}_{k \in \mathbb{Z}_{\geq 0}}$, the piecewise ODE solution $\tilde{\boldsymbol{\theta}}(t)$ and some auxiliary terms for each of the four algorithms: see Definition SA-2.1, Definition SA-

3.1, Definition SA-4.1, Definition SA-5.1. This way, we avoid cluttering the notation significantly. We are careful to reference the relevant definition in all theorem statements.

## SA-2   RMSProp with $\varepsilon$ outside the square root

**Definition SA-2.1.** In this section, for some $\boldsymbol{\theta}^{(0)} \in \mathbb{R}^p$, $\nu^{(0)} = \mathbf{0} \in \mathbb{R}^p$, $\rho \in (0, 1)$, let the sequence of $p$-vectors $\left\{\boldsymbol{\theta}^{(k)}\right\}_{k \in \mathbb{Z}_{\geq 0}}$ be defined for $n \geq 0$ by

$$
\begin{aligned}
\nu_j^{(n+1)} &= \rho \nu_j^{(n)} + (1 - \rho)\left(\nabla_j E_n\left(\boldsymbol{\theta}^{(n)}\right)\right)^2, \\
\theta_j^{(n+1)} &= \theta_j^{(n)} - \frac{h}{\sqrt{\nu_j^{(n+1)} + \varepsilon}} \nabla_j E_n\left(\boldsymbol{\theta}^{(n)}\right).
\end{aligned}
\tag{SA-2.1}
$$

Let $\tilde{\boldsymbol{\theta}}(t)$ be defined as a continuous solution to the piecewise ODE

$$
\begin{aligned}
\dot{\tilde{\theta}}_j(t) = &-\frac{\nabla_j E_n\left(\tilde{\boldsymbol{\theta}}(t)\right)}{R_j^{(n)}\left(\tilde{\boldsymbol{\theta}}(t)\right) + \varepsilon} \\
&+ h\left(\frac{\nabla_j E_n\left(\tilde{\boldsymbol{\theta}}(t)\right)\left(2 P_j^{(n)}\left(\tilde{\boldsymbol{\theta}}(t)\right) + \bar{P}_j^{(n)}\left(\tilde{\boldsymbol{\theta}}(t)\right)\right)}{2\left(R_j^{(n)}\left(\tilde{\boldsymbol{\theta}}(t)\right) + \varepsilon\right)^2 R_j^{(n)}\left(\tilde{\boldsymbol{\theta}}(t)\right)} - \frac{\sum_{i=1}^p \nabla_{ij} E_n\left(\tilde{\boldsymbol{\theta}}(t)\right) \frac{\nabla_i E_n\left(\tilde{\boldsymbol{\theta}}(t)\right)}{R_i^{(n)}\left(\tilde{\boldsymbol{\theta}}(t)\right) + \varepsilon}}{2\left(R_j^{(n)}(\tilde{\boldsymbol{\theta}}(t)) + \varepsilon\right)}\right)
\end{aligned}
\tag{SA-2.2}
$$

with the initial condition $\tilde{\boldsymbol{\theta}}(0) = \boldsymbol{\theta}^{(0)}$, where $\mathbf{R}^{(n)}(\boldsymbol{\theta})$, $\mathbf{P}^{(n)}(\boldsymbol{\theta})$ and $\bar{\mathbf{P}}^{(n)}(\boldsymbol{\theta})$ are $p$-dimensional functions with components

$$
\begin{aligned}
R_j^{(n)}(\boldsymbol{\theta}) &:= \sqrt{\sum_{k=0}^n \rho^{n-k}(1 - \rho)\left(\nabla_j E_k(\boldsymbol{\theta})\right)^2}, \\
P_j^{(n)}(\boldsymbol{\theta}) &:= \sum_{k=0}^n \rho^{n-k}(1 - \rho)\nabla_j E_k(\boldsymbol{\theta}) \sum_{i=1}^p \nabla_{ij} E_k(\boldsymbol{\theta}) \sum_{l=k}^{n-1} \frac{\nabla_i E_l(\boldsymbol{\theta})}{R_i^{(l)}(\boldsymbol{\theta}) + \varepsilon}, \\
\bar{P}_j^{(n)}(\boldsymbol{\theta}) &:= \sum_{k=0}^n \rho^{n-k}(1 - \rho)\nabla_j E_k(\boldsymbol{\theta}) \sum_{i=1}^p \nabla_{ij} E_k(\boldsymbol{\theta}) \frac{\nabla_i E_n(\boldsymbol{\theta})}{R_i^{(n)}(\boldsymbol{\theta}) + \varepsilon}.
\end{aligned}
$$

**Assumption SA-2.2.**
1. For some positive constants $M_1$, $M_2$, $M_3$, $M_4$ we have

$$
\begin{aligned}
\sup_i \sup_k \sup_{\boldsymbol{\theta}} \left|\nabla_i E_k(\boldsymbol{\theta})\right| &\leq M_1, \\
\sup_{i,j} \sup_k \sup_{\boldsymbol{\theta}} \left|\nabla_{ij} E_k(\boldsymbol{\theta})\right| &\leq M_2, \\
\sup_{i,j,s} \sup_k \sup_{\boldsymbol{\theta}} \left|\nabla_{ijs} E_k(\boldsymbol{\theta})\right| &\leq M_3, \\
\sup_{i,j,s,r} \sup_k \sup_{\boldsymbol{\theta}} \left|\nabla_{ijsr} E_k(\boldsymbol{\theta})\right| &\leq M_4.
\end{aligned}
$$

2. For some $R > 0$ we have for all $n \in \left\{0, 1, \ldots, \lfloor T/h \rfloor\right\}$

$$
R_j^{(n)}\left(\tilde{\boldsymbol{\theta}}(t_n)\right) \geq R, \quad \sum_{k=0}^n \rho^{n-k}(1 - \rho)\left(\nabla_j E_k\left(\tilde{\boldsymbol{\theta}}(t_k)\right)\right)^2 \geq R^2,
$$

where $\tilde{\boldsymbol{\theta}}(t)$ is defined in Definition SA-2.1.

**Theorem SA-2.3** (RMSProp with $\varepsilon$ outside: local error bound)**.** *Suppose Assumption SA-2.2 holds. Then for all $n \in \{0, 1, \ldots, \lfloor T/h \rfloor\}$*

$$\left| \tilde{\theta}_j(t_{n+1}) - \tilde{\theta}_j(t_n) + h \frac{\nabla_j E_n\left(\tilde{\boldsymbol{\theta}}(t_n)\right)}{\sqrt{\sum_{k=0}^{n} \rho^{n-k}(1-\rho)\left(\nabla_j E_k\left(\tilde{\boldsymbol{\theta}}(t_k)\right)\right)^2 + \varepsilon}} \right| \leq C_1 h^3$$

*for a positive constant $C_1$ depending on $\rho$.*

The proof of Theorem SA-2.3 is conceptually simple but very technical, and we delay it until Section SA-7. For now assuming it as given and combining it with a simple induction argument gives a global error bound which follows.

**Theorem SA-2.4** (RMSProp with $\varepsilon$ outside: global error bound)**.** *Suppose Assumption SA-2.2 holds, and*

$$\sum_{k=0}^{n} \rho^{n-k}(1-\rho)\left(\nabla_j E_k\left(\boldsymbol{\theta}^{(k)}\right)\right)^2 \geq R^2$$

*for $\left\{\boldsymbol{\theta}^{(k)}\right\}_{k \in \mathbb{Z}_{\geq 0}}$ defined in Definition SA-2.1. Then there exist positive constants $d_1$, $d_2$, $d_3$ such that for all $n \in \{0, 1, \ldots, \lfloor T/h \rfloor\}$*

$$\|\mathbf{e}_n\| \leq d_1 e^{d_2 nh} h^2 \quad \text{and} \quad \|\mathbf{e}_{n+1} - \mathbf{e}_n\| \leq d_3 e^{d_2 nh} h^3,$$

*where $\mathbf{e}_n := \tilde{\boldsymbol{\theta}}(t_n) - \boldsymbol{\theta}^{(n)}$. The constants can be defined as*

$$d_1 := C_1,$$

$$d_2 := \left[1 + \frac{M_2 \sqrt{p}}{R + \varepsilon}\left(\frac{M_1^2}{R(R + \varepsilon)} + 1\right) d_1\right]\sqrt{p},$$

$$d_3 := C_1 d_2.$$

*Proof.* We will show this by induction over $n$, the same way an analogous bound is shown in [1].

The base case is $n = 0$. Indeed, $\mathbf{e}_0 = \tilde{\boldsymbol{\theta}}(0) - \boldsymbol{\theta}^{(0)} = \mathbf{0}$. Then the $j$th component of $\mathbf{e}_1 - \mathbf{e}_0$ is

$$[\mathbf{e}_1 - \mathbf{e}_0]_j = [\mathbf{e}_1]_j = \tilde{\theta}_j(t_1) - \theta_j^{(0)} + \frac{h\nabla_j E_0\left(\boldsymbol{\theta}^{(0)}\right)}{\sqrt{(1-\rho)\left(\nabla_j E_0\left(\boldsymbol{\theta}^{(0)}\right)\right)^2 + \varepsilon}}$$

$$= \tilde{\theta}_j(t_1) - \tilde{\theta}_j(t_0) + \frac{h\nabla_j E_0\left(\tilde{\boldsymbol{\theta}}(t_0)\right)}{\sqrt{(1-\rho)\left(\nabla_j E_0\left(\tilde{\boldsymbol{\theta}}(t_0)\right)\right)^2 + \varepsilon}}.$$

By Theorem SA-2.3, the absolute value of the right-hand side does not exceed $C_1 h^3$, which means $\|\mathbf{e}_1 - \mathbf{e}_0\| \leq C_1 h^3 \sqrt{p}$. Since $C_1 \sqrt{p} \leq d_3$, the base case is proven.

Now suppose that for all $k = 0, 1, \ldots, n-1$ the claim

$$\|\mathbf{e}_k\| \leq d_1 e^{d_2 kh} h^2 \quad \text{and} \quad \|\mathbf{e}_{k+1} - \mathbf{e}_k\| \leq d_3 e^{d_2 kh} h^3$$

is proven. Then

$$\|\mathbf{e}_n\| \overset{(a)}{\leq} \|\mathbf{e}_{n-1}\| + \|\mathbf{e}_n - \mathbf{e}_{n-1}\| \leq d_1 e^{d_2(n-1)h} h^2 + d_3 e^{d_2(n-1)h} h^3$$

$$= d_1 e^{d_2(n-1)h} h^2 \left(1 + \frac{d_3}{d_1} h\right) \overset{(b)}{\leq} d_1 e^{d_2(n-1)h} h^2 (1 + d_2 h)$$

$$\overset{(c)}{\leq} d_1 e^{d_2(n-1)h} h^2 \cdot e^{d_2 h} = d_1 e^{d_2 nh} h^2,$$

where (a) is by the triangle inequality, (b) is by $d_3/d_1 \leq d_2$, in (c) we used $1 + x \leq e^x$ for all $x \geq 0$.

Next, combining Theorem SA-2.3 with (SA-2.1), we have

$$\left| [\mathbf{e}_{n+1} - \mathbf{e}_n]_j \right| \leq C_1 h^3 + h \left| \frac{\nabla_j E_n \left( \tilde{\boldsymbol{\theta}}(t_n) \right)}{\sqrt{A} + \varepsilon} - \frac{\nabla_j E_n \left( \boldsymbol{\theta}^{(n)} \right)}{\sqrt{B} + \varepsilon} \right|, \tag{SA-2.3}$$

where to simplify notation we put

$$A := \sum_{k=0}^{n} \rho^{n-k} (1-\rho) \left( \nabla_j E_k \left( \tilde{\boldsymbol{\theta}}(t_k) \right) \right)^2,$$

$$B := \sum_{k=0}^{n} \rho^{n-k} (1-\rho) \left( \nabla_j E_k \left( \boldsymbol{\theta}^{(k)} \right) \right)^2.$$

Using $A \geq R^2$, $B \geq R^2$, we have

$$\left| \frac{1}{\sqrt{A} + \varepsilon} - \frac{1}{\sqrt{B} + \varepsilon} \right| = \frac{|A - B|}{\left( \sqrt{A} + \varepsilon \right) \left( \sqrt{B} + \varepsilon \right) \left( \sqrt{A} + \sqrt{B} \right)} \leq \frac{|A - B|}{2R(R + \varepsilon)^2}. \tag{SA-2.4}$$

But since

$$\left| \left( \nabla_j E_k \left( \tilde{\boldsymbol{\theta}}(t_k) \right) \right)^2 - \left( \nabla_j E_k \left( \boldsymbol{\theta}^{(k)} \right) \right)^2 \right|$$

$$= \left| \nabla_j E_k \left( \tilde{\boldsymbol{\theta}}(t_k) \right) - \nabla_j E_k \left( \boldsymbol{\theta}^{(k)} \right) \right| \cdot \left| \nabla_j E_k \left( \tilde{\boldsymbol{\theta}}(t_k) \right) + \nabla_j E_k \left( \boldsymbol{\theta}^{(k)} \right) \right|$$

$$\leq 2M_1 \left| \nabla_j E_k \left( \tilde{\boldsymbol{\theta}}(t_k) \right) - \nabla_j E_k \left( \boldsymbol{\theta}^{(k)} \right) \right| \leq 2M_1 M_2 \sqrt{p} \left\| \tilde{\boldsymbol{\theta}}(t_k) - \boldsymbol{\theta}^{(k)} \right\|,$$

we have

$$|A - B| \leq 2M_1 M_2 \sqrt{p} \sum_{k=0}^{n} \rho^{n-k} (1-\rho) \left\| \tilde{\boldsymbol{\theta}}(t_k) - \boldsymbol{\theta}^{(k)} \right\|. \tag{SA-2.5}$$

Combining (SA-2.4) and (SA-2.5), we obtain

$$\left| \frac{\nabla_j E_n \left( \tilde{\boldsymbol{\theta}}(t_n) \right)}{\sqrt{A} + \varepsilon} - \frac{\nabla_j E_n \left( \boldsymbol{\theta}^{(n)} \right)}{\sqrt{B} + \varepsilon} \right|$$

$$\leq \left| \nabla_j E_n \left( \tilde{\boldsymbol{\theta}}(t_n) \right) \right| \cdot \left| \frac{1}{\sqrt{A} + \varepsilon} - \frac{1}{\sqrt{B} + \varepsilon} \right| + \frac{\left| \nabla_j E_n \left( \tilde{\boldsymbol{\theta}}(t_n) \right) - \nabla_j E_n \left( \boldsymbol{\theta}^{(n)} \right) \right|}{\sqrt{B} + \varepsilon}$$

$$\leq M_1 \cdot \frac{2M_1 M_2 \sqrt{p} \sum_{k=0}^{n} \rho^{n-k} (1-\rho) \left\| \tilde{\boldsymbol{\theta}}(t_k) - \boldsymbol{\theta}^{(k)} \right\|}{2R(R + \varepsilon)^2} + \frac{M_2 \sqrt{p} \left\| \tilde{\boldsymbol{\theta}}(t_n) - \boldsymbol{\theta}^{(n)} \right\|}{R + \varepsilon}$$

$$= \frac{M_1^2 M_2 \sqrt{p}}{R(R + \varepsilon)^2} \sum_{k=0}^{n} \rho^{n-k} (1-\rho) \left\| \tilde{\boldsymbol{\theta}}(t_k) - \boldsymbol{\theta}^{(k)} \right\| + \frac{M_2 \sqrt{p}}{R + \varepsilon} \left\| \tilde{\boldsymbol{\theta}}(t_n) - \boldsymbol{\theta}^{(n)} \right\|$$

$$\overset{(a)}{\leq} \frac{M_1^2 M_2 \sqrt{p}}{R(R + \varepsilon)^2} \sum_{k=0}^{n} \rho^{n-k} (1-\rho) d_1 e^{d_2 kh} h^2 + \frac{M_2 \sqrt{p}}{R + \varepsilon} d_1 e^{d_2 nh} h^2, \tag{SA-2.6}$$

where in (a) we used the induction hypothesis and that the bound on $\|\mathbf{e}_n\|$ is already proven.

Now note that since $0 < \rho e^{-d_2 h} \leq \rho$, we have $\sum_{k=0}^{n} \left( \rho e^{-d_2 h} \right)^k \leq \sum_{k=0}^{\infty} \rho^k = \frac{1}{1-\rho}$, which is rewritten as

$$\sum_{k=0}^{n} \rho^{n-k} (1-\rho) e^{d_2 kh} \leq e^{d_2 nh}.$$

Then we can continue (SA-2.6):

$$\left| \frac{\nabla_j E_n\left(\tilde{\boldsymbol{\theta}}(t_n)\right)}{\sqrt{A}+\varepsilon} - \frac{\nabla_j E_n\left(\boldsymbol{\theta}^{(n)}\right)}{\sqrt{B}+\varepsilon} \right| \leq \frac{M_2\sqrt{p}}{R+\varepsilon}\left( \frac{M_1^2}{R(R+\varepsilon)}+1 \right) d_1 e^{d_2 nh} h^2 \tag{SA-2.7}$$

Again using $1 \leq e^{d_2 nh}$, we conclude from (SA-2.3) and (SA-2.7) that

$$\|\mathbf{e}_{n+1} - \mathbf{e}_n\| \leq \underbrace{\left( C_1 + \frac{M_2\sqrt{p}}{R+\varepsilon}\left( \frac{M_1^2}{R(R+\varepsilon)}+1 \right) d_1 \right)}_{\leq d_3} \sqrt{p}\, e^{d_2 nh} h^3,$$

finishing the induction step. $\qquad\qquad\qquad\qquad\qquad\qquad\qquad\qquad\qquad\qquad\qquad\qquad\square$

**SA-2.5 RMSProp with $\varepsilon$ outside: full-batch.** In the full-batch setting $E_k \equiv E$, the terms in (SA-2.2) simplify to

$$R_j^{(n)}(\boldsymbol{\theta}) = \left| \nabla_j E(\boldsymbol{\theta}) \right| \sqrt{1-\rho^{n+1}},$$

$$P_j^{(n)}(\boldsymbol{\theta}) = \sum_{k=0}^{n} \rho^{n-k}(1-\rho)\nabla_j E(\boldsymbol{\theta}) \sum_{i=1}^{p} \nabla_{ij} E(\boldsymbol{\theta}) \sum_{l=k}^{n-1} \frac{\nabla_i E(\boldsymbol{\theta})}{\left| \nabla_i E(\boldsymbol{\theta}) \right| \sqrt{1-\rho^{l+1}} + \varepsilon},$$

$$\bar{P}_j^{(n)}(\boldsymbol{\theta}) = \left( 1-\rho^{n+1} \right) \nabla_j E(\boldsymbol{\theta}) \sum_{i=1}^{p} \nabla_{ij} E(\boldsymbol{\theta}) \frac{\nabla_i E(\boldsymbol{\theta})}{\left| \nabla_i E(\boldsymbol{\theta}) \right| \sqrt{1-\rho^{n+1}} + \varepsilon}.$$

If $\varepsilon$ is small and the iteration number $n$ is large, (SA-2.2) simplifies to

$$\dot{\tilde{\theta}}_j(t) = -\operatorname{sign}\nabla_j E(\tilde{\boldsymbol{\theta}}(t)) + h\frac{\rho}{1-\rho} \cdot \frac{\sum_{i=1}^{p} \nabla_{ij} E(\tilde{\boldsymbol{\theta}}(t))\operatorname{sign}\nabla_i E(\tilde{\boldsymbol{\theta}}(t))}{\left| \nabla_j E(\tilde{\boldsymbol{\theta}}(t)) \right|}$$

$$= \left| \nabla_j E(\tilde{\boldsymbol{\theta}}(t)) \right|^{-1} \left[ -\nabla_j E(\tilde{\boldsymbol{\theta}}(t)) + h\frac{\rho}{1-\rho}\nabla_j \left\| \nabla E(\tilde{\boldsymbol{\theta}}(t)) \right\|_1 \right].$$

## SA-3  RMSProp with $\varepsilon$ inside the square root

**Definition SA-3.1.** In this section, for some $\boldsymbol{\theta}^{(0)} \in \mathbb{R}^p$, $\nu^{(0)} = \mathbf{0} \in \mathbb{R}^p$, $\rho \in (0,1)$, let the sequence of $p$-vectors $\left\{ \boldsymbol{\theta}^{(k)} \right\}_{k\in\mathbb{Z}_{\geq 0}}$ be defined for $n \geq 0$ by

$$\nu_j^{(n+1)} = \rho\nu_j^{(n)} + (1-\rho)\left( \nabla_j E_n\left(\boldsymbol{\theta}^{(n)}\right) \right)^2,$$

$$\theta_j^{(n+1)} = \theta_j^{(n)} - \frac{h}{\sqrt{\nu_j^{(n+1)}+\varepsilon}}\nabla_j E_n\left(\boldsymbol{\theta}^{(n)}\right). \tag{SA-3.1}$$

Let $\tilde{\boldsymbol{\theta}}(t)$ be defined as a continuous solution to the piecewise ODE

$$\dot{\tilde{\theta}}_j(t) = -\frac{\nabla_j E_n\left(\tilde{\boldsymbol{\theta}}(t)\right)}{R_j^{(n)}\left(\tilde{\boldsymbol{\theta}}(t)\right)}$$

$$+ h\left( \frac{\nabla_j E_n\left(\tilde{\boldsymbol{\theta}}(t)\right)\left( 2P_j^{(n)}\left(\tilde{\boldsymbol{\theta}}(t)\right) + \bar{P}_j^{(n)}\left(\tilde{\boldsymbol{\theta}}(t)\right) \right)}{2R_j^{(n)}\left(\tilde{\boldsymbol{\theta}}(t)\right)^3} - \frac{\sum_{i=1}^{p} \nabla_{ij} E_n\left(\tilde{\boldsymbol{\theta}}(t)\right)\frac{\nabla_i E_n\left(\tilde{\boldsymbol{\theta}}(t)\right)}{R_i^{(n)}\left(\tilde{\boldsymbol{\theta}}(t)\right)}}{2R_j^{(n)}(\tilde{\boldsymbol{\theta}}(t))} \right). \tag{SA-3.2}$$

with the initial condition $\tilde{\boldsymbol{\theta}}(0) = \boldsymbol{\theta}^{(0)}$, where $\mathbf{R}^{(n)}(\boldsymbol{\theta})$, $\mathbf{P}^{(n)}(\boldsymbol{\theta})$ and $\bar{\mathbf{P}}^{(n)}(\boldsymbol{\theta})$ are $p$-dimensional functions with components

$$
R_j^{(n)}(\boldsymbol{\theta}) := \sqrt{\sum_{k=0}^{n} \rho^{n-k}(1-\rho)\big(\nabla_j E_k(\boldsymbol{\theta})\big)^2 + \varepsilon},
$$

$$
P_j^{(n)}(\boldsymbol{\theta}) := \sum_{k=0}^{n} \rho^{n-k}(1-\rho)\nabla_j E_k(\boldsymbol{\theta}) \sum_{i=1}^{p} \nabla_{ij} E_k(\boldsymbol{\theta}) \sum_{l=k}^{n-1} \frac{\nabla_i E_l(\boldsymbol{\theta})}{R_i^{(l)}(\boldsymbol{\theta})}, \tag{SA-3.3}
$$

$$
\bar{P}_j^{(n)}(\boldsymbol{\theta}) := \sum_{k=0}^{n} \rho^{n-k}(1-\rho)\nabla_j E_k(\boldsymbol{\theta}) \sum_{i=1}^{p} \nabla_{ij} E_k(\boldsymbol{\theta}) \frac{\nabla_i E_n(\boldsymbol{\theta})}{R_i^{(n)}(\boldsymbol{\theta})}.
$$

**Assumption SA-3.2.** For some positive constants $M_1$, $M_2$, $M_3$, $M_4$ we have

$$
\sup_i \sup_k \sup_{\boldsymbol{\theta}} \big|\nabla_i E_k(\boldsymbol{\theta})\big| \le M_1,
$$

$$
\sup_{i,j} \sup_k \sup_{\boldsymbol{\theta}} \big|\nabla_{ij} E_k(\boldsymbol{\theta})\big| \le M_2,
$$

$$
\sup_{i,j,s} \sup_k \sup_{\boldsymbol{\theta}} \big|\nabla_{ijs} E_k(\boldsymbol{\theta})\big| \le M_3,
$$

$$
\sup_{i,j,s,r} \sup_k \sup_{\boldsymbol{\theta}} \big|\nabla_{ijsr} E_k(\boldsymbol{\theta})\big| \le M_4.
$$

**Theorem SA-3.3** (RMSProp with $\varepsilon$ inside: local error bound). *Suppose Assumption SA-3.2 holds. Then for all $n \in \{0, 1, \ldots, \lfloor T/h \rfloor\}$*

$$
\left| \tilde{\theta}_j(t_{n+1}) - \tilde{\theta}_j(t_n) + h \frac{\nabla_j E_n\big(\tilde{\boldsymbol{\theta}}(t_n)\big)}{\sqrt{\sum_{k=0}^{n} \rho^{n-k}(1-\rho)\big(\nabla_j E_k\big(\tilde{\boldsymbol{\theta}}(t_k)\big)\big)^2 + \varepsilon}} \right| \le C_2 h^3
$$

*for a positive constant $C_2$ depending on $\rho$, where $\tilde{\boldsymbol{\theta}}(t)$ is defined in Definition SA-3.1.*

The argument is the same as for Theorem SA-2.3.

**Theorem SA-3.4** (RMSProp with $\varepsilon$ inside: global error bound). *Suppose Assumption SA-3.2 holds. Then there exist positive constants $d_4$, $d_5$, $d_6$ such that for all $n \in \{0, 1, \ldots, \lfloor T/h \rfloor\}$*

$$
\|\mathbf{e}_n\| \le d_4 e^{d_5 nh} h^2 \quad \text{and} \quad \|\mathbf{e}_{n+1} - \mathbf{e}_n\| \le d_6 e^{d_5 nh} h^3,
$$

*where $\mathbf{e}_n := \tilde{\boldsymbol{\theta}}(t_n) - \boldsymbol{\theta}^{(n)}$; $\tilde{\boldsymbol{\theta}}(t)$ and $\big\{\boldsymbol{\theta}^{(k)}\big\}_{k \in \mathbb{Z}_{\ge 0}}$ are defined in Definition SA-3.1. The constants can be defined as*

$$
d_4 := C_2,
$$

$$
d_5 := \left[1 + \frac{M_2\sqrt{p}}{\sqrt{\varepsilon}}\left(\frac{M_1^2}{\varepsilon} + 1\right)d_4\right]\sqrt{p},
$$

$$
d_6 := C_2 d_5.
$$

The argument is the same as for Theorem SA-2.4.

**SA-3.5 RMSProp with $\varepsilon$ inside: full-batch.** In the full-batch setting $E_k \equiv E$, the terms in (SA-3.2) simplify to

$$
R_j^{(n)}(\boldsymbol{\theta}) = \sqrt{\big|\nabla_j E(\boldsymbol{\theta})\big|^2(1 - \rho^{n+1}) + \varepsilon},
$$

$$P_j^{(n)}(\boldsymbol{\theta}) = \sum_{k=0}^{n} \rho^{n-k}(1-\rho)\nabla_j E(\boldsymbol{\theta}) \sum_{i=1}^{p} \nabla_{ij} E(\boldsymbol{\theta}) \sum_{l=k}^{n-1} \frac{\nabla_i E(\boldsymbol{\theta})}{\sqrt{|\nabla_i E(\boldsymbol{\theta})|^2(1-\rho^{l+1})+\varepsilon}},$$

$$\bar{P}_j^{(n)}(\boldsymbol{\theta}) = (1-\rho^{n+1})\nabla_j E(\boldsymbol{\theta}) \sum_{i=1}^{p} \nabla_{ij} E(\boldsymbol{\theta}) \frac{\nabla_i E(\boldsymbol{\theta})}{\sqrt{|\nabla_i E(\boldsymbol{\theta})|^2(1-\rho^{n+1})+\varepsilon}}.$$

If the iteration number $n$ is large, (SA-3.2) rapidly becomes

$$\dot{\tilde{\theta}}_j(t) = -\frac{1}{\sqrt{|\nabla_j E(\tilde{\boldsymbol{\theta}}(t))|^2+\varepsilon}}\left(\nabla_j E(\tilde{\boldsymbol{\theta}}(t)) + \text{bias}\right), \tag{SA-3.4}$$

where

$$\text{bias} := \frac{h}{2}\left\{-\frac{2\rho}{1-\rho} + \frac{1+\rho}{1-\rho}\cdot\frac{\varepsilon}{|\nabla_j E(\tilde{\boldsymbol{\theta}}(t))|^2+\varepsilon}\right\}\nabla_j\big\|\nabla E(\tilde{\boldsymbol{\theta}}(t))\big\|_{1,\varepsilon}. \tag{SA-3.5}$$

## SA-4 Adam with $\varepsilon$ outside the square root

**Definition SA-4.1.** In this section, for some $\boldsymbol{\theta}^{(0)} \in \mathbb{R}^p$, $\boldsymbol{\nu}^{(0)} = \mathbf{0} \in \mathbb{R}^p$, $\beta, \rho \in (0,1)$, let the sequence of $p$-vectors $\left\{\boldsymbol{\theta}^{(k)}\right\}_{k\in\mathbb{Z}_{\geq 0}}$ be defined for $n \geq 0$ by

$$\nu_j^{(n+1)} = \rho\nu_j^{(n)} + (1-\rho)\left(\nabla_j E_n\left(\boldsymbol{\theta}^{(n)}\right)\right)^2,$$

$$m_j^{(n+1)} = \beta m_j^{(n)} + (1-\beta)\nabla_j E_n\left(\boldsymbol{\theta}^{(n)}\right),$$

$$\theta_j^{(n+1)} = \theta_j^{(n)} - h\frac{m_j^{(n+1)}/(1-\beta^{n+1})}{\sqrt{\nu_j^{(n+1)}/(1-\rho^{n+1})}+\varepsilon}$$

or, rewriting,

$$\theta_j^{(n+1)} = \theta_j^{(n)} - h\frac{\frac{1}{1-\beta^{n+1}}\sum_{k=0}^{n}\beta^{n-k}(1-\beta)\nabla_j E_k\left(\boldsymbol{\theta}^{(k)}\right)}{\sqrt{\frac{1}{1-\rho^{n+1}}\sum_{k=0}^{n}\rho^{n-k}(1-\rho)\left(\nabla_j E_k\left(\boldsymbol{\theta}^{(k)}\right)\right)^2}+\varepsilon}. \tag{SA-4.1}$$

Let $\tilde{\boldsymbol{\theta}}(t)$ be defined as a continuous solution to the piecewise ODE

$$\dot{\tilde{\theta}}_j(t) = -\frac{M_j^{(n)}\left(\tilde{\boldsymbol{\theta}}(t)\right)}{R_j^{(n)}\left(\tilde{\boldsymbol{\theta}}(t)\right)+\varepsilon}$$

$$+ h\left(\frac{M_j^{(n)}\left(\tilde{\boldsymbol{\theta}}(t)\right)\left(2P_j^{(n)}\left(\tilde{\boldsymbol{\theta}}(t)\right)+\bar{P}_j^{(n)}\left(\tilde{\boldsymbol{\theta}}(t)\right)\right)}{2\left(R_j^{(n)}\left(\tilde{\boldsymbol{\theta}}(t)\right)+\varepsilon\right)^2 R_j^{(n)}\left(\tilde{\boldsymbol{\theta}}(t)\right)} - \frac{2L_j^{(n)}\left(\tilde{\boldsymbol{\theta}}(t)\right)+\bar{L}_j^{(n)}\left(\tilde{\boldsymbol{\theta}}(t)\right)}{2\left(R_j^{(n)}\left(\tilde{\boldsymbol{\theta}}(t)\right)+\varepsilon\right)}\right). \tag{SA-4.2}$$

with the initial condition $\tilde{\boldsymbol{\theta}}(0) = \boldsymbol{\theta}^{(0)}$, where $\mathbf{R}^{(n)}(\boldsymbol{\theta})$, $\mathbf{P}^{(n)}(\boldsymbol{\theta})$, $\bar{\mathbf{P}}^{(n)}(\boldsymbol{\theta})$, $\mathbf{M}^{(n)}(\boldsymbol{\theta})$, $\mathbf{L}^{(n)}(\boldsymbol{\theta})$, $\bar{\mathbf{L}}^{(n)}(\boldsymbol{\theta})$ are

$p$-dimensional functions with components

$$R_j^{(n)}(\boldsymbol{\theta}) := \sqrt{\sum_{k=0}^{n} \rho^{n-k}(1-\rho)\big(\nabla_j E_k(\boldsymbol{\theta})\big)^2 / (1-\rho^{n+1})},$$

$$M_j^{(n)}(\boldsymbol{\theta}) := \frac{1}{1-\beta^{n+1}} \sum_{k=0}^{n} \beta^{n-k}(1-\beta)\nabla_j E_k(\boldsymbol{\theta}),$$

$$L_j^{(n)}(\boldsymbol{\theta}) := \frac{1}{1-\beta^{n+1}} \sum_{k=0}^{n} \beta^{n-k}(1-\beta) \sum_{i=1}^{p} \nabla_{ij} E_k(\boldsymbol{\theta}) \sum_{l=k}^{n-1} \frac{M_i^{(l)}(\boldsymbol{\theta})}{R_i^{(l)}(\boldsymbol{\theta})+\varepsilon}, \tag{SA-4.3}$$

$$\bar{L}_j^{(n)}(\boldsymbol{\theta}) := \frac{1}{1-\beta^{n+1}} \sum_{k=0}^{n} \beta^{n-k}(1-\beta) \sum_{i=1}^{p} \nabla_{ij} E_k(\boldsymbol{\theta}) \frac{M_i^{(n)}(\boldsymbol{\theta})}{R_i^{(n)}(\boldsymbol{\theta})+\varepsilon},$$

$$P_j^{(n)}(\boldsymbol{\theta}) := \frac{1}{1-\rho^{n+1}} \sum_{k=0}^{n} \rho^{n-k}(1-\rho)\nabla_j E_k(\boldsymbol{\theta}) \sum_{i=1}^{p} \nabla_{ij} E_k(\boldsymbol{\theta}) \sum_{l=k}^{n-1} \frac{M_i^{(l)}(\boldsymbol{\theta})}{R_i^{(l)}(\boldsymbol{\theta})+\varepsilon},$$

$$\bar{P}_j^{(n)}(\boldsymbol{\theta}) := \frac{1}{1-\rho^{n+1}} \sum_{k=0}^{n} \rho^{n-k}(1-\rho)\nabla_j E_k(\boldsymbol{\theta}) \sum_{i=1}^{p} \nabla_{ij} E_k(\boldsymbol{\theta}) \frac{M_i^{(n)}(\boldsymbol{\theta})}{R_i^{(n)}(\boldsymbol{\theta})+\varepsilon}.$$

**Assumption SA-4.2.**

1. For some positive constants $M_1$, $M_2$, $M_3$, $M_4$ we have

$$\sup_{i} \sup_{k} \sup_{\boldsymbol{\theta}} \big|\nabla_i E_k(\boldsymbol{\theta})\big| \leq M_1,$$

$$\sup_{i,j} \sup_{k} \sup_{\boldsymbol{\theta}} \big|\nabla_{ij} E_k(\boldsymbol{\theta})\big| \leq M_2,$$

$$\sup_{i,j,s} \sup_{k} \sup_{\boldsymbol{\theta}} \big|\nabla_{ijs} E_k(\boldsymbol{\theta})\big| \leq M_3,$$

$$\sup_{i,j,s,r} \sup_{k} \sup_{\boldsymbol{\theta}} \big|\nabla_{ijsr} E_k(\boldsymbol{\theta})\big| \leq M_4.$$

2. For some $R > 0$ we have for all $n \in \big\{0, 1, \ldots, \lfloor T/h \rfloor\big\}$

$$R_j^{(n)}\big(\tilde{\boldsymbol{\theta}}(t_n)\big) \geq R, \quad \frac{1}{1-\rho^{n+1}} \sum_{k=0}^{n} \rho^{n-k}(1-\rho)\Big(\nabla_j E_k\big(\tilde{\boldsymbol{\theta}}(t_k)\big)\Big)^2 \geq R^2,$$

where $\tilde{\boldsymbol{\theta}}(t)$ is defined in [Definition SA-4.1](#).

**Theorem SA-4.3** (Adam with $\varepsilon$ outside: local error bound). *Suppose [Assumption SA-4.2](#) holds. Then for all $n \in \big\{0, 1, \ldots, \lfloor T/h \rfloor\big\}$*

$$\left| \tilde{\theta}_j(t_{n+1}) - \tilde{\theta}_j(t_n) + h \frac{\frac{1}{1-\beta^{n+1}} \sum_{k=0}^{n} \beta^{n-k}(1-\beta)\nabla_j E_k\big(\tilde{\boldsymbol{\theta}}(t_k)\big)}{\sqrt{\frac{1}{1-\rho^{n+1}} \sum_{k=0}^{n} \rho^{n-k}(1-\rho)\Big(\nabla_j E_k\big(\tilde{\boldsymbol{\theta}}(t_k)\big)\Big)^2} + \varepsilon} \right| \leq C_3 h^3$$

*for a positive constant $C_3$ depending on $\beta$ and $\rho$.*

The argument is the same as for [Theorem SA-2.3](#).

**Theorem SA-4.4** (Adam with $\varepsilon$ outside: global error bound). *Suppose [Assumption SA-4.2](#) holds, and*

$$\frac{1}{1-\rho^{n+1}} \sum_{k=0}^{n} \rho^{n-k}(1-\rho)\Big(\nabla_j E_k\big(\boldsymbol{\theta}^{(k)}\big)\Big)^2 \geq R^2$$

*for $\big\{\boldsymbol{\theta}^{(k)}\big\}_{k \in \mathbb{Z}_{\geq 0}}$ defined in [Definition SA-4.1](#). Then there exist positive constants $d_7$, $d_8$, $d_9$ such that for all $n \in \big\{0, 1, \ldots, \lfloor T/h \rfloor\big\}$*

$$\|\mathbf{e}_n\| \leq d_7 e^{d_8 nh} h^2 \quad and \quad \|\mathbf{e}_{n+1} - \mathbf{e}_n\| \leq d_9 e^{d_8 nh} h^3,$$

*where* $\mathbf{e}_n := \tilde{\boldsymbol{\theta}}(t_n) - \boldsymbol{\theta}^{(n)}$. *The constants can be defined as*

$$d_7 := C_3,$$

$$d_8 := \left[1 + \frac{M_2\sqrt{p}}{R + \varepsilon}\left(\frac{M_1^2}{R(R + \varepsilon)} + 1\right)d_7\right]\sqrt{p},$$

$$d_9 := C_3 d_8.$$

*Proof.* Analogously to Theorem SA-2.4, we will prove this by induction over $n$.

The base case is $n = 0$. Indeed, $\mathbf{e}_0 = \tilde{\boldsymbol{\theta}}(0) - \boldsymbol{\theta}^{(0)} = \mathbf{0}$. Then the $j$th component of $\mathbf{e}_1 - \mathbf{e}_0$ is

$$[\mathbf{e}_1 - \mathbf{e}_0]_j = [\mathbf{e}_1]_j = \tilde{\theta}_j(t_1) - \theta_j^{(0)} + \frac{h\nabla_j E_0\left(\boldsymbol{\theta}^{(0)}\right)}{\left|\nabla_j E_0\left(\boldsymbol{\theta}^{(0)}\right)\right| + \varepsilon}$$

$$= \tilde{\theta}_j(t_1) - \tilde{\theta}_j(t_0) + \frac{h\nabla_j E_0\left(\tilde{\boldsymbol{\theta}}(t_0)\right)}{\sqrt{\left(\nabla_j E_0\left(\tilde{\boldsymbol{\theta}}(t_0)\right)\right)^2 + \varepsilon}}.$$

By Theorem SA-4.3, the absolute value of the right-hand side does not exceed $C_3 h^3$, which means $\|\mathbf{e}_1 - \mathbf{e}_0\| \leq C_3 h^3 \sqrt{p}$. Since $C_3 \sqrt{p} \leq d_9$, the base case is proven.

Now suppose that for all $k = 0, 1, \ldots, n - 1$ the claim

$$\|\mathbf{e}_k\| \leq d_7 e^{d_8 kh} h^2 \quad \text{and} \quad \|\mathbf{e}_{k+1} - \mathbf{e}_k\| \leq d_9 e^{d_8 kh} h^3$$

is proven. Then

$$\|\mathbf{e}_n\| \overset{(a)}{\leq} \|\mathbf{e}_{n-1}\| + \|\mathbf{e}_n - \mathbf{e}_{n-1}\| \leq d_7 e^{d_8(n-1)h} h^2 + d_9 e^{d_8(n-1)h} h^3$$

$$= d_7 e^{d_8(n-1)h} h^2 \left(1 + \frac{d_9}{d_7} h\right) \overset{(b)}{\leq} d_7 e^{d_8(n-1)h} h^2 (1 + d_8 h)$$

$$\overset{(c)}{\leq} d_7 e^{d_8(n-1)h} h^2 \cdot e^{d_8 h} = d_7 e^{d_8 nh} h^2,$$

where (a) is by the triangle inequality, (b) is by $d_9/d_7 \leq d_8$, in (c) we used $1 + x \leq e^x$ for all $x \geq 0$.

Next, combining Theorem SA-4.3 with (SA-4.1), we have

$$\left|[\mathbf{e}_{n+1} - \mathbf{e}_n]_j\right| \leq C_3 h^3 + h\left|\frac{N'}{\sqrt{D'} + \varepsilon} - \frac{N''}{\sqrt{D''} + \varepsilon}\right|, \tag{SA-4.4}$$

where to simplify notation we put

$$N' := \frac{1}{1 - \beta^{n+1}} \sum_{k=0}^{n} \beta^{n-k}(1 - \beta)\nabla_j E_k\left(\boldsymbol{\theta}^{(k)}\right),$$

$$N'' := \frac{1}{1 - \beta^{n+1}} \sum_{k=0}^{n} \beta^{n-k}(1 - \beta)\nabla_j E_k\left(\tilde{\boldsymbol{\theta}}(t_k)\right),$$

$$D' := \frac{1}{1 - \rho^{n+1}} \sum_{k=0}^{n} \rho^{n-k}(1 - \rho)\left(\nabla_j E_k\left(\boldsymbol{\theta}^{(k)}\right)\right)^2,$$

$$D'' := \frac{1}{1 - \rho^{n+1}} \sum_{k=0}^{n} \rho^{n-k}(1 - \rho)\left(\nabla_j E_k\left(\tilde{\boldsymbol{\theta}}(t_k)\right)\right)^2.$$

Using $D' \geq R^2$, $D'' \geq R^2$, we have

$$\left|\frac{1}{\sqrt{D'} + \varepsilon} - \frac{1}{\sqrt{D''} + \varepsilon}\right| = \frac{|D' - D''|}{\left(\sqrt{D'} + \varepsilon\right)\left(\sqrt{D''} + \varepsilon\right)\left(\sqrt{D'} + \sqrt{D''}\right)} \leq \frac{|D' - D''|}{2R(R + \varepsilon)^2}. \tag{SA-4.5}$$

But since

$$\left| \left( \nabla_j E_k \left( \boldsymbol{\theta}^{(k)} \right) \right)^2 - \left( \nabla_j E_k \left( \tilde{\boldsymbol{\theta}}(t_k) \right) \right)^2 \right|$$

$$= \left| \nabla_j E_k \left( \boldsymbol{\theta}^{(k)} \right) - \nabla_j E_k \left( \tilde{\boldsymbol{\theta}}(t_k) \right) \right| \cdot \left| \nabla_j E_k \left( \boldsymbol{\theta}^{(k)} \right) + \nabla_j E_k \left( \tilde{\boldsymbol{\theta}}(t_k) \right) \right|$$

$$\leq 2M_1 \left| \nabla_j E_k \left( \boldsymbol{\theta}^{(k)} \right) - \nabla_j E_k \left( \tilde{\boldsymbol{\theta}}(t_k) \right) \right| \leq 2M_1 M_2 \sqrt{p} \left\| \boldsymbol{\theta}^{(k)} - \tilde{\boldsymbol{\theta}}(t_k) \right\|,$$

we have

$$\left| D' - D'' \right| \leq \frac{2M_1 M_2 \sqrt{p}}{1 - \rho^{n+1}} \sum_{k=0}^{n} \rho^{n-k} (1 - \rho) \left\| \boldsymbol{\theta}^{(k)} - \tilde{\boldsymbol{\theta}}(t_k) \right\|. \tag{SA-4.6}$$

Similarly,

$$\left| N' - N'' \right| \leq \frac{1}{1 - \beta^{n+1}} \sum_{k=0}^{n} \beta^{n-k} (1 - \beta) \left| \nabla_j E_k \left( \boldsymbol{\theta}^{(k)} \right) - \nabla_j E_k \left( \tilde{\boldsymbol{\theta}}(t_k) \right) \right|$$

$$\leq \frac{1}{1 - \beta^{n+1}} \sum_{k=0}^{n} \beta^{n-k} (1 - \beta) M_2 \sqrt{p} \left\| \boldsymbol{\theta}^{(k)} - \tilde{\boldsymbol{\theta}}(t_k) \right\|. \tag{SA-4.7}$$

Combining (SA-4.5), (SA-4.6) and (SA-4.7), we get

$$\left| \frac{N'}{\sqrt{D'} + \varepsilon} - \frac{N''}{\sqrt{D''} + \varepsilon} \right| \leq |N'| \cdot \left| \frac{1}{\sqrt{D'} + \varepsilon} - \frac{1}{\sqrt{D''} + \varepsilon} \right| + \frac{|N' - N''|}{\sqrt{D''} + \varepsilon}$$

$$\leq \frac{1}{1 - \beta^{n+1}} \sum_{k=0}^{n} \beta^{n-k} (1 - \beta) M_1 \cdot \frac{2M_1 M_2 \sqrt{p}}{2R(R + \varepsilon)^2 (1 - \rho^{n+1})} \sum_{k=0}^{n} \rho^{n-k} (1 - \rho) \left\| \boldsymbol{\theta}^{(k)} - \tilde{\boldsymbol{\theta}}(t_k) \right\|$$

$$+ \frac{M_2 \sqrt{p}}{(R + \varepsilon)(1 - \beta^{n+1})} \sum_{k=0}^{n} \beta^{n-k} (1 - \beta) \left\| \boldsymbol{\theta}^{(k)} - \tilde{\boldsymbol{\theta}}(t_k) \right\|$$

$$= \frac{M_1^2 M_2 \sqrt{p}}{R(R + \varepsilon)^2 (1 - \rho^{n+1})} \sum_{k=0}^{n} \rho^{n-k} (1 - \rho) \left\| \boldsymbol{\theta}^{(k)} - \tilde{\boldsymbol{\theta}}(t_k) \right\|$$

$$+ \frac{M_2 \sqrt{p}}{(R + \varepsilon)(1 - \beta^{n+1})} \sum_{k=0}^{n} \beta^{n-k} (1 - \beta) \left\| \boldsymbol{\theta}^{(k)} - \tilde{\boldsymbol{\theta}}(t_k) \right\|$$

$$\overset{(a)}{\leq} \frac{M_1^2 M_2 \sqrt{p}}{R(R + \varepsilon)^2 (1 - \rho^{n+1})} \sum_{k=0}^{n} \rho^{n-k} (1 - \rho) d_7 e^{d_8 k h} h^2$$

$$+ \frac{M_2 \sqrt{p}}{(R + \varepsilon)(1 - \beta^{n+1})} \sum_{k=0}^{n} \beta^{n-k} (1 - \beta) d_7 e^{d_8 k h} h^2, \tag{SA-4.8}$$

where in (a) we used the induction hypothesis and that the bound on $\|\mathbf{e}_n\|$ is already proven.

Now note that since $0 < \rho e^{-d_8 h} < \rho$, we have $\sum_{k=0}^{n} \left( \rho e^{-d_8 h} \right)^k \leq \sum_{k=0}^{n} \rho^k = \left( 1 - \rho^{n+1} \right)/(1 - \rho)$, which is rewritten as

$$\frac{1}{1 - \rho^{n+1}} \sum_{k=0}^{n} \rho^{n-k} (1 - \rho) e^{d_8 k h} \leq e^{d_8 n h}.$$

By the same logic,

$$\frac{1}{1 - \beta^{n+1}} \sum_{k=0}^{n} \beta^{n-k} (1 - \beta) e^{d_8 k h} \leq e^{d_8 n h}.$$

Then we can continue (SA-4.8):

$$\left| \frac{N'}{\sqrt{D'} + \varepsilon} - \frac{N''}{\sqrt{D''} + \varepsilon} \right| \leq \frac{M_2 \sqrt{p}}{R + \varepsilon} \left( \frac{M_1^2}{R(R + \varepsilon)} + 1 \right) d_7 e^{d_8 n h} h^2 \tag{SA-4.9}$$

Again using $1 \leq e^{d_8 nh}$, we conclude from (SA-4.4) and (SA-4.9) that

$$\|\mathbf{e}_{n+1} - \mathbf{e}_n\| \leq \underbrace{\left( C_3 + \frac{M_2\sqrt{p}}{R+\varepsilon} \left( \frac{M_1^2}{R(R+\varepsilon)} + 1 \right) d_7 \right) \sqrt{p}}_{\leq d_9} \, e^{d_8 nh} h^3,$$

finishing the induction step. □

## SA-5   Adam with $\varepsilon$ inside the square root

**Definition SA-5.1.** In this section, for some $\boldsymbol{\theta}^{(0)} \in \mathbb{R}^p$, $\boldsymbol{\nu}^{(0)} = \mathbf{0} \in \mathbb{R}^p$, $\beta, \rho \in (0,1)$, let the sequence of $p$-vectors $\left\{ \boldsymbol{\theta}^{(k)} \right\}_{k \in \mathbb{Z}_{\geq 0}}$ be defined for $n \geq 0$ by

$$\begin{aligned}
\nu_j^{(n+1)} &= \rho \nu_j^{(n)} + (1-\rho)\left( \nabla_j E_n\left(\boldsymbol{\theta}^{(n)}\right) \right)^2, \\
m_j^{(n+1)} &= \beta m_j^{(n)} + (1-\beta)\nabla_j E_n\left(\boldsymbol{\theta}^{(n)}\right), \\
\theta_j^{(n+1)} &= \theta_j^{(n)} - h\frac{m_j^{(n+1)}/\left(1-\beta^{n+1}\right)}{\sqrt{\nu_j^{(n+1)}/(1-\rho^{n+1}) + \varepsilon}}.
\end{aligned} \tag{SA-5.1}$$

Let $\tilde{\boldsymbol{\theta}}(t)$ be defined as a continuous solution to the piecewise ODE

$$\begin{aligned}
\dot{\tilde{\theta}}_j(t) = &-\frac{M_j^{(n)}\left(\tilde{\boldsymbol{\theta}}(t)\right)}{R_j^{(n)}\left(\tilde{\boldsymbol{\theta}}(t)\right)} \\
&+ h\left( \frac{M_j^{(n)}\left(\tilde{\boldsymbol{\theta}}(t)\right)\left(2P_j^{(n)}\left(\tilde{\boldsymbol{\theta}}(t)\right) + \bar{P}_j^{(n)}\left(\tilde{\boldsymbol{\theta}}(t)\right)\right)}{2R_j^{(n)}\left(\tilde{\boldsymbol{\theta}}(t)\right)^3} - \frac{2L_j^{(n)}\left(\tilde{\boldsymbol{\theta}}(t)\right) + \bar{L}_j^{(n)}\left(\tilde{\boldsymbol{\theta}}(t)\right)}{2R_j^{(n)}\left(\tilde{\boldsymbol{\theta}}(t)\right)} \right).
\end{aligned} \tag{SA-5.2}$$

with the initial condition $\tilde{\boldsymbol{\theta}}(0) = \boldsymbol{\theta}^{(0)}$, where $\mathbf{R}^{(n)}(\boldsymbol{\theta})$, $\mathbf{P}^{(n)}(\boldsymbol{\theta})$, $\bar{\mathbf{P}}^{(n)}(\boldsymbol{\theta})$, $\mathbf{M}^{(n)}(\boldsymbol{\theta})$, $\mathbf{L}^{(n)}(\boldsymbol{\theta})$, $\bar{\mathbf{L}}^{(n)}(\boldsymbol{\theta})$ are $p$-dimensional functions with components

$$\begin{aligned}
R_j^{(n)}(\boldsymbol{\theta}) &:= \sqrt{\sum_{k=0}^{n} \rho^{n-k}(1-\rho)\left(\nabla_j E_k(\boldsymbol{\theta})\right)^2/(1-\rho^{n+1}) + \varepsilon}, \\
M_j^{(n)}(\boldsymbol{\theta}) &:= \frac{1}{1-\beta^{n+1}} \sum_{k=0}^{n} \beta^{n-k}(1-\beta)\nabla_j E_k(\boldsymbol{\theta}), \\
L_j^{(n)}(\boldsymbol{\theta}) &:= \frac{1}{1-\beta^{n+1}} \sum_{k=0}^{n} \beta^{n-k}(1-\beta) \sum_{i=1}^{p} \nabla_{ij} E_k(\boldsymbol{\theta}) \sum_{l=k}^{n-1} \frac{M_i^{(l)}(\boldsymbol{\theta})}{R_i^{(l)}(\boldsymbol{\theta})}, \\
\bar{L}_j^{(n)}(\boldsymbol{\theta}) &:= \frac{1}{1-\beta^{n+1}} \sum_{k=0}^{n} \beta^{n-k}(1-\beta) \sum_{i=1}^{p} \nabla_{ij} E_k(\boldsymbol{\theta}) \frac{M_i^{(n)}(\boldsymbol{\theta})}{R_i^{(n)}(\boldsymbol{\theta})}, \\
P_j^{(n)}(\boldsymbol{\theta}) &:= \frac{1}{1-\rho^{n+1}} \sum_{k=0}^{n} \rho^{n-k}(1-\rho)\nabla_j E_k(\boldsymbol{\theta}) \sum_{i=1}^{p} \nabla_{ij} E_k(\boldsymbol{\theta}) \sum_{l=k}^{n-1} \frac{M_i^{(l)}(\boldsymbol{\theta})}{R_i^{(l)}(\boldsymbol{\theta})}, \\
\bar{P}_j^{(n)}(\boldsymbol{\theta}) &:= \frac{1}{1-\rho^{n+1}} \sum_{k=0}^{n} \rho^{n-k}(1-\rho)\nabla_j E_k(\boldsymbol{\theta}) \sum_{i=1}^{p} \nabla_{ij} E_k(\boldsymbol{\theta}) \frac{M_i^{(n)}(\boldsymbol{\theta})}{R_i^{(n)}(\boldsymbol{\theta})}.
\end{aligned} \tag{SA-5.3}$$

**Assumption SA-5.2.** For some positive constants $M_1$, $M_2$, $M_3$, $M_4$ we have

$$\sup_i \sup_k \sup_{\boldsymbol{\theta}} \left| \nabla_i E_k(\boldsymbol{\theta}) \right| \leq M_1,$$

$$\sup_{i,j}\sup_k \sup_{\boldsymbol{\theta}}\left|\nabla_{ij}E_k(\boldsymbol{\theta})\right| \le M_2,$$

$$\sup_{i,j,s}\sup_k \sup_{\boldsymbol{\theta}}\left|\nabla_{ijs}E_k(\boldsymbol{\theta})\right| \le M_3,$$

$$\sup_{i,j,s,r}\sup_k \sup_{\boldsymbol{\theta}}\left|\nabla_{ijsr}E_k(\boldsymbol{\theta})\right| \le M_4.$$

**Theorem SA-5.3** (Adam with $\varepsilon$ inside: local error bound). *Suppose Assumption SA-5.2 holds. Then for all $n \in \{0,1,\ldots,\lfloor T/h \rfloor\}$*

$$\left| \tilde{\theta}_j(t_{n+1}) - \tilde{\theta}_j(t_n) + h \frac{\frac{1}{1-\beta^{n+1}}\sum_{k=0}^n \beta^{n-k}(1-\beta)\nabla_j E_k\left(\tilde{\boldsymbol{\theta}}(t_k)\right)}{\sqrt{\frac{1}{1-\rho^{n+1}}\sum_{k=0}^n \rho^{n-k}(1-\rho)\left(\nabla_j E_k\left(\tilde{\boldsymbol{\theta}}(t_k)\right)\right)^2 + \varepsilon}} \right| \le C_4 h^3$$

*for a positive constant $C_4$ depending on $\beta$ and $\rho$.*

The argument is the same as for Theorem SA-2.3.

**Theorem SA-5.4** (Adam with $\varepsilon$ inside: global error bound). *Suppose Assumption SA-5.2 holds for $\left\{\boldsymbol{\theta}^{(k)}\right\}_{k\in\mathbb{Z}_{\ge 0}}$ defined in Definition SA-5.1. Then there exist positive constants $d_{10}$, $d_{11}$, $d_{12}$ such that for all $n \in \{0,1,\ldots,\lfloor T/h \rfloor\}$*

$$\|\mathbf{e}_n\| \le d_{10}e^{d_{11}nh}h^2 \quad and \quad \|\mathbf{e}_{n+1} - \mathbf{e}_n\| \le d_{12}e^{d_{11}nh}h^3,$$

*where $\mathbf{e}_n := \tilde{\boldsymbol{\theta}}(t_n) - \boldsymbol{\theta}^{(n)}$. The constants can be defined as*

$$d_{10} := C_4,$$

$$d_{11} := \left[1 + \frac{M_2\sqrt{p}}{\sqrt{\varepsilon}}\left(\frac{M_1^2}{\varepsilon}+1\right)d_{10}\right]\sqrt{p},$$

$$d_{12} := C_4 d_{11}.$$

The argument is the same as for Theorem SA-4.4.

## SA-6  Technical bounding lemmas

We will need the following lemmas to prove Theorem SA-2.3.

**Lemma SA-6.1.** *Suppose Assumption SA-2.2 holds. Then*

$$\sup_{\boldsymbol{\theta}}\left|P_j^{(n)}(\boldsymbol{\theta})\right| \le C_5, \tag{SA-6.1}$$

$$\sup_{\boldsymbol{\theta}}\left|\bar{P}_j^{(n)}(\boldsymbol{\theta})\right| \le C_6, \tag{SA-6.2}$$

*with constants $C_5$, $C_6$ defined as follows:*

$$C_5 := p\frac{M_1^2 M_2}{R+\varepsilon}\cdot\frac{\rho}{1-\rho},$$

$$C_6 := p\frac{M_1^2 M_2}{R+\varepsilon}.$$

*Proof of Lemma SA-6.1.* The proof is done in the following simple steps.

**SA-6.2 Proof of** (SA-6.1). This bound is straightforward:

$$
\sup_{\boldsymbol{\theta}}\left|P_j^{(n)}(\boldsymbol{\theta})\right| = \sup_{\boldsymbol{\theta}}\left|\sum_{k=0}^{n}\rho^{n-k}(1-\rho)\nabla_j E_k(\boldsymbol{\theta})\sum_{i=1}^{p}\nabla_{ij}E_k(\boldsymbol{\theta})\sum_{l=k}^{n-1}\frac{\nabla_i E_l(\boldsymbol{\theta})}{R_i^{(l)}(\boldsymbol{\theta})+\varepsilon}\right|
$$

$$
\leq p\frac{M_1^2 M_2}{R+\varepsilon}(1-\rho)\sum_{k=0}^{n}\rho^{n-k}(n-k) \leq p\frac{M_1^2 M_2}{R+\varepsilon}(1-\rho)\sum_{k=0}^{\infty}\rho^k k = C_5.
$$

**SA-6.3 Proof of** (SA-6.2). This bound is straightforward:

$$
\sup_{\boldsymbol{\theta}}\left|\bar{P}_j^{(n)}(\boldsymbol{\theta})\right| = \sup_{\boldsymbol{\theta}}\left|\sum_{k=0}^{n}\rho^{n-k}(1-\rho)\nabla_j E_k(\boldsymbol{\theta})\sum_{i=1}^{p}\nabla_{ij}E_k(\boldsymbol{\theta})\frac{\nabla_i E_n(\boldsymbol{\theta})}{R_i^{(n)}(\boldsymbol{\theta})+\varepsilon}\right|
$$

$$
\leq p\frac{M_1^2 M_2}{R+\varepsilon}(1-\rho)\sum_{k=0}^{n}\rho^{n-k} \leq p\frac{M_1^2 M_2}{R+\varepsilon} = C_6.
$$

This concludes the proof of Lemma SA-6.1. $\qquad\square$

**Lemma SA-6.4.** *Suppose Assumption SA-2.2 holds. Then the first derivative of $t \mapsto \tilde{\theta}_j(t)$ is uniformly over $j$ and $t \in [0,T]$ bounded in absolute value by some positive constant, say $D_1$.*

*Proof.* This follows immediately from $h \leq T$, (SA-6.1), (SA-6.2) and the definition of $\tilde{\boldsymbol{\theta}}(t)$ given in (SA-2.2). $\qquad\square$

**Lemma SA-6.5.** *Suppose Assumption SA-2.2 holds. Then*

$$
\sup_{t\in[0,T]}\sup_{j}\left|\left(\nabla_j E_n\left(\tilde{\boldsymbol{\theta}}(t)\right)\right)^{\!\cdot}\right| \leq C_7, \tag{SA-6.3}
$$

$$
\sup_{n,k}\sup_{t\in[t_n,t_{n+1}]}\left|\sum_{i=1}^{p}\nabla_{ij}E_k\left(\tilde{\boldsymbol{\theta}}(t)\right)\left[\dot{\tilde{\theta}}_i(t)+\frac{\nabla_i E_n\left(\tilde{\boldsymbol{\theta}}(t)\right)}{R_i^{(n)}\left(\tilde{\boldsymbol{\theta}}(t)\right)+\varepsilon}\right]\right| \leq C_8 h, \tag{SA-6.4}
$$

$$
\sup_{k\leq n}\sup_{t\in[0,T]}\left|\sum_{i=1}^{p}\nabla_{ij}E_k\left(\tilde{\boldsymbol{\theta}}(t)\right)\sum_{l=k}^{n-1}\frac{\nabla_i E_l\left(\tilde{\boldsymbol{\theta}}(t)\right)}{R_i^{(l)}\left(\tilde{\boldsymbol{\theta}}(t)\right)+\varepsilon}\right| \leq (n-k)C_9, \tag{SA-6.5}
$$

$$
\left|\left(P_j^{(n)}\left(\tilde{\boldsymbol{\theta}}(t)\right)\right)^{\!\cdot}\right| \leq C_{10}+C_{14}, \tag{SA-6.6}
$$

$$
\left|\left(\bar{P}_j^{(n)}(\tilde{\boldsymbol{\theta}}(t))\right)^{\!\cdot}\right| \leq C_{15}, \tag{SA-6.7}
$$

$$
\left|\left(\sum_{i=1}^{p}\nabla_{ij}E_k\left(\tilde{\boldsymbol{\theta}}(t)\right)\frac{\nabla_i E_n\left(\tilde{\boldsymbol{\theta}}(t)\right)}{R_i^{(n)}\left(\tilde{\boldsymbol{\theta}}(t)\right)+\varepsilon}\right)^{\!\cdot}\right| \leq C_{13}, \tag{SA-6.8}
$$

$$
\left|\left(\frac{\nabla_j E_n\left(\tilde{\boldsymbol{\theta}}(t)\right)\left(2P_j^{(n)}\left(\tilde{\boldsymbol{\theta}}(t)\right)+\bar{P}_j^{(n)}\left(\tilde{\boldsymbol{\theta}}(t)\right)\right)}{2\left(R_j^{(n)}\left(\tilde{\boldsymbol{\theta}}(t)\right)+\varepsilon\right)^2 R_j^{(n)}\left(\tilde{\boldsymbol{\theta}}(t)\right)}\right)^{\!\cdot}\right| \leq C_{17}, \tag{SA-6.9}
$$

$$
\left|\left(\frac{\sum_{i=1}^{p}\nabla_{ij}E_n\left(\tilde{\boldsymbol{\theta}}(t)\right)\frac{\nabla_i E_n\left(\tilde{\boldsymbol{\theta}}(t)\right)}{R_i^{(n)}\left(\tilde{\boldsymbol{\theta}}(t)\right)+\varepsilon}}{2\left(R_j^{(n)}(\tilde{\boldsymbol{\theta}}(t))+\varepsilon\right)}\right)^{\!\cdot}\right| \leq C_{18}, \tag{SA-6.10}
$$

*with constants $C_7$, $C_8$, $C_9$, $C_{10}$, $C_{11}$, $C_{12}$, $C_{13}$, $C_{14}$, $C_{15}$, $C_{16}$, $C_{17}$, $C_{18}$ defined as follows:*

$$C_7 := pM_2 D_1,$$

$$C_8 := pM_2 \left[ \frac{M_1(2C_5 + C_6)}{2(R+\varepsilon)^2 R} + \frac{pM_1 M_2}{2(R+\varepsilon)^2} \right],$$

$$C_9 := p\frac{M_1 M_2}{R+\varepsilon},$$

$$C_{10} := D_1 p^2 \frac{M_1 M_2^2}{R+\varepsilon} \cdot \frac{\rho}{1-\rho},$$

$$C_{11} := \frac{D_1 p M_1 M_2}{R},$$

$$C_{12} := D_1 p^2 \frac{M_1 M_3}{R+\varepsilon},$$

$$C_{13} := C_{12} + pM_2 \left( \frac{D_1 p M_2}{R+\varepsilon} + \frac{M_1}{(R+\varepsilon)^2} C_{11} \right)$$

$$= \frac{D_1 p^2}{R+\varepsilon} \left( M_1 M_3 + M_2^2 + \frac{M_1^2 M_2^2}{(R+\varepsilon)R} \right),$$

$$C_{14} := M_1 C_{13} \frac{\rho}{1-\rho},$$

$$C_{15} := \frac{D_1 p^2 M_1 M_2^2}{R+\varepsilon} + \frac{D_1 p^2 M_1^2 M_3}{R+\varepsilon} + \frac{D_1 p^2 M_1 M_2^2}{R+\varepsilon} + \frac{p M_1^2 M_2 C_{11}}{(R+\varepsilon)^2},$$

$$C_{16} := \frac{2C_{11}}{R(R+\varepsilon)^3} + \frac{C_{11}}{(R+\varepsilon)^4},$$

$$C_{17} := \frac{D_1 p M_2 \cdot (2C_5 + C_6)}{2(R+\varepsilon)^2 R} + \frac{M_1 \big( 2(C_{10} + C_{14}) + C_{15} \big)}{2(R+\varepsilon)^2 R} + \frac{M_1(2C_5 + C_6)C_{16}}{2},$$

$$C_{18} := \frac{1}{2(R+\varepsilon)} \left( \frac{p^2 D_1 M_1 M_3}{R+\varepsilon} + \frac{p^2 D_1 M_2^2}{R+\varepsilon} + \frac{p M_1 M_2 C_{11}}{(R+\varepsilon)^2} \right) + \frac{1}{2} \cdot \frac{p M_1 M_2}{R+\varepsilon} \cdot \frac{C_{11}}{(R+\varepsilon)^2}.$$

*Proof of Lemma SA-6.5.* We divide this argument in several steps.

**SA-6.6 Proof of** (SA-6.3). This bound is straightforward:

$$\left| \left( \nabla_j E_n\big(\tilde{\boldsymbol{\theta}}(t)\big) \right)^{\cdot} \right| = \left| \sum_{i=1}^{p} \nabla_{ij} E_n\big(\tilde{\boldsymbol{\theta}}(t)\big) \dot{\tilde{\theta}}_i(t) \right| \leq C_7.$$

**SA-6.7 Proof of** (SA-6.4). By (SA-2.2) we have for $t = t_{n+1}^-$

$$\left| \dot{\tilde{\theta}}_j(t) + \frac{\nabla_j E_n\big(\tilde{\boldsymbol{\theta}}(t)\big)}{R_j^{(n)}\big(\tilde{\boldsymbol{\theta}}(t)\big) + \varepsilon} \right| \leq h\left[ \frac{M_1(2C_5 + C_6)}{2(R+\varepsilon)^2 R} + \frac{pM_1 M_2}{2(R+\varepsilon)^2} \right],$$

giving (SA-6.4) immediately.

**SA-6.8 Proof of** (SA-6.5). This bound follows from the assumptions immediately.

**SA-6.9 Proof of** (SA-6.6). We will prove this by bounding the two terms in the expression

$$\frac{\mathrm{d}}{\mathrm{d}t}P_j^{(n)}\left(\tilde{\boldsymbol{\theta}}(t)\right)$$

$$= \sum_{k=0}^{n}\rho^{n-k}(1-\rho)\sum_{u=1}^{p}\nabla_{ju}E_k\left(\tilde{\boldsymbol{\theta}}(t)\right)\dot{\tilde{\theta}}_u(t)\sum_{i=1}^{p}\nabla_{ij}E_k\left(\tilde{\boldsymbol{\theta}}(t)\right)\sum_{l=k}^{n-1}\frac{\nabla_i E_l\left(\tilde{\boldsymbol{\theta}}(t)\right)}{R_i^{(l)}\left(\tilde{\boldsymbol{\theta}}(t)\right)+\varepsilon} \qquad \text{(SA-6.11)}$$

$$+ \sum_{k=0}^{n}\rho^{n-k}(1-\rho)\nabla_j E_k\left(\tilde{\boldsymbol{\theta}}(t)\right)\sum_{i=1}^{p}\frac{\mathrm{d}}{\mathrm{d}t}\left\{\nabla_{ij}E_k\left(\tilde{\boldsymbol{\theta}}(t)\right)\sum_{l=k}^{n-1}\frac{\nabla_i E_l\left(\tilde{\boldsymbol{\theta}}(t)\right)}{R_i^{(l)}\left(\tilde{\boldsymbol{\theta}}(t)\right)+\varepsilon}\right\}.$$

It is easily shown that the first term in (SA-6.11) is bounded in absolute value by $C_{10}$:

$$\left|\sum_{k=0}^{n}\rho^{n-k}(1-\rho)\sum_{u=1}^{p}\nabla_{ju}E_k\left(\tilde{\boldsymbol{\theta}}(t)\right)\dot{\tilde{\theta}}_u(t)\sum_{i=1}^{p}\nabla_{ij}E_k\left(\tilde{\boldsymbol{\theta}}(t)\right)\sum_{l=k}^{n-1}\frac{\nabla_i E_l\left(\tilde{\boldsymbol{\theta}}(t)\right)}{R_i^{(l)}\left(\tilde{\boldsymbol{\theta}}(t)\right)+\varepsilon}\right|$$

$$\leq D_1 p^2 \frac{M_1 M_2^2}{R+\varepsilon}(1-\rho)\sum_{k=0}^{n}\rho^k k$$

$$\leq D_1 p^2 \frac{M_1 M_2^2}{R+\varepsilon}(1-\rho)\sum_{k=0}^{\infty}\rho^k k$$

$$= C_{10}.$$

For the proof of (SA-6.6), it is left to show that the second term in (SA-6.11) is bounded in absolute value by $C_{14}$.

To bound $\sum_{i=1}^{p}\frac{\mathrm{d}}{\mathrm{d}t}\left\{\nabla_{ij}E_k\left(\tilde{\boldsymbol{\theta}}(t)\right)\sum_{l=k}^{n-1}\frac{\nabla_i E_l\left(\tilde{\boldsymbol{\theta}}(t)\right)}{R_i^{(l)}\left(\tilde{\boldsymbol{\theta}}(t)\right)+\varepsilon}\right\}$, we can use

$$\left|\sum_{i=1}^{p}\frac{\mathrm{d}}{\mathrm{d}t}\left\{\nabla_{ij}E_k\left(\tilde{\boldsymbol{\theta}}(t)\right)\sum_{l=k}^{n-1}\frac{\nabla_i E_l\left(\tilde{\boldsymbol{\theta}}(t)\right)}{R_i^{(l)}\left(\tilde{\boldsymbol{\theta}}(t)\right)+\varepsilon}\right\}\right|$$

$$\leq \left|\sum_{i=1}^{p}\frac{\mathrm{d}}{\mathrm{d}t}\left\{\nabla_{ij}E_k\left(\tilde{\boldsymbol{\theta}}(t)\right)\right\}\sum_{l=k}^{n-1}\frac{\nabla_i E_l\left(\tilde{\boldsymbol{\theta}}(t)\right)}{R_i^{(l)}\left(\tilde{\boldsymbol{\theta}}(t)\right)+\varepsilon}\right|$$

$$+ \left|\sum_{i=1}^{p}\nabla_{ij}E_k\left(\tilde{\boldsymbol{\theta}}(t)\right)\sum_{l=k}^{n-1}\frac{\mathrm{d}}{\mathrm{d}t}\left\{\frac{\nabla_i E_l\left(\tilde{\boldsymbol{\theta}}(t)\right)}{R_i^{(l)}\left(\tilde{\boldsymbol{\theta}}(t)\right)+\varepsilon}\right\}\right|$$

By the Cauchy-Schwarz inequality applied twice,

$$\left|\sum_{i=1}^{p}\frac{\mathrm{d}}{\mathrm{d}t}\left\{\nabla_{ij}E_k\left(\tilde{\boldsymbol{\theta}}(t)\right)\right\}\sum_{l=k}^{n-1}\frac{\nabla_i E_l\left(\tilde{\boldsymbol{\theta}}(t)\right)}{R_i^{(l)}\left(\tilde{\boldsymbol{\theta}}(t)\right)+\varepsilon}\right|$$

$$\leq \sqrt{\sum_{i=1}^{p}\sum_{s=1}^{p}\left(\nabla_{ijs}E_k\left(\tilde{\boldsymbol{\theta}}(t)\right)\right)^2}\sqrt{\sum_{u=1}^{p}\dot{\tilde{\theta}}_u(t)^2}\sqrt{\sum_{i=1}^{p}\left|\sum_{l=k}^{n-1}\frac{\nabla_i E_l\left(\tilde{\boldsymbol{\theta}}(t)\right)}{R_i^{(l)}\left(\tilde{\boldsymbol{\theta}}(t)\right)+\varepsilon}\right|^2}$$

$$\leq M_3 p \cdot D_1 \sqrt{p}\cdot\sqrt{\sum_{i=1}^{p}\left|\sum_{l=k}^{n-1}\frac{\nabla_i E_l\left(\tilde{\boldsymbol{\theta}}(t)\right)}{R_i^{(l)}\left(\tilde{\boldsymbol{\theta}}(t)\right)+\varepsilon}\right|^2}\leq (n-k)C_{12}.$$

Next, for any $n$ and $j$

$$\left| \frac{\mathrm{d}}{\mathrm{d}t} R_j^{(n)}\big(\tilde{\boldsymbol{\theta}}(t)\big) \right| = \frac{1}{R_j^{(n)}\big(\tilde{\boldsymbol{\theta}}(t)\big)} \left| \sum_{k=0}^{n} \rho^{n-k}(1-\rho)\nabla_j E_k\big(\tilde{\boldsymbol{\theta}}(t)\big) \sum_{i=1}^{p} \nabla_{ij} E_k\big(\tilde{\boldsymbol{\theta}}(t)\big) \dot{\tilde{\theta}}_i(t) \right|$$

$$\leq \frac{1}{R_j^{(n)}\big(\tilde{\boldsymbol{\theta}}(t)\big)} D_1 p M_1 M_2 \sum_{k=0}^{n} \rho^{n-k}(1-\rho) \leq C_{11}.$$

(SA-6.12)

This gives

$$\left| \frac{\mathrm{d}}{\mathrm{d}t}\left\{ \frac{\nabla_i E_l\big(\tilde{\boldsymbol{\theta}}(t)\big)}{R_i^{(l)}\big(\tilde{\boldsymbol{\theta}}(t)\big) + \varepsilon} \right\} \right| \leq \frac{\left| \sum_{s=1}^{p} \nabla_{is} E_l\big(\tilde{\boldsymbol{\theta}}(t)\big) \dot{\tilde{\theta}}_s(t) \right|}{R_i^{(l)}\big(\tilde{\boldsymbol{\theta}}(t)\big) + \varepsilon} + \frac{\left| \nabla_i E_l\big(\tilde{\boldsymbol{\theta}}(t)\big) \right| \cdot \left| \frac{\mathrm{d}}{\mathrm{d}t} R_i^{(l)}\big(\tilde{\boldsymbol{\theta}}(t)\big) \right|}{\left( R_i^{(l)}\big(\tilde{\boldsymbol{\theta}}(t)\big) + \varepsilon \right)^2}$$

$$\leq \frac{D_1 p M_2}{R + \varepsilon} + \frac{M_1}{(R + \varepsilon)^2} C_{11}.$$

We have obtained

$$\left| \sum_{i=1}^{p} \frac{\mathrm{d}}{\mathrm{d}t}\left\{ \nabla_{ij} E_k\big(\tilde{\boldsymbol{\theta}}(t)\big) \sum_{l=k}^{n-1} \frac{\nabla_i E_l\big(\tilde{\boldsymbol{\theta}}(t)\big)}{R_i^{(l)}\big(\tilde{\boldsymbol{\theta}}(t)\big) + \varepsilon} \right\} \right| \leq (n-k)C_{13}.$$

(SA-6.13)

This gives a bound on the second term in (SA-6.11):

$$\left| \sum_{k=0}^{n} \rho^{n-k}(1-\rho)\nabla_j E_k\big(\tilde{\boldsymbol{\theta}}(t)\big) \sum_{i=1}^{p} \frac{\mathrm{d}}{\mathrm{d}t}\left\{ \nabla_{ij} E_k\big(\tilde{\boldsymbol{\theta}}(t)\big) \sum_{l=k}^{n-1} \frac{\nabla_i E_l\big(\tilde{\boldsymbol{\theta}}(t)\big)}{R_i^{(l)}\big(\tilde{\boldsymbol{\theta}}(t)\big) + \varepsilon} \right\} \right|$$

$$\leq M_1 \sum_{k=0}^{n} \rho^{n-k}(1-\rho)(n-k)C_{13} \leq C_{14},$$

concluding the proof of (SA-6.6).

**SA-6.10 Proof of** (SA-6.7)**.** We will prove this by bounding the four terms in the expression

$$\frac{\mathrm{d}}{\mathrm{d}t}\left\{ \sum_{k=0}^{n} \rho^{n-k}(1-\rho)\nabla_j E_k\big(\tilde{\boldsymbol{\theta}}(t)\big) \sum_{i=1}^{p} \nabla_{ij} E_k\big(\tilde{\boldsymbol{\theta}}(t)\big) \frac{\nabla_i E_n\big(\tilde{\boldsymbol{\theta}}(t)\big)}{R_i^{(n)}\big(\tilde{\boldsymbol{\theta}}(t)\big) + \varepsilon} \right\}$$

$$= \mathrm{Term1} + \mathrm{Term2} + \mathrm{Term3} + \mathrm{Term4},$$

where

Term1

$$:= \sum_{k=0}^{n} \rho^{n-k}(1-\rho)\frac{\mathrm{d}}{\mathrm{d}t}\left\{ \nabla_j E_k\big(\tilde{\boldsymbol{\theta}}(t)\big) \right\} \sum_{i=1}^{p} \nabla_{ij} E_k\big(\tilde{\boldsymbol{\theta}}(t)\big) \frac{\nabla_i E_n\big(\tilde{\boldsymbol{\theta}}(t)\big)}{R_i^{(n)}\big(\tilde{\boldsymbol{\theta}}(t)\big) + \varepsilon},$$

Term2

$$:= \sum_{k=0}^{n} \rho^{n-k}(1-\rho)\nabla_j E_k\big(\tilde{\boldsymbol{\theta}}(t)\big) \sum_{i=1}^{p} \frac{\mathrm{d}}{\mathrm{d}t}\left\{ \nabla_{ij} E_k\big(\tilde{\boldsymbol{\theta}}(t)\big) \right\} \frac{\nabla_i E_n\big(\tilde{\boldsymbol{\theta}}(t)\big)}{R_i^{(n)}\big(\tilde{\boldsymbol{\theta}}(t)\big) + \varepsilon},$$

Term3

$$:= \sum_{k=0}^{n} \rho^{n-k}(1-\rho) \nabla_j E_k\left(\tilde{\boldsymbol{\theta}}(t)\right) \sum_{i=1}^{p} \nabla_{ij} E_k\left(\tilde{\boldsymbol{\theta}}(t)\right) \frac{\frac{\mathrm{d}}{\mathrm{d}t}\left\{\nabla_i E_n\left(\tilde{\boldsymbol{\theta}}(t)\right)\right\}}{R_i^{(n)}\left(\tilde{\boldsymbol{\theta}}(t)\right) + \varepsilon},$$

Term4

$$:= -\sum_{k=0}^{n} \rho^{n-k}(1-\rho) \nabla_j E_k\left(\tilde{\boldsymbol{\theta}}(t)\right) \sum_{i=1}^{p} \nabla_{ij} E_k\left(\tilde{\boldsymbol{\theta}}(t)\right) \frac{\nabla_i E_n\left(\tilde{\boldsymbol{\theta}}(t)\right)\frac{\mathrm{d}}{\mathrm{d}t} R_i^{(n)}\left(\tilde{\boldsymbol{\theta}}(t)\right)}{\left(R_i^{(n)}\left(\tilde{\boldsymbol{\theta}}(t)\right) + \varepsilon\right)^2}.$$

To bound Term1, use $\left|\frac{\mathrm{d}}{\mathrm{d}t}\left\{\nabla_j E_k\left(\tilde{\boldsymbol{\theta}}(t)\right)\right\}\right| \leq D_1 p M_2$, giving

$$|\text{Term1}| \leq \frac{D_1 p^2 M_1 M_2^2}{R+\varepsilon} \sum_{k=0}^{n} \rho^{n-k}(1-\rho) \leq \frac{D_1 p^2 M_1 M_2^2}{R+\varepsilon}.$$

To bound Term2, use $\left|\frac{\mathrm{d}}{\mathrm{d}t}\left\{\nabla_{ij} E_k\left(\tilde{\boldsymbol{\theta}}(t)\right)\right\}\right| \leq D_1 p M_3$, giving

$$|\text{Term2}| \leq \frac{D_1 p^2 M_1^2 M_3}{R+\varepsilon} \sum_{k=0}^{n} \rho^{n-k}(1-\rho) \leq \frac{D_1 p^2 M_1^2 M_3}{R+\varepsilon}.$$

To bound Term3, use $\left|\frac{\mathrm{d}}{\mathrm{d}t}\left\{\nabla_i E_n\left(\tilde{\boldsymbol{\theta}}(t)\right)\right\}\right| \leq D_1 p M_2$, giving

$$|\text{Term3}| \leq \frac{D_1 p^2 M_1 M_2^2}{R+\varepsilon} \sum_{k=0}^{n} \rho^{n-k}(1-\rho) \leq \frac{D_1 p^2 M_1 M_2^2}{R+\varepsilon}.$$

To bound Term4, use (SA-6.12), giving

$$|\text{Term4}| \leq \frac{p M_1^2 M_2 C_{11}}{(R+\varepsilon)^2} \sum_{k=0}^{n} \rho^{n-k}(1-\rho) \leq \frac{p M_1^2 M_2 C_{11}}{(R+\varepsilon)^2}.$$

**SA-6.11 Proof of** (SA-6.8)**.** This is proven in (SA-6.13).

**SA-6.12 Proof of** (SA-6.9)**.** (SA-6.12) gives

$$\left|\frac{\mathrm{d}}{\mathrm{d}t}\left\{\frac{1}{R_j^{(n)}\left(\tilde{\boldsymbol{\theta}}(t)\right)}\right\}\right| = \frac{\left|\frac{\mathrm{d}}{\mathrm{d}t} R_j^{(n)}\left(\tilde{\boldsymbol{\theta}}(t)\right)\right|}{R_j^{(n)}\left(\tilde{\boldsymbol{\theta}}(t)\right)^2} \leq \frac{C_{11}}{R^2}, \tag{SA-6.14}$$

$$\left|\frac{\mathrm{d}}{\mathrm{d}t}\left\{\frac{1}{R_j^{(n)}\left(\tilde{\boldsymbol{\theta}}(t)\right)+\varepsilon}\right\}\right| = \frac{\left|\frac{\mathrm{d}}{\mathrm{d}t} R_j^{(n)}\left(\tilde{\boldsymbol{\theta}}(t)\right)\right|}{\left(R_j^{(n)}\left(\tilde{\boldsymbol{\theta}}(t)\right)+\varepsilon\right)^2} \leq \frac{C_{11}}{(R+\varepsilon)^2}, \tag{SA-6.15}$$

$$\left|\frac{\mathrm{d}}{\mathrm{d}t}\left\{\frac{1}{\left(R_j^{(n)}\left(\tilde{\boldsymbol{\theta}}(t)\right)+\varepsilon\right)^2}\right\}\right| = \frac{2\left|\frac{\mathrm{d}}{\mathrm{d}t} R_j^{(n)}\left(\tilde{\boldsymbol{\theta}}(t)\right)\right|}{\left(R_j^{(n)}\left(\tilde{\boldsymbol{\theta}}(t)\right)+\varepsilon\right)^3} \leq \frac{2C_{11}}{(R+\varepsilon)^3}. \tag{SA-6.16}$$

Combining two bounds above, we have

$$\left|\frac{\mathrm{d}}{\mathrm{d}t}\left\{\left(R_j^{(n)}\left(\tilde{\boldsymbol{\theta}}(t)\right)+\varepsilon\right)^{-2} R_j^{(n)}(\tilde{\boldsymbol{\theta}}(t))^{-1}\right\}\right|$$

$$\leq \frac{\left| \frac{\mathrm{d}}{\mathrm{d}t} \left\{ \left( R_j^{(n)}\left(\tilde{\boldsymbol{\theta}}(t)\right) + \varepsilon \right)^{-2} \right\} \right|}{R_j^{(n)}(\tilde{\boldsymbol{\theta}}(t))} + \frac{\left| \frac{\mathrm{d}}{\mathrm{d}t} \left\{ R_j^{(n)}(\tilde{\boldsymbol{\theta}}(t))^{-1} \right\} \right|}{\left( R_j^{(n)}\left(\tilde{\boldsymbol{\theta}}(t)\right) + \varepsilon \right)^2} \leq C_{16}.$$

We are ready to bound

$$\left| \left( \frac{\nabla_j E_n\left(\tilde{\boldsymbol{\theta}}(t)\right)\left(2P_j^{(n)}\left(\tilde{\boldsymbol{\theta}}(t)\right) + \bar{P}_j^{(n)}\left(\tilde{\boldsymbol{\theta}}(t)\right)\right)}{2\left(R_j^{(n)}\left(\tilde{\boldsymbol{\theta}}(t)\right) + \varepsilon\right)^2 R_j^{(n)}\left(\tilde{\boldsymbol{\theta}}(t)\right)} \right)^{\cdot} \right|$$

$$\leq \left| \frac{\left(\nabla_j E_n\left(\tilde{\boldsymbol{\theta}}(t)\right)\right)^{\cdot}\left(2P_j^{(n)}\left(\tilde{\boldsymbol{\theta}}(t)\right) + \bar{P}_j^{(n)}\left(\tilde{\boldsymbol{\theta}}(t)\right)\right)}{2\left(R_j^{(n)}\left(\tilde{\boldsymbol{\theta}}(t)\right) + \varepsilon\right)^2 R_j^{(n)}\left(\tilde{\boldsymbol{\theta}}(t)\right)} \right| +$$

$$+ \left| \frac{\nabla_j E_n\left(\tilde{\boldsymbol{\theta}}(t)\right)\left(2P_j^{(n)}\left(\tilde{\boldsymbol{\theta}}(t)\right) + \bar{P}_j^{(n)}\left(\tilde{\boldsymbol{\theta}}(t)\right)\right)^{\cdot}}{2\left(R_j^{(n)}\left(\tilde{\boldsymbol{\theta}}(t)\right) + \varepsilon\right)^2 R_j^{(n)}\left(\tilde{\boldsymbol{\theta}}(t)\right)} \right|$$

$$+ \left| \frac{\nabla_j E_n\left(\tilde{\boldsymbol{\theta}}(t)\right)\left(2P_j^{(n)}\left(\tilde{\boldsymbol{\theta}}(t)\right) + \bar{P}_j^{(n)}\left(\tilde{\boldsymbol{\theta}}(t)\right)\right)}{2} \right.$$

$$\left. \times \left( \left(R_j^{(n)}\left(\tilde{\boldsymbol{\theta}}(t)\right) + \varepsilon\right)^{-2} R_j^{(n)}(\tilde{\boldsymbol{\theta}}(t))^{-1} \right)^{\cdot} \right| \leq C_{17}.$$

**SA-6.13 Proof of** (SA-6.10)**.** Since

$$\left| \sum_{i=1}^{p} \nabla_{ij} E_n\left(\tilde{\boldsymbol{\theta}}(t)\right) \frac{\nabla_i E_n\left(\tilde{\boldsymbol{\theta}}(t)\right)}{R_i^{(n)}\left(\tilde{\boldsymbol{\theta}}(t)\right) + \varepsilon} \right| \leq \frac{p M_1 M_2}{R + \varepsilon}$$

and, as we have already seen in the argument for (SA-6.7),

$$\left| \left( \sum_{i=1}^{p} \nabla_{ij} E_n\left(\tilde{\boldsymbol{\theta}}(t)\right) \frac{\nabla_i E_n\left(\tilde{\boldsymbol{\theta}}(t)\right)}{R_i^{(n)}\left(\tilde{\boldsymbol{\theta}}(t)\right) + \varepsilon} \right)^{\cdot} \right| \leq \frac{p^2 D_1 M_1 M_3}{R + \varepsilon} + \frac{p^2 D_1 M_2^2}{R + \varepsilon} + \frac{p M_1 M_2 C_{11}}{(R + \varepsilon)^2},$$

we are ready to bound

$$\left| \left( \frac{\sum_{i=1}^{p} \nabla_{ij} E_n\left(\tilde{\boldsymbol{\theta}}(t)\right) \frac{\nabla_i E_n\left(\tilde{\boldsymbol{\theta}}(t)\right)}{R_i^{(n)}\left(\tilde{\boldsymbol{\theta}}(t)\right) + \varepsilon}}{2\left(R_j^{(n)}(\tilde{\boldsymbol{\theta}}(t)) + \varepsilon\right)} \right)^{\cdot} \right| \leq C_{18}.$$

The proof of Lemma SA-6.5 is concluded. $\qquad\square$

**Lemma SA-6.14.** *Suppose Assumption SA-2.2 holds. Then the second derivative of $t \mapsto \tilde{\theta}_j(t)$ is uniformly over $j$ and $t \in [0, T]$ bounded in absolute value by some positive constant, say $D_2$.*

*Proof.* This follows from the definition of $\tilde{\boldsymbol{\theta}}(t)$ given in (SA-2.2), $h \le T$ and that the first derivatives of all three terms in (SA-2.2) are bounded by Lemma SA-6.5. $\square$

**Lemma SA-6.15.** *Suppose Assumption SA-2.2 holds. Then*

$$\left| \left( \nabla_j E_n\left(\tilde{\boldsymbol{\theta}}(t)\right) \right)^{\cdot\cdot} \right| \le C_{19}, \tag{SA-6.17}$$

$$\left| \left( R_j^{(n)}\left(\tilde{\boldsymbol{\theta}}(t)\right) \right)^{\cdot\cdot} \right| \le C_{20}, \tag{SA-6.18}$$

$$\left| \left( \left( R_j^{(n)}\left(\tilde{\boldsymbol{\theta}}(t)\right) + \varepsilon \right)^{-2} \right)^{\cdot\cdot} \right| \le C_{21}, \tag{SA-6.19}$$

$$\left| \left( R_j^{(n)}\left(\tilde{\boldsymbol{\theta}}(t)\right)^{-1} \right)^{\cdot\cdot} \right| \le C_{22}, \tag{SA-6.20}$$

$$\left| \left( \left( R_j^{(n)}\left(\tilde{\boldsymbol{\theta}}(t)\right) + \varepsilon \right)^{-2} R_j^{(n)}\left(\tilde{\boldsymbol{\theta}}(t)\right)^{-1} \right)^{\cdot\cdot} \right| \le C_{23}, \tag{SA-6.21}$$

$$\left| \left( \sum_{i=1}^{p} \nabla_{ij} E_k\left(\tilde{\boldsymbol{\theta}}(t)\right) \sum_{l=k}^{n-1} \frac{\nabla_i E_l\left(\tilde{\boldsymbol{\theta}}(t)\right)}{R_i^{(l)}\left(\tilde{\boldsymbol{\theta}}(t)\right) + \varepsilon} \right)^{\cdot\cdot} \right| \le (n-k)C_{24}, \tag{SA-6.22}$$

*with constants $C_{19}, C_{20}, C_{21}, C_{22}, C_{23}, C_{24}$ defined as follows:*

$$C_{19} := p^2 M_3 D_1^2 + p M_2 D_2,$$

$$C_{20} := \frac{C_{11}}{R^2} p M_1 M_2 D_1 + \frac{1}{R} p^2 M_2^2 D_1^2 + \frac{1}{R} p^2 M_1 M_3 D_1^2 + \frac{1}{R} p M_1 M_2 D_2,$$

$$C_{21} := \frac{6 C_{11}^2}{(R+\varepsilon)^4} + \frac{2 C_{20}}{(R+\varepsilon)^3},$$

$$C_{22} := \frac{2 C_{11}^2}{R^3} + \frac{C_{20}}{R^2},$$

$$C_{23} := \frac{C_{21}}{R} + \frac{4 C_{11}^2}{R^2(R+\varepsilon)^3} + \frac{C_{22}}{(R+\varepsilon)^2},$$

$$C_{24} := p \left[ \frac{2 C_{11}\left(D_1 M_2^2 p + D_1 M_1 M_3 p\right)}{(R+\varepsilon)^2} + M_1 M_2 \left( \frac{2 C_{11}^2}{(R+\varepsilon)^3} + \frac{C_{20}}{(R+\varepsilon)^2} \right) \right.$$

$$\left. + \frac{2 D_1^2 M_2 M_3 p^2 + M_2\left(D_1^2 M_3 p^2 + D_2 M_2 p\right) + M_1\left(D_1^2 M_4 p^2 + D_2 M_3 p\right)}{R+\varepsilon} \right].$$

*Proof of Lemma SA-6.15.* We divide this argument in several steps.

**SA-6.16 Proof of (SA-6.17).** This bound is straightforward:

$$\left| \left( \nabla_j E_n\left(\tilde{\boldsymbol{\theta}}(t)\right) \right)^{\cdot\cdot} \right| = \left| \sum_{i=1}^{p} \sum_{s=1}^{p} \nabla_{ijs} E_n\left(\tilde{\boldsymbol{\theta}}(t)\right) \dot{\tilde{\theta}}_s(t) \dot{\tilde{\theta}}_i(t) + \sum_{i=1}^{p} \nabla_{ij} E_n\left(\tilde{\boldsymbol{\theta}}(t)\right) \ddot{\tilde{\theta}}_t(t) \right| \le C_{19}.$$

**SA-6.17 Proof of (SA-6.18).** Note that

$$\left( R_j^{(n)}\left(\tilde{\boldsymbol{\theta}}(t)\right) \right)^{\cdot\cdot} = \left( R_j^{(n)}\left(\tilde{\boldsymbol{\theta}}(t)\right)^{-1} \right)^{\cdot} \sum_{k=0}^{n} \rho^{n-k}(1-\rho) \nabla_j E_k\left(\tilde{\boldsymbol{\theta}}(t)\right) \sum_{i=1}^{p} \nabla_{ij} E_k\left(\tilde{\boldsymbol{\theta}}(t)\right) \dot{\tilde{\theta}}_i(t)$$

$$+ R_j^{(n)}\left(\tilde{\boldsymbol{\theta}}(t)\right)^{-1} \sum_{k=0}^{n} \rho^{n-k}(1-\rho) \left( \nabla_j E_k\left(\tilde{\boldsymbol{\theta}}(t)\right) \right)^{\cdot} \sum_{i=1}^{p} \nabla_{ij} E_k\left(\tilde{\boldsymbol{\theta}}(t)\right) \dot{\tilde{\theta}}_i(t)$$

$$+ R_j^{(n)}\left(\tilde{\boldsymbol{\theta}}(t)\right)^{-1} \sum_{k=0}^{n} \rho^{n-k}(1-\rho)\nabla_j E_k\left(\tilde{\boldsymbol{\theta}}(t)\right) \sum_{i=1}^{p}\left(\nabla_{ij} E_k\left(\tilde{\boldsymbol{\theta}}(t)\right)\right)^{\cdot} \dot{\tilde{\theta}}_i(t)$$

$$+ R_j^{(n)}\left(\tilde{\boldsymbol{\theta}}(t)\right)^{-1} \sum_{k=0}^{n} \rho^{n-k}(1-\rho)\nabla_j E_k\left(\tilde{\boldsymbol{\theta}}(t)\right) \sum_{i=1}^{p}\nabla_{ij} E_k\left(\tilde{\boldsymbol{\theta}}(t)\right)\ddot{\tilde{\theta}}_i(t),$$

giving by (SA-6.14)

$$\left|\left(R_j^{(n)}\left(\tilde{\boldsymbol{\theta}}(t)\right)\right)^{\cdot\cdot}\right| \leq \frac{C_{11}}{R^2}pM_1M_2D_1\sum_{k=0}^{n}\rho^{n-k}(1-\rho) + \frac{1}{R}p^2M_2^2D_1^2\sum_{k=0}^{n}\rho^{n-k}(1-\rho)$$

$$+ \frac{1}{R}p^2M_1M_3D_1^2\sum_{k=0}^{n}\rho^{n-k}(1-\rho) + \frac{1}{R}pM_1M_2D_2\sum_{k=0}^{n}\rho^{n-k}(1-\rho)$$

$$\leq C_{20}.$$

**SA-6.18 Proof of** (SA-6.19)**.** Note that

$$\left(\left(R_j^{(n)}\left(\tilde{\boldsymbol{\theta}}(t)\right) + \varepsilon\right)^{-2}\right)^{\cdot\cdot} = \frac{6\left(\left(R_j^{(n)}\left(\tilde{\boldsymbol{\theta}}(t)\right)\right)^{\cdot}\right)^2}{\left(R_j^{(n)}\left(\tilde{\boldsymbol{\theta}}(t)\right) + \varepsilon\right)^4} - \frac{2\left(R_j^{(n)}\left(\tilde{\boldsymbol{\theta}}(t)\right)\right)^{\cdot\cdot}}{\left(R_j^{(n)}\left(\tilde{\boldsymbol{\theta}}(t)\right) + \varepsilon\right)^3},$$

giving by (SA-6.12) and (SA-6.18)

$$\left|\left(\left(R_j^{(n)}\left(\tilde{\boldsymbol{\theta}}(t)\right) + \varepsilon\right)^{-2}\right)^{\cdot\cdot}\right| \leq C_{21}.$$

**SA-6.19 Proof of** (SA-6.20)**.** The bound follows from (SA-6.12), (SA-6.18) and

$$\left(R_j^{(n)}\left(\tilde{\boldsymbol{\theta}}(t)\right)^{-1}\right)^{\cdot\cdot} = \frac{2\left(\left(R_j^{(n)}\left(\tilde{\boldsymbol{\theta}}(t)\right)\right)^{\cdot}\right)^2}{R_j^{(n)}\left(\tilde{\boldsymbol{\theta}}(t)\right)^3} - \frac{\left(R_j^{(n)}\left(\tilde{\boldsymbol{\theta}}(t)\right)\right)^{\cdot\cdot}}{R_j^{(n)}\left(\tilde{\boldsymbol{\theta}}(t)\right)^2}.$$

**SA-6.20 Proof of** (SA-6.21)**.** Putting $a := \left(R_j^{(n)}\left(\tilde{\boldsymbol{\theta}}(t)\right) + \varepsilon\right)^{-2}$, $b := R_j^{(n)}\left(\tilde{\boldsymbol{\theta}}(t)\right)^{-1}$, use

$$|a| \leq \frac{1}{(R+\varepsilon)^2}, \quad |b| \leq \frac{1}{R},$$

$$|\dot{a}| \leq \frac{2C_{11}}{(R+\varepsilon)^3}, \quad \left|\dot{b}\right| \leq \frac{C_{11}}{R^2},$$

$$|\ddot{a}| \leq C_{21}, \quad \left|\ddot{b}\right| \leq C_{22},$$

and

$$(ab)^{\cdot\cdot} = \ddot{a}b + 2\dot{a}\dot{b} + a\ddot{b}.$$

**SA-6.21 Proof of** (SA-6.22)**.** Putting

$$a := \nabla_{ij} E_k\left(\tilde{\boldsymbol{\theta}}(t)\right),$$

$$b := \nabla_i E_l\left(\tilde{\boldsymbol{\theta}}(t)\right),$$

$$c := \left(R_i^{(l)}\left(\tilde{\boldsymbol{\theta}}(t)\right) + \varepsilon\right)^{-1},$$

we have

$$|a| \le M_2, \quad |\dot{a}| \le pM_3D_1, \quad |\ddot{a}| \le p^2M_4D_1^2 + pM_3D_2,$$

$$|b| \le M_1, \quad |\dot{b}| \le pM_2D_1, \quad |\ddot{b}| \le p^2M_3D_1^2 + pM_2D_2,$$

$$|c| \le \frac{1}{R+\varepsilon}, \quad |\dot{c}| \le \frac{C_{11}}{(R+\varepsilon)^2}, \quad |\ddot{c}| \le \frac{2C_{11}^2}{(R+\varepsilon)^3} + \frac{C_{20}}{(R+\varepsilon)^2}.$$

(SA-6.22) follows.

The proof of Lemma SA-6.15 is concluded. $\qquad\square$

**Lemma SA-6.22.** *Suppose Assumption SA-2.2 holds. Then the third derivative of $t \mapsto \tilde{\theta}_j(t)$ is uniformly over $j$ and $t \in [0, T]$ bounded in absolute value by some positive constant, say $D_3$.*

*Proof.* By (SA-6.5), (SA-6.13) and (SA-6.22)

$$\left| \sum_{i=1}^{p} \nabla_{ij} E_k\left(\tilde{\boldsymbol{\theta}}(t)\right) \sum_{l=k}^{n-1} \frac{\nabla_i E_l\left(\tilde{\boldsymbol{\theta}}(t)\right)}{R_i^{(l)}\left(\tilde{\boldsymbol{\theta}}(t)\right) + \varepsilon} \right| \le (n-k)C_9,$$

$$\left| \left( \sum_{i=1}^{p} \nabla_{ij} E_k\left(\tilde{\boldsymbol{\theta}}(t)\right) \sum_{l=k}^{n-1} \frac{\nabla_i E_l\left(\tilde{\boldsymbol{\theta}}(t)\right)}{R_i^{(l)}\left(\tilde{\boldsymbol{\theta}}(t)\right) + \varepsilon} \right)^{\cdot} \right| \le (n-k)C_{13},$$

$$\left| \left( \sum_{i=1}^{p} \nabla_{ij} E_k\left(\tilde{\boldsymbol{\theta}}(t)\right) \sum_{l=k}^{n-1} \frac{\nabla_i E_l\left(\tilde{\boldsymbol{\theta}}(t)\right)}{R_i^{(l)}\left(\tilde{\boldsymbol{\theta}}(t)\right) + \varepsilon} \right)^{\cdot\cdot} \right| \le (n-k)C_{24}.$$

From the definition of $t \mapsto P_j^{(n)}\left(\tilde{\boldsymbol{\theta}}(t)\right)$, it means that its derivatives up to order two are bounded. Similarly, the same is true for $t \mapsto \bar{P}_j^{(n)}\left(\tilde{\boldsymbol{\theta}}(t)\right)$.

It follows from (SA-6.19) and its proof that the derivatives up to order two of

$$t \mapsto \left( R_j^{(n)}\left(\tilde{\boldsymbol{\theta}}(t)\right) + \varepsilon \right)^{-2} R_j^{(n)}\left(\tilde{\boldsymbol{\theta}}(t)\right)^{-1}$$

are also bounded.

These considerations give the boundedness of the second derivative of the term

$$t \mapsto \frac{\nabla_j E_n\left(\tilde{\boldsymbol{\theta}}(t)\right) \left( 2P_j^{(n)}\left(\tilde{\boldsymbol{\theta}}(t)\right) + \bar{P}_j^{(n)}\left(\tilde{\boldsymbol{\theta}}(t)\right) \right)}{2\left( R_j^{(n)}\left(\tilde{\boldsymbol{\theta}}(t)\right) + \varepsilon \right)^2 R_j^{(n)}\left(\tilde{\boldsymbol{\theta}}(t)\right)}$$

in (SA-2.2). The boundedness of the second derivatives of the other two terms is shown analogously. By (SA-2.2) and since $h \le T$, this means

$$\sup_{j} \sup_{t \in [0,T]} \left| \dddot{\tilde{\theta}}_j(t) \right| \le D_3$$

for some positive constant $D_3$. $\qquad\square$

# SA-7  Proof of Theorem SA-2.3

**Lemma SA-7.1.** *Suppose Assumption SA-2.2 holds. Then for all $n \in \{0, 1, \ldots, \lfloor T/h \rfloor\}$, $k \in \{0, 1, \ldots, n-1\}$ we have*

$$\left| \nabla_j E_k\left(\tilde{\boldsymbol{\theta}}(t_k)\right) - \nabla_j E_k\left(\tilde{\boldsymbol{\theta}}(t_n)\right) \right| \le C_7(n-k)h \qquad \text{(SA-7.1)}$$

*Proof.* (SA-7.1) follows from the mean value theorem applied $n - k$ times. $\qquad\square$

**Lemma SA-7.2.** *In the setting of Lemma SA-7.1, for any $l \in \{k, k+1, \ldots, n-1\}$ we have*

$$\left| \nabla_j E_k\left(\tilde{\boldsymbol{\theta}}(t_l)\right) - \nabla_j E_k\left(\tilde{\boldsymbol{\theta}}(t_{l+1})\right) - h \sum_{i=1}^{p} \nabla_{ij} E_k\left(\tilde{\boldsymbol{\theta}}(t_n)\right) \frac{\nabla_i E_l\left(\tilde{\boldsymbol{\theta}}(t_n)\right)}{R_i^{(l)}\left(\tilde{\boldsymbol{\theta}}(t_n)\right) + \varepsilon} \right|$$

$$\leq \left( C_{19}/2 + C_8 + (n - l - 1)C_{13} \right) h^2.$$

*Proof.* By the Taylor expansion of $t \mapsto \nabla_j E_k\left(\tilde{\boldsymbol{\theta}}(t)\right)$ on the segment $[t_l, t_{l+1}]$ at $t_{l+1}$ on the left

$$\left| \nabla_j E_k\left(\tilde{\boldsymbol{\theta}}(t_l)\right) - \nabla_j E_k\left(\tilde{\boldsymbol{\theta}}(t_{l+1})\right) + h \sum_{i=1}^{p} \nabla_{ij} E_k\left(\tilde{\boldsymbol{\theta}}(t_{l+1})\right) \dot{\tilde{\boldsymbol{\theta}}}_i\left(t_{l+1}^-\right) \right| \leq \frac{C_{19}}{2} h^2.$$

Combining this with (SA-6.4) gives

$$\left| \nabla_j E_k\left(\tilde{\boldsymbol{\theta}}(t_l)\right) - \nabla_j E_k\left(\tilde{\boldsymbol{\theta}}(t_{l+1})\right) - h \sum_{i=1}^{p} \nabla_{ij} E_k\left(\tilde{\boldsymbol{\theta}}(t_{l+1})\right) \frac{\nabla_i E_l\left(\tilde{\boldsymbol{\theta}}(t_{l+1})\right)}{R_i^{(l)}\left(\tilde{\boldsymbol{\theta}}(t_{l+1})\right) + \varepsilon} \right| \qquad \text{(SA-7.2)}$$

$$\leq \left( C_{19}/2 + C_8 \right) h^2.$$

Now applying the mean-value theorem $n - l - 1$ times, we have

$$\left| \sum_{i=1}^{p} \nabla_{ij} E_k\left(\tilde{\boldsymbol{\theta}}(t_{l+1})\right) \frac{\nabla_i E_l\left(\tilde{\boldsymbol{\theta}}(t_{l+1})\right)}{R_i^{(l)}\left(\tilde{\boldsymbol{\theta}}(t_{l+1})\right) + \varepsilon} - \sum_{i=1}^{p} \nabla_{ij} E_k\left(\tilde{\boldsymbol{\theta}}(t_{l+2})\right) \frac{\nabla_i E_l\left(\tilde{\boldsymbol{\theta}}(t_{l+2})\right)}{R_i^{(l)}\left(\tilde{\boldsymbol{\theta}}(t_{l+2})\right) + \varepsilon} \right| \leq C_{13} h,$$

$$\ldots$$

$$\left| \sum_{i=1}^{p} \nabla_{ij} E_l\left(\tilde{\boldsymbol{\theta}}(t_{n-1})\right) \frac{\nabla_i E_k\left(\tilde{\boldsymbol{\theta}}(t_{n-1})\right)}{R_i^{(l)}\left(\tilde{\boldsymbol{\theta}}(t_{n-1})\right) + \varepsilon} - \sum_{i=1}^{p} \nabla_{ij} E_k\left(\tilde{\boldsymbol{\theta}}(t_n)\right) \frac{\nabla_i E_l\left(\tilde{\boldsymbol{\theta}}(t_n)\right)}{R_i^{(l)}\left(\tilde{\boldsymbol{\theta}}(t_n)\right) + \varepsilon} \right| \leq C_{13} h,$$

and in particular

$$\left| \sum_{i=1}^{p} \nabla_{ij} E_k\left(\tilde{\boldsymbol{\theta}}(t_{l+1})\right) \frac{\nabla_i E_l\left(\tilde{\boldsymbol{\theta}}(t_{l+1})\right)}{R_i^{(l)}\left(\tilde{\boldsymbol{\theta}}(t_{l+1})\right) + \varepsilon} - \sum_{i=1}^{p} \nabla_{ij} E_k\left(\tilde{\boldsymbol{\theta}}(t_n)\right) \frac{\nabla_i E_l\left(\tilde{\boldsymbol{\theta}}(t_n)\right)}{R_i^{(l)}\left(\tilde{\boldsymbol{\theta}}(t_n)\right) + \varepsilon} \right|$$

$$\leq (n - l - 1) C_{13} h.$$

Combining this with (SA-7.2), we conclude the proof of Lemma SA-7.2. $\qquad\square$

**Lemma SA-7.3.** *In the setting of Lemma SA-7.1,*

$$\left| \nabla_j E_k\left(\tilde{\boldsymbol{\theta}}(t_k)\right) - \nabla_j E_k\left(\tilde{\boldsymbol{\theta}}(t_n)\right) - h \sum_{i=1}^{p} \nabla_{ij} E_k\left(\tilde{\boldsymbol{\theta}}(t_n)\right) \sum_{l=k}^{n-1} \frac{\nabla_i E_l\left(\tilde{\boldsymbol{\theta}}(t_n)\right)}{R_i^{(l)}\left(\tilde{\boldsymbol{\theta}}(t_n)\right) + \varepsilon} \right|$$

$$\leq \left( (n - k)(C_{19}/2 + C_8) + \frac{(n - k)(n - k - 1)}{2} C_{13} \right) h^2.$$

*Proof.* Fix $n \in \mathbb{Z}_{\geq 0}$.

Note that

$$\left| \nabla_j E_k\left(\tilde{\boldsymbol{\theta}}(t_k)\right) - \nabla_j E_k\left(\tilde{\boldsymbol{\theta}}(t_n)\right) - h \sum_{i=1}^{p} \nabla_{ij} E_k\left(\tilde{\boldsymbol{\theta}}(t_n)\right) \sum_{l=k}^{n-1} \frac{\nabla_i E_l\left(\tilde{\boldsymbol{\theta}}(t_n)\right)}{R_i^{(l)}\left(\tilde{\boldsymbol{\theta}}(t_n)\right) + \varepsilon} \right|$$

$$= \left| \sum_{l=k}^{n-1} \left\{ \nabla_j E_k\left(\tilde{\boldsymbol{\theta}}(t_l)\right) - \nabla_j E_k\left(\tilde{\boldsymbol{\theta}}(t_{l+1})\right) - h \sum_{i=1}^{p} \nabla_{ij} E_k\left(\tilde{\boldsymbol{\theta}}(t_n)\right) \frac{\nabla_i E_l\left(\tilde{\boldsymbol{\theta}}(t_n)\right)}{R_i^{(l)}\left(\tilde{\boldsymbol{\theta}}(t_n)\right) + \varepsilon} \right\} \right|$$

$$\leq \sum_{l=k}^{n-1} \left| \nabla_j E_k\left(\tilde{\boldsymbol{\theta}}(t_l)\right) - \nabla_j E_k\left(\tilde{\boldsymbol{\theta}}(t_{l+1})\right) - h \sum_{i=1}^{p} \nabla_{ij} E_k\left(\tilde{\boldsymbol{\theta}}(t_n)\right) \frac{\nabla_i E_l\left(\tilde{\boldsymbol{\theta}}(t_n)\right)}{R_i^{(l)}\left(\tilde{\boldsymbol{\theta}}(t_n)\right) + \varepsilon} \right|$$

$$\overset{(a)}{\leq} \sum_{l=k}^{n-1} \left( C_{19}/2 + C_8 + (n-l-1)C_{13} \right) h^2 = \left( (n-k)(C_{19}/2 + C_8) + \frac{(n-k)(n-k-1)}{2} C_{13} \right) h^2,$$

where (a) is by Lemma SA-7.2. $\qquad\qquad\square$

**Lemma SA-7.4.** *Suppose Assumption SA-2.2 holds. Then for all $n \in \{0, 1, \ldots, \lfloor T/h \rfloor\}$*

$$\left| \sum_{k=0}^{n} \rho^{n-k}(1-\rho)\left( \nabla_j E_k\left(\tilde{\boldsymbol{\theta}}(t_k)\right) \right)^2 - R_j^{(n)}\left(\tilde{\boldsymbol{\theta}}(t_n)\right)^2 \right| \leq C_{25} h \qquad\qquad \text{(SA-7.3)}$$

*and*

$$\left| \sum_{k=0}^{n} \rho^{n-k}(1-\rho)\left( \nabla_j E_k\left(\tilde{\boldsymbol{\theta}}(t_k)\right) \right)^2 - R_j^{(n)}\left(\tilde{\boldsymbol{\theta}}(t_n)\right)^2 - 2h P_j^{(n)}\left(\tilde{\boldsymbol{\theta}}(t_n)\right) \right| \leq C_{26} h^2 \qquad\qquad \text{(SA-7.4)}$$

*with $C_{25}$ and $C_{26}$ defined as follows:*

$$C_{25}(\rho) := 2M_1 C_7 \frac{\rho}{1-\rho},$$

$$C_{26}(\rho) := M_1 |C_{19} + 2C_8 - C_{13}| \frac{\rho}{1-\rho}$$

$$+ \left( M_1 C_{13} + |C_{19} + 2C_8 - C_{13}| C_9 + \frac{(C_{19} + 2C_8 - C_{13})^2}{4} \right) \frac{\rho(1+\rho)}{(1-\rho)^2}$$

$$+ \left( C_{13} C_9 + \frac{C_{13}}{2} |C_{19} + 2C_8 - C_{13}| \right) \frac{\rho(1 + 4\rho + \rho^2)}{(1-\rho)^3} + \frac{C_{13}^2}{4} \cdot \frac{\rho(1 + 11\rho + 11\rho^2 + \rho^3)}{(1-\rho)^4}.$$

*Proof.* Note that

$$\left| \left( \nabla_j E_k\left(\tilde{\boldsymbol{\theta}}(t_k)\right) \right)^2 - \left( \nabla_j E_k\left(\tilde{\boldsymbol{\theta}}(t_n)\right) \right)^2 \right|$$

$$\leq \left| \nabla_j E_k\left(\tilde{\boldsymbol{\theta}}(t_k)\right) - \nabla_j E_k\left(\tilde{\boldsymbol{\theta}}(t_n)\right) \right| \cdot \left| \nabla_j E_k\left(\tilde{\boldsymbol{\theta}}(t_k)\right) + \nabla_j E_k\left(\tilde{\boldsymbol{\theta}}(t_n)\right) \right|$$

$$\overset{(a)}{\leq} C_7 (n-k) h \cdot 2M_1,$$

where (a) is by (SA-7.1). Using the triangle inequality, we can conclude

$$\left| \sum_{k=0}^{n} \rho^{n-k}(1-\rho)\left( \nabla_j E_k\left(\tilde{\boldsymbol{\theta}}(t_k)\right) \right)^2 - R_j^{(n)}\left(\tilde{\boldsymbol{\theta}}(t_n)\right)^2 \right|$$

$$\leq 2M_1 C_7 h (1-\rho) \sum_{k=0}^{n} (n-k)\rho^{n-k} = 2M_1 C_7 h (1-\rho) \sum_{k=0}^{n} k\rho^k = 2M_1 C_7 \frac{\rho}{1-\rho} h.$$

(SA-7.3) is proven.

We continue by showing

$$\left| \left( \nabla_j E_k\left(\tilde{\boldsymbol{\theta}}(t_k)\right) \right)^2 - \left( \nabla_j E_k\left(\tilde{\boldsymbol{\theta}}(t_n)\right) \right)^2 \right.$$

$$\left. -2\nabla_j E_k\left(\tilde{\boldsymbol{\theta}}(t_n)\right) h \sum_{i=1}^{p} \nabla_{ij} E_k\left(\tilde{\boldsymbol{\theta}}(t_n)\right) \sum_{l=k}^{n-1} \frac{\nabla_i E_l\left(\tilde{\boldsymbol{\theta}}(t_n)\right)}{R_i^{(l)}\left(\tilde{\boldsymbol{\theta}}(t_n)\right) + \varepsilon} \right|$$

$$\leq 2M_1\left( (n-k)(C_{19}/2 + C_8) + \frac{(n-k)(n-k-1)}{2} C_{13} \right) h^2 \tag{SA-7.5}$$

$$+ 2(n-k)C_9\left( (n-k)(C_{19}/2 + C_8) + \frac{(n-k)(n-k-1)}{2} C_{13} \right) h^3$$

$$+ \left( (n-k)(C_{19}/2 + C_8) + \frac{(n-k)(n-k-1)}{2} C_{13} \right)^2 h^4.$$

To prove this, use

$$\left| a^2 - b^2 - 2bKh \right| \leq 2|b| \cdot |a - b - Kh| + 2|K| \cdot h \cdot |a - b - Kh| + (a - b - Kh)^2$$

with

$$a := \nabla_j E_k\left(\tilde{\boldsymbol{\theta}}(t_k)\right), \quad b := \nabla_j E_k\left(\tilde{\boldsymbol{\theta}}(t_n)\right), \quad K := \sum_{i=1}^{p} \nabla_{ij} E_k\left(\tilde{\boldsymbol{\theta}}(t_n)\right) \sum_{l=k}^{n-1} \frac{\nabla_i E_l\left(\tilde{\boldsymbol{\theta}}(t_n)\right)}{R_i^{(l)}\left(\tilde{\boldsymbol{\theta}}(t_n)\right) + \varepsilon},$$

and bounding

$$|a - b - Kh| \overset{(a)}{\leq} \left( (n-k)(C_{19}/2 + C_8) + \frac{(n-k)(n-k-1)}{2} C_{13} \right) h^2,$$

$$|b| \leq M_1, \quad |K| \leq (n-k)C_9,$$

where (a) is by Lemma SA-7.3. (SA-7.5) is proven.

We turn to the proof of (SA-7.4). By (SA-7.5) and the triangle inequality

$$\left| \sum_{k=0}^{n} \rho^{n-k}(1-\rho)\left( \nabla_j E_k\left(\tilde{\boldsymbol{\theta}}(t_k)\right) \right)^2 - R_j^{(n)}\left(\tilde{\boldsymbol{\theta}}(t_n)\right)^2 - 2hP_j^{(n)}\left(\tilde{\boldsymbol{\theta}}(t_n)\right) \right|$$

$$\leq (1-\rho) \sum_{k=0}^{n} \rho^{n-k}\left( \text{Poly}_1(n-k)h^2 + \text{Poly}_2(n-k)h^3 + \text{Poly}_3(n-k)h^4 \right)$$

$$= (1-\rho) \sum_{k=0}^{n} \rho^{k}\left( \text{Poly}_1(k)h^2 + \text{Poly}_2(k)h^3 + \text{Poly}_3(k)h^4 \right),$$

where

$$\text{Poly}_1(k) := 2M_1\left( k(C_{19}/2 + C_8) + \frac{k(k-1)}{2} C_{13} \right) = M_1 C_{13} k^2 + M_1(C_{19} + 2C_8 - C_{13})k,$$

$$\text{Poly}_2(k) := 2kC_9\left( k(C_{19}/2 + C_8) + \frac{k(k-1)}{2} C_{13} \right) = C_{13}C_9 k^3 + (C_{19} + 2C_8 - C_{13})C_9 k^2,$$

$$\text{Poly}_3(k) := \left( k(C_{19}/2 + C_8) + \frac{k(k-1)}{2} C_{13} \right)^2$$

$$= \frac{C_{13}^2}{4} k^4 + \frac{C_{13}}{2}(C_{19} + 2C_8 - C_{13})k^3 + \frac{1}{4}(C_{19} + 2C_8 - C_{13})^2 k^2.$$

It is left to combine this with

$$\sum_{k=0}^{n} k\rho^k \leq \sum_{k=0}^{\infty} k\rho^k = \frac{\rho}{(1-\rho)^2},$$

$$\sum_{k=0}^{n} k^2 \rho^k \le \sum_{k=0}^{\infty} k^2 \rho^k = \frac{\rho(1+\rho)}{(1-\rho)^3},$$

$$\sum_{k=0}^{n} k^3 \rho^k \le \sum_{k=0}^{\infty} k^3 \rho^k = \frac{\rho(1+4\rho+\rho^2)}{(1-\rho)^4},$$

$$\sum_{k=0}^{n} k^4 \rho^k \le \sum_{k=0}^{\infty} k^4 \rho^k = \frac{\rho(1+11\rho+11\rho^2+\rho^3)}{(1-\rho)^5}.$$

This gives

$$\left| \sum_{k=0}^{n} \rho^{n-k}(1-\rho)\left(\nabla_j E_k\left(\tilde{\boldsymbol{\theta}}(t_k)\right)\right)^2 - R_j^{(n)}\left(\tilde{\boldsymbol{\theta}}(t_n)\right)^2 - 2hP_j^{(n)}\left(\tilde{\boldsymbol{\theta}}(t_n)\right) \right|$$

$$\le \left( M_1 C_{13}\frac{\rho(1+\rho)}{(1-\rho)^2} + M_1|C_{19} + 2C_8 - C_{13}|\frac{\rho}{1-\rho} \right)h^2$$

$$+ \left( C_{13}C_9\frac{\rho(1+4\rho+\rho^2)}{(1-\rho)^3} + |C_{19} + 2C_8 - C_{13}|C_9\frac{\rho(1+\rho)}{(1-\rho)^2} \right)h^3$$

$$+ \left( \frac{C_{13}^2}{4} \cdot \frac{\rho(1+11\rho+11\rho^2+\rho^3)}{(1-\rho)^4} + \frac{C_{13}}{2}|C_{19} + 2C_8 - C_{13}|\frac{\rho(1+4\rho+\rho^2)}{(1-\rho)^3} \right.$$

$$\left. + \frac{1}{4}(C_{19} + 2C_8 - C_{13})^2\frac{\rho(1+\rho)}{(1-\rho)^2} \right)h^4$$

$$\overset{(a)}{\le} \left[ M_1|C_{19} + 2C_8 - C_{13}|\frac{\rho}{1-\rho} \right.$$

$$+ \left( M_1 C_{13} + |C_{19} + 2C_8 - C_{13}|C_9 + \frac{(C_{19} + 2C_8 - C_{13})^2}{4} \right)\frac{\rho(1+\rho)}{(1-\rho)^2}$$

$$+ \left( C_{13}C_9 + \frac{C_{13}}{2}|C_{19} + 2C_8 - C_{13}| \right)\frac{\rho(1+4\rho+\rho^2)}{(1-\rho)^3}$$

$$\left. + \frac{C_{13}^2}{4} \cdot \frac{\rho(1+11\rho+11\rho^2+\rho^3)}{(1-\rho)^4} \right]h^2,$$

where in (a) we used that $h < 1$. (SA-7.4) is proven. $\qquad\square$

**Lemma SA-7.5.** *Suppose Assumption SA-2.2 holds. Then*

$$\left| \left( \sqrt{\sum_{k=0}^{n} \rho^{n-k}(1-\rho)\left(\nabla_j E_k\left(\tilde{\boldsymbol{\theta}}(t_k)\right)\right)^2} + \varepsilon \right)^{-1} - \left( R_j^{(n)}\left(\tilde{\boldsymbol{\theta}}(t_n)\right) + \varepsilon \right)^{-1} \right.$$

$$\left. + h\frac{P_j^{(n)}\left(\tilde{\boldsymbol{\theta}}(t_n)\right)}{\left( R_j^{(n)}\left(\tilde{\boldsymbol{\theta}}(t_n)\right) + \varepsilon \right)^2 R_j^{(n)}\left(\tilde{\boldsymbol{\theta}}(t_n)\right)} \right| \le \frac{C_{25}(\rho)^2 + R^2 C_{26}(\rho)}{2R^3(R+\varepsilon)^2}h^2.$$

*Proof.* Note that if $a \ge R^2$, $b \ge R^2$, we have

$$\left| \frac{1}{\sqrt{a}+\varepsilon} - \frac{1}{\sqrt{b}+\varepsilon} + \frac{a-b}{2\left(\sqrt{b}+\varepsilon\right)^2\sqrt{b}} \right|$$

$$= \frac{(a-b)^2}{2\sqrt{b}\left(\sqrt{b}+\varepsilon\right)\left(\sqrt{a}+\varepsilon\right)\left(\sqrt{a}+\sqrt{b}\right)} \underbrace{\left\{ \frac{1}{\sqrt{b}+\varepsilon} + \frac{1}{\sqrt{a}+\sqrt{b}} \right\}}_{\le 2/R}$$

$$\leq \frac{(a-b)^2}{2R^3(R+\varepsilon)^2}.$$

By the triangle inequality,

$$\left| \frac{1}{\sqrt{a}+\varepsilon} - \frac{1}{\sqrt{b}+\varepsilon} + \frac{c}{2\left(\sqrt{b}+\varepsilon\right)^2\sqrt{b}} \right| \leq \frac{(a-b)^2}{2R^3(R+\varepsilon)^2} + \frac{|a-b-c|}{2\left(\sqrt{b}+\varepsilon\right)^2\sqrt{b}}$$

$$\leq \frac{(a-b)^2}{2R^3(R+\varepsilon)^2} + \frac{|a-b-c|}{2R(R+\varepsilon)^2}$$

Apply this with

$$a := \sum_{k=0}^{n} \rho^{n-k}(1-\rho)\left(\nabla_j E_k\left(\tilde{\boldsymbol{\theta}}(t_k)\right)\right)^2,$$

$$b := R_j^{(n)}\left(\tilde{\boldsymbol{\theta}}(t_n)\right)^2,$$

$$c := 2hP_j^{(n)}\left(\tilde{\boldsymbol{\theta}}(t_n)\right)$$

and use bounds

$$|a-b| \leq 2M_1 C_7 \frac{\rho}{1-\rho} h, \quad |a-b-c| \leq C_{26}(\rho)h^2$$

by Lemma SA-7.4. $\qquad\square$

**SA-7.6.** We are finally ready to prove Theorem SA-2.3.

*Proof of Theorem SA-2.3.* By (SA-6.9) and (SA-6.10), the first derivative of the function

$$t \mapsto \left( \frac{\nabla_j E_n\left(\tilde{\boldsymbol{\theta}}(t)\right)\left(2P_j^{(n)}\left(\tilde{\boldsymbol{\theta}}(t)\right) + \bar{P}_j^{(n)}\left(\tilde{\boldsymbol{\theta}}(t)\right)\right)}{2\left(R_j^{(n)}\left(\tilde{\boldsymbol{\theta}}(t)\right) + \varepsilon\right)^2 R_j^{(n)}\left(\tilde{\boldsymbol{\theta}}(t)\right)} - \frac{\sum_{i=1}^{p} \nabla_{ij} E_n\left(\tilde{\boldsymbol{\theta}}(t)\right)\frac{\nabla_i E_n\left(\tilde{\boldsymbol{\theta}}(t)\right)}{R_i^{(n)}\left(\tilde{\boldsymbol{\theta}}(t)\right)+\varepsilon}}{2\left(R_j^{(n)}(\tilde{\boldsymbol{\theta}}(t)) + \varepsilon\right)} \right)$$

is bounded in absolute value by a positive constant $C_{27} = C_{17} + C_{18}$. By (SA-2.2), this means

$$\left| \ddot{\tilde{\theta}}_j(t) + \frac{\mathrm{d}}{\mathrm{d}t}\left( \frac{\nabla_j E_n\left(\tilde{\boldsymbol{\theta}}(t)\right)}{R_j^{(n)}\left(\tilde{\boldsymbol{\theta}}(t)\right) + \varepsilon} \right) \right| \leq C_{27}h.$$

Combining this with

$$\left| \tilde{\theta}_j(t_{n+1}) - \tilde{\theta}_j(t_n) - \dot{\tilde{\theta}}_j\left(t_n^+\right)h - \frac{\ddot{\tilde{\theta}}_j\left(t_n^+\right)}{2}h^2 \right| \leq \frac{D_3}{6}$$

by Taylor expansion, we get

$$\left| \tilde{\theta}_j(t_{n+1}) - \tilde{\theta}_j(t_n) - \dot{\tilde{\theta}}_j\left(t_n^+\right)h + \frac{h^2}{2}\cdot\frac{\mathrm{d}}{\mathrm{d}t}\left( \frac{\nabla_j E_n\left(\tilde{\boldsymbol{\theta}}(t)\right)}{R_j^{(n)}\left(\tilde{\boldsymbol{\theta}}(t)\right) + \varepsilon} \right)\Bigg|_{t=t_n^+} \right| \tag{SA-7.6}$$

$$\leq \left( \frac{D_3}{6} + \frac{C_{27}}{2} \right)h^3.$$

Using

$$\left| \dot{\tilde{\theta}}_j(t_n) + \frac{\nabla_j E_n\left(\tilde{\boldsymbol{\theta}}(t_n)\right)}{R_j^{(n)}\left(\tilde{\boldsymbol{\theta}}(t_n)\right) + \varepsilon} \right| \leq C_{28}h$$

with $C_{28}$ defined as

$$C_{28} := \frac{M_1(2C_5 + C_6)}{2(R + \varepsilon)^2 R} + \frac{pM_1 M_2}{2(R + \varepsilon)^2}$$

by (SA-2.2), and calculating the derivative, it is easy to show

$$\left| \left. \frac{\mathrm{d}}{\mathrm{d}t} \left( \frac{\nabla_j E_n\big(\tilde{\boldsymbol{\theta}}(t)\big)}{R_j^{(n)}\big(\tilde{\boldsymbol{\theta}}(t)\big) + \varepsilon} \right) \right|_{t = t_n^+} - \mathrm{FrDer} \right| \leq C_{29} h \tag{SA-7.7}$$

for a positive constant $C_{29}$, where

$$\mathrm{FrDer} := \frac{\mathrm{FrDerNum}}{\left( R_j^{(n)}\big(\tilde{\boldsymbol{\theta}}(t_n)\big) + \varepsilon \right)^2 R_j^{(n)}\big(\tilde{\boldsymbol{\theta}}(t_n)\big)}$$

$$\mathrm{FrDerNum} := \nabla_j E_n\big(\tilde{\boldsymbol{\theta}}(t_n)\big) \bar{P}_j^{(n)}\big(\tilde{\boldsymbol{\theta}}(t_n)\big)$$

$$- \left( R_j^{(n)}\big(\tilde{\boldsymbol{\theta}}(t_n)\big) + \varepsilon \right) R_j^{(n)}\big(\tilde{\boldsymbol{\theta}}(t_n)\big) \sum_{i=1}^{p} \nabla_{ij} E_n\big(\tilde{\boldsymbol{\theta}}(t_n)\big) \frac{\nabla_i E_n\big(\tilde{\boldsymbol{\theta}}(t_n)\big)}{R_i^{(n)}\big(\tilde{\boldsymbol{\theta}}(t_n)\big) + \varepsilon},$$

$$C_{29} := \left\{ \frac{pM_2}{R + \varepsilon} + \frac{M_1^2 M_2 p}{(R + \varepsilon)^2 R} \right\} C_{28}.$$

From (SA-7.6) and (SA-7.7), by the triangle inequality

$$\left| \tilde{\theta}_j(t_{n+1}) - \tilde{\theta}_j(t_n) - \dot{\tilde{\theta}}_j(t_n^+) h + \frac{h^2}{2} \mathrm{FrDer} \right| \leq \left( \frac{D_3}{6} + \frac{C_{27} + C_{29}}{2} \right) h^3,$$

which, using (SA-2.2), is rewritten as

$$\left| \tilde{\theta}_j(t_{n+1}) - \tilde{\theta}_j(t_n) + h \frac{\nabla_j E_n\big(\tilde{\boldsymbol{\theta}}(t_n)\big)}{R_j^{(n)}\big(\tilde{\boldsymbol{\theta}}(t_n)\big) + \varepsilon} - h^2 \frac{\nabla_j E_n\big(\tilde{\boldsymbol{\theta}}(t_n)\big) P_j^{(n)}\big(\tilde{\boldsymbol{\theta}}(t_n)\big)}{\left( R_j^{(n)}\big(\tilde{\boldsymbol{\theta}}(t_n)\big) + \varepsilon \right)^2 R_j^{(n)}\big(\tilde{\boldsymbol{\theta}}(t_n)\big)} \right|$$

$$\leq \left( \frac{D_3}{6} + \frac{C_{27} + C_{29}}{2} \right) h^3.$$

It is left to combine this with Lemma SA-7.5, giving the assertion of the theorem with

$$C_1 = \frac{D_3}{6} + \frac{C_{27} + C_{29}}{2} + M_1 \frac{C_{25}^2 + R^2 C_{26}}{2R^3(R + \varepsilon)^2}. \qquad \square$$

## SA-8  Numerical experiments

**SA-8.1 Models.** We use small modifications of default Keras Resnet-50 and Resnet-101 architectures[1] for training on CIFAR-10 and CIFAR-100 (since image sizes are not the same as Imagenet), after verifying their correctness. The first convolution layer `conv1` has $3 \times 3$ kernel, stride 1 and "same" padding. Then comes batch normalization, and relu. Max pooling is removed, and otherwise `conv2_x` to `conv5_x` are as described in [2], see Table 1 there (downsampling is performed by the first convolution of each bottleneck block, same as in this original paper, not the middle one as in version 1.5[2]; all convolution layers have learned biases). After `conv5` there is global average pooling, 10 or 100-way fully connected layer (for CIFAR-10 and CIFAR-100 respectively), and softmax.

---

[1] https://github.com/keras-team/keras/blob/v2.13.1/keras/applications/resnet.py
[2] https://catalog.ngc.nvidia.com/orgs/nvidia/resources/resnet_50_v1_5_for_pytorch

**SA-8.2 Data augmentation.** We subtract the per-pixel mean and divide by standard deviation, and we use the data augmentation scheme from [3], following [2], section 4.2. We take inspiration and some code snippets from [4] (though we do not use their models). During each pass over the training dataset, each $32 \times 32$ initial image is padded evenly with zeros so that it becomes $36 \times 36$, then random crop is applied so that the picture becomes $32 \times 32$ again, and finally random (probability 0.5) horizontal (left to right) flip is used.

**SA-8.3 Experiment details.** In experiments whose results are reported in Figures 4 and 5 of the main paper, we train for more than 3600 epochs and stop training when the train accuracy is near-perfect (Figure SA-1) and the testing accuracy does not significantly improve (Figure SA-2). Therefore, the maximal test accuracies are the final ones reached, and the maximal perturbed one-norms, after excluding the initial fall at the beginning of training, are at peaks of the "hills" on the norm curves (Figure SA-2).

Additional evidence (for ResNet-101 on CIFAR-100 and with hyperparameters different from the ones in Figures 4 and 5) is provided in Figures SA-3 and SA-4.

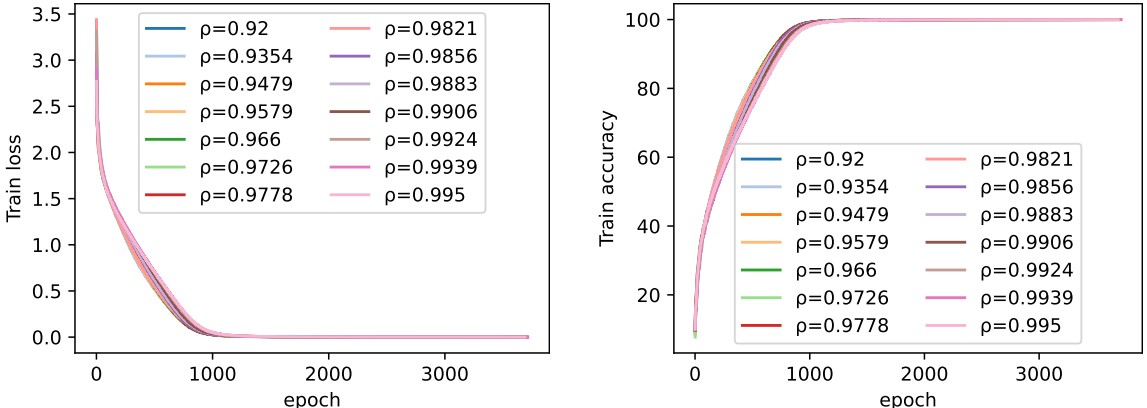

Figure SA-1: Train loss and train accuracy curves for full-batch Adam, ResNet-50 on CIFAR-10, $\beta = 0.99$, $\varepsilon = 10^{-8}$, $h = 7.5 \cdot 10^{-5}$.

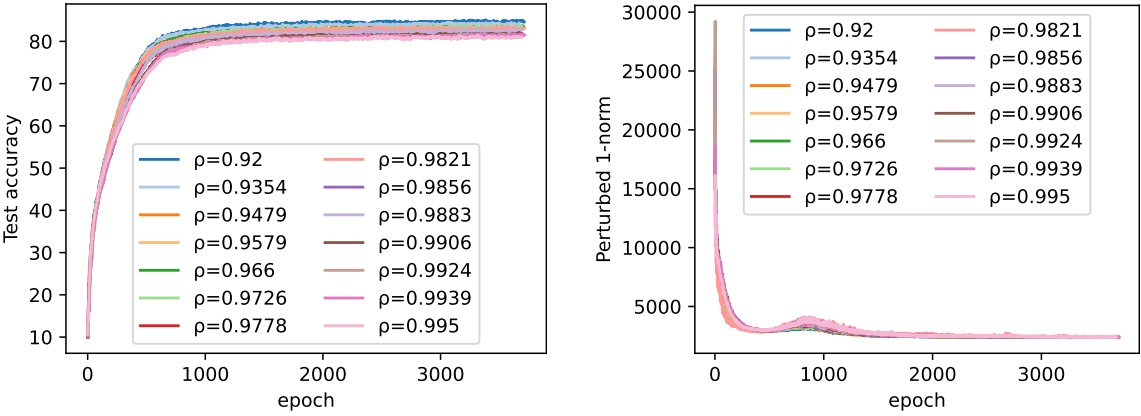

Figure SA-2: Test accuracy and $\|\nabla E\|_{1,\varepsilon}$ after each epoch. The setting is the same as in Figure SA-1.

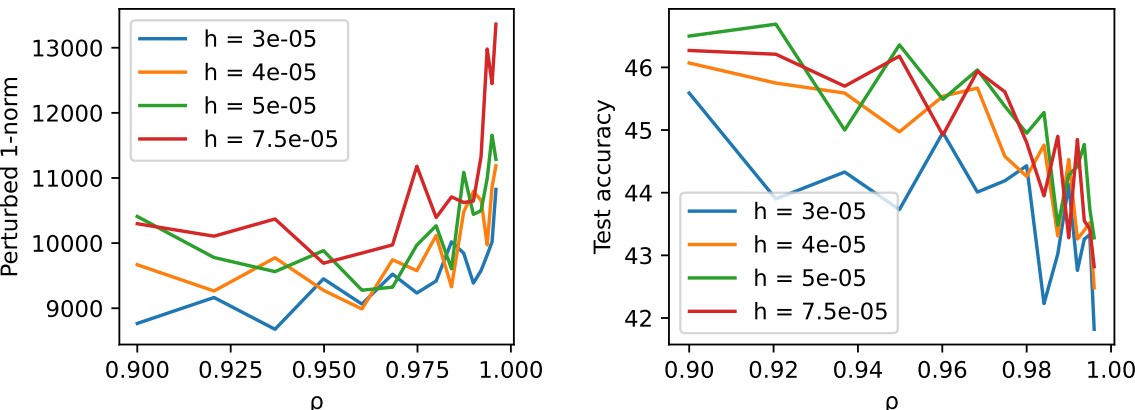

Figure SA-3: Resnet-101 on CIFAR-100 trained with full-batch Adam, $\varepsilon = 10^{-8}$, $\beta = 0.95$. As $\rho$ increases, the perturbed one-norm seems to rise and the test accuracy seems to fall (in the stable regime of training). Both metrics are calculated as in Figures 4 and 5 of the main paper.

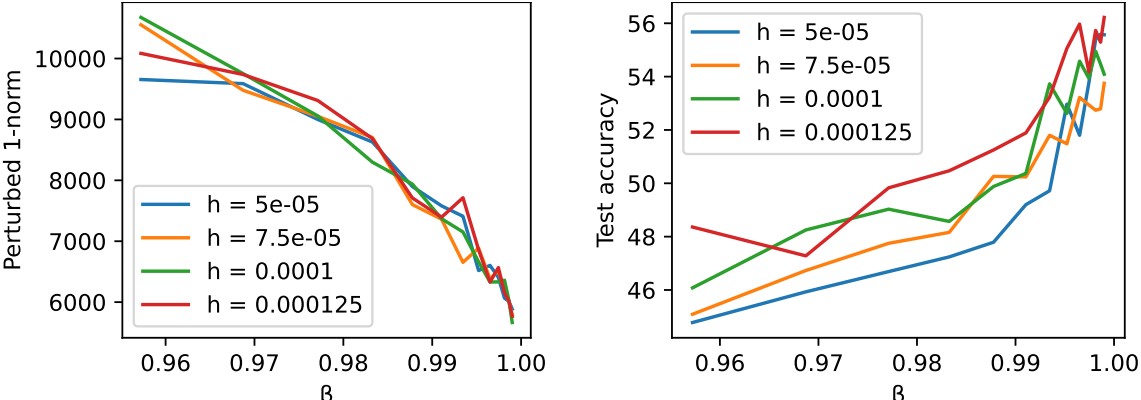

Figure SA-4: Resnet-101 on CIFAR-100 trained with full-batch Adam, $\rho = 0.99$, $\varepsilon = 10^{-8}$. The perturbed one-norm seems to fall as $\beta$ increases, and the test accuracy seems to rise. Both metrics are calculated as in Figures 4 and 5 of the main paper.

## SA-9    Adam with $\varepsilon$ inside the square root: informal derivation

**Result SA-9.1.** *For $n \in \{0, 1, 2, \ldots\}$ we have*

$$\tilde{\theta}_j(t_{n+1}) = \tilde{\theta}_j(t_n) - h \frac{M_j^{(n)}\left(\tilde{\boldsymbol{\theta}}(t_n)\right)}{R_j^{(n)}\left(\tilde{\boldsymbol{\theta}}(t_n)\right)}$$

$$+ h^2 \left( \frac{M_j^{(n)}\left(\tilde{\boldsymbol{\theta}}(t_n)\right) P_j^{(n)}\left(\tilde{\boldsymbol{\theta}}(t_n)\right)}{R_j^{(n)}\left(\tilde{\boldsymbol{\theta}}(t_n)\right)^3} - \frac{L_j^{(n)}\left(\tilde{\boldsymbol{\theta}}(t_n)\right)}{R_j^{(n)}\left(\tilde{\boldsymbol{\theta}}(t_n)\right)} \right) + O\left(h^3\right).$$

(SA-9.1)

*Derivation.* We take

$$\tilde{\theta}_j(t_{n+1}) = \tilde{\theta}_j(t_n) - h \frac{M_j^{(n)}\left(\tilde{\boldsymbol{\theta}}(t_n)\right)}{R_j^{(n)}\left(\tilde{\boldsymbol{\theta}}(t_n)\right)} + O\left(h^2\right)$$

for granted. Using this and the Taylor series, we can write

$$\nabla_j E_k\left(\tilde{\boldsymbol{\theta}}(t_{n-1})\right)$$

$$= \nabla_j E_k\left(\tilde{\boldsymbol{\theta}}(t_n)\right) + \sum_{i=1}^{p} \nabla_{ij} E_k\left(\tilde{\boldsymbol{\theta}}(t_n)\right)\left\{\tilde{\theta}_i(t_{n-1}) - \tilde{\theta}_i(t_n)\right\} + O\left(h^2\right)$$

$$= \nabla_j E_k\left(\tilde{\boldsymbol{\theta}}(t_n)\right) + h \sum_{i=1}^{p} \nabla_{ij} E_k\left(\tilde{\boldsymbol{\theta}}(t_n)\right)\frac{M_j^{(n-1)}\left(\tilde{\boldsymbol{\theta}}(t_{n-1})\right)}{R_j^{(n-1)}\left(\tilde{\boldsymbol{\theta}}(t_{n-1})\right)} + O\left(h^2\right)$$

$$= \nabla_j E_k\left(\tilde{\boldsymbol{\theta}}(t_n)\right) + h \sum_{i=1}^{p} \nabla_{ij} E_k\left(\tilde{\boldsymbol{\theta}}(t_n)\right)\frac{M_j^{(n-1)}\left(\tilde{\boldsymbol{\theta}}(t_n)\right)}{R_j^{(n-1)}\left(\tilde{\boldsymbol{\theta}}(t_n)\right)} + O\left(h^2\right),$$

where in the last equality we just replaced $t_{n-1}$ with $t_n$ in the $h$-term since it only affects higher-order terms. Now doing this again for step $n-1$ instead of step $n$, we will have

$$\nabla_j E_k\left(\tilde{\boldsymbol{\theta}}(t_{n-2})\right)$$

$$= \nabla_j E_k\left(\tilde{\boldsymbol{\theta}}(t_{n-1})\right) + h \sum_{i=1}^{p} \nabla_{ij} E_k\left(\tilde{\boldsymbol{\theta}}(t_{n-1})\right)\frac{M_j^{(n-2)}\left(\tilde{\boldsymbol{\theta}}(t_{n-1})\right)}{R_j^{(n-2)}\left(\tilde{\boldsymbol{\theta}}(t_{n-1})\right)} + O\left(h^2\right)$$

$$= \nabla_j E_k\left(\tilde{\boldsymbol{\theta}}(t_{n-1})\right) + h \sum_{i=1}^{p} \nabla_{ij} E_k\left(\tilde{\boldsymbol{\theta}}(t_{n-1})\right)\frac{M_j^{(n-2)}\left(\tilde{\boldsymbol{\theta}}(t_n)\right)}{R_j^{(n-2)}\left(\tilde{\boldsymbol{\theta}}(t_n)\right)} + O\left(h^2\right),$$

where in the last equality we again replaced $t_{n-1}$ with $t_n$ since it only affects higher-order terms. Proceeding like this and adding the resulting equations, we have for $n \in \{0, 1, \ldots\}$, $k \in \{0, \ldots, n-1\}$ that

$$\nabla_j E_k\left(\tilde{\boldsymbol{\theta}}(t_k)\right)$$

$$= \nabla_j E_k\left(\tilde{\boldsymbol{\theta}}(t_n)\right) + h \sum_{i=1}^{p} \nabla_{ij} E_k\left(\tilde{\boldsymbol{\theta}}(t_n)\right) \sum_{l=k}^{n-1} \frac{M_i^{(l)}\left(\tilde{\boldsymbol{\theta}}(t_n)\right)}{R_i^{(l)}\left(\tilde{\boldsymbol{\theta}}(t_n)\right)} + O\left(h^2\right),$$

where we ignored the fact that $n - k$ is not bounded (we will get away with this because of exponential averaging). Hence, taking the square of this formal power series,

$$\rho^{n-k}(1-\rho)\left(\nabla_j E_k\left(\tilde{\boldsymbol{\theta}}(t_k)\right)\right)^2 = \rho^{n-k}(1-\rho)\left(\nabla_j E_k\left(\tilde{\boldsymbol{\theta}}(t_n)\right)\right)^2$$

$$+ h \cdot 2\rho^{n-k}(1-\rho)\nabla_j E_k\left(\tilde{\boldsymbol{\theta}}(t_n)\right)\sum_{i=1}^{p}\nabla_{ij}E_k\left(\tilde{\boldsymbol{\theta}}(t_n)\right)\sum_{l=k}^{n-1}\frac{M_i^{(l)}\left(\tilde{\boldsymbol{\theta}}(t_n)\right)}{R_i^{(l)}\left(\tilde{\boldsymbol{\theta}}(t_n)\right)} + O\left(h^2\right).$$

Summing up over $k$, we have

$$\frac{1}{1-\rho^{n+1}}\sum_{k=0}^{n}\rho^{n-k}(1-\rho)\left(\nabla_j E_k\left(\tilde{\boldsymbol{\theta}}(t_k)\right)\right)^2 + \varepsilon = R_j^{(n)}\left(\tilde{\boldsymbol{\theta}}(t_n)\right)^2 + 2hP_j^{(n)}\left(\tilde{\boldsymbol{\theta}}(t_n)\right) + O\left(h^2\right),$$

which, using the expression for the inverse square root $\left(\sum_{r=0}^{\infty} a_r h^r\right)^{-1/2}$ of a formal power series $\sum_{r=0}^{\infty} a_r h^r$, gives us

$$\left(\sqrt{\frac{1}{1-\rho^{n+1}}\sum_{k=0}^{n}\rho^{n-k}(1-\rho)\left(\nabla_j E_k\left(\tilde{\boldsymbol{\theta}}(t_k)\right)\right)^2 + \varepsilon}\right)^{-1}$$

$$= \frac{1}{R_j^{(n)}\left(\tilde{\boldsymbol{\theta}}(t_n)\right)} - h\frac{P_j^{(n)}\left(\tilde{\boldsymbol{\theta}}(t_n)\right)}{R_j^{(n)}\left(\tilde{\boldsymbol{\theta}}(t_n)\right)^3} + O\left(h^2\right).$$

Similarly,

$$\frac{1}{1-\beta^{n+1}}\sum_{k=0}^{n}(1-\beta)\beta^{n-k}\nabla_j E_k\left(\tilde{\boldsymbol{\theta}}(t_k)\right) = \frac{1}{1-\beta^{n+1}}\sum_{k=0}^{n}(1-\beta)\beta^{n-k}\nabla_j E_k\left(\tilde{\boldsymbol{\theta}}(t_n)\right)$$

$$+ \frac{h}{1-\beta^{n+1}}\sum_{k=0}^{n}(1-\beta)\beta^{n-k}\sum_{i=1}^{p}\nabla_{ij}E_k\left(\tilde{\boldsymbol{\theta}}(t_n)\right)\sum_{l=k}^{n-1}\frac{M_i^{(l)}\left(\tilde{\boldsymbol{\theta}}(t_n)\right)}{R_i^{(l)}\left(\tilde{\boldsymbol{\theta}}(t_n)\right)} + O\left(h^2\right)$$

$$= M_j^{(n)}\left(\tilde{\boldsymbol{\theta}}(t_n)\right) + hL_j^{(n)}\left(\tilde{\boldsymbol{\theta}}(t_n)\right) + O\left(h^2\right).$$

We conclude

$$\tilde{\theta}_j(t_{n+1}) = \tilde{\theta}_j(t_n) - h\left(M_j^{(n)}\left(\tilde{\boldsymbol{\theta}}(t_n)\right) + hL_j^{(n)}\left(\tilde{\boldsymbol{\theta}}(t_n)\right) + O\left(h^2\right)\right)$$

$$\times \left(\frac{1}{R_j^{(n)}\left(\tilde{\boldsymbol{\theta}}(t_n)\right)} - h\frac{P_j^{(n)}\left(\tilde{\boldsymbol{\theta}}(t_n)\right)}{R_j^{(n)}\left(\tilde{\boldsymbol{\theta}}(t_n)\right)^3} + O\left(h^2\right)\right) + O\left(h^3\right)$$

$$= \tilde{\theta}_j(t_n) - h\frac{M_j^{(n)}\left(\tilde{\boldsymbol{\theta}}(t_n)\right)}{R_j^{(n)}\left(\tilde{\boldsymbol{\theta}}(t_n)\right)}$$

$$+ h^2\left(\frac{M_j^{(n)}\left(\tilde{\boldsymbol{\theta}}(t_n)\right)P_j^{(n)}\left(\tilde{\boldsymbol{\theta}}(t_n)\right)}{R_j^{(n)}\left(\tilde{\boldsymbol{\theta}}(t_n)\right)^3} - \frac{L_j^{(n)}\left(\tilde{\boldsymbol{\theta}}(t_n)\right)}{R_j^{(n)}\left(\tilde{\boldsymbol{\theta}}(t_n)\right)}\right) + O\left(h^3\right). \qquad \square$$

**Result SA-9.2.** *For $t_n \leq t < t_{n+1}$, the modified equation is* (SA-5.2).

*Derivation.* Assume that the modified flow for $t_n \leq t < t_{n+1}$ satisfies $\dot{\tilde{\boldsymbol{\theta}}} = \tilde{\mathbf{f}}\left(\tilde{\boldsymbol{\theta}}(t)\right)$ where

$$\tilde{\mathbf{f}}(\boldsymbol{\theta}) = \mathbf{f}(\boldsymbol{\theta}) + h\mathbf{f}_1(\boldsymbol{\theta}) + O\left(h^2\right).$$

By Taylor expansion, we have

$$\tilde{\boldsymbol{\theta}}(t_{n+1}) = \tilde{\boldsymbol{\theta}}(t_n) + h\dot{\tilde{\boldsymbol{\theta}}}\left(t_n^+\right) + \frac{h^2}{2}\ddot{\tilde{\boldsymbol{\theta}}}\left(t_n^+\right) + O\left(h^3\right)$$

$$= \tilde{\boldsymbol{\theta}}(t_n) + h\left[\mathbf{f}\left(\tilde{\boldsymbol{\theta}}(t_n)\right) + h\mathbf{f}_1\left(\tilde{\boldsymbol{\theta}}(t_n)\right) + O\left(h^2\right)\right]$$

$$+ \frac{h^2}{2}\left[\nabla\mathbf{f}\left(\tilde{\boldsymbol{\theta}}(t_n)\right)\mathbf{f}\left(\tilde{\boldsymbol{\theta}}(t_n)\right) + O(h)\right] + O\left(h^3\right) \qquad\text{(SA-9.2)}$$

$$= \tilde{\boldsymbol{\theta}}(t_n) + h\mathbf{f}\left(\tilde{\boldsymbol{\theta}}(t_n)\right) + h^2\left[\mathbf{f}_1\left(\tilde{\boldsymbol{\theta}}(t_n)\right) + \frac{\nabla\mathbf{f}\left(\tilde{\boldsymbol{\theta}}(t_n)\right)\mathbf{f}\left(\tilde{\boldsymbol{\theta}}(t_n)\right)}{2}\right] + O\left(h^3\right).$$

Using Lemma SA-9.1 and equating the terms before the corresponding powers of $h$ in (SA-9.1) and (SA-9.2), we obtain

$$f_j(\boldsymbol{\theta}) = -\frac{M_j^{(n)}(\boldsymbol{\theta})}{R_j^{(n)}(\boldsymbol{\theta})},$$

$$f_{1,j}(\boldsymbol{\theta}) = -\frac{1}{2}\sum_{i=1}^{p}\nabla_i f_j(\boldsymbol{\theta})f_i(\boldsymbol{\theta}) + \frac{M_j^{(n)}(\boldsymbol{\theta})P_j^{(n)}(\boldsymbol{\theta})}{R_j^{(n)}(\boldsymbol{\theta})^3} - \frac{L_j^{(n)}(\boldsymbol{\theta})}{R_j^{(n)}(\boldsymbol{\theta})}. \qquad\text{(SA-9.3)}$$

It is left to find $\nabla_i f_j(\boldsymbol{\theta})$. Using

$$\nabla_i R_j^{(n)}(\boldsymbol{\theta}) = \frac{\sum_{k=0}^{n}\rho^{n-k}(1-\rho)\nabla_{ij}E_k(\boldsymbol{\theta})\nabla_j E_k(\boldsymbol{\theta})}{(1-\rho^{n+1})R_j^{(n)}(\boldsymbol{\theta})},$$

$$\nabla_i M_j^{(n)}(\boldsymbol{\theta}) = \frac{\sum_{k=0}^n \beta^{n-k}(1-\beta)\nabla_{ij}E_k(\boldsymbol{\theta})}{1-\beta^{n+1}}$$

we have

$$\nabla_i\left(-\frac{M_j^{(n)}(\boldsymbol{\theta})}{R_j^{(n)}(\boldsymbol{\theta})}\right)$$

$$= -\frac{\frac{R_j^{(n)}(\boldsymbol{\theta})^2}{1-\beta^{n+1}}\sum_{k=0}^n \beta^{n-k}(1-\beta)\nabla_{ij}E_k(\boldsymbol{\theta}) - \frac{M_j^{(n)}(\boldsymbol{\theta})}{1-\rho^{n+1}}\sum_{k=0}^n \rho^{n-k}(1-\rho)\nabla_{ij}E_k(\boldsymbol{\theta})\nabla_j E_k(\boldsymbol{\theta})}{R_j^{(n)}(\boldsymbol{\theta})^3}$$

$$= -\frac{\sum_{k=0}^n \beta^{n-k}(1-\beta)\nabla_{ij}E_k(\boldsymbol{\theta})}{(1-\beta^{n+1})R_j^{(n)}(\boldsymbol{\theta})} + \frac{M_j^{(n)}(\boldsymbol{\theta})\sum_{k=0}^n \rho^{n-k}(1-\rho)\nabla_{ij}E_k(\boldsymbol{\theta})\nabla_j E_k(\boldsymbol{\theta})}{(1-\rho^{n+1})R_j^{(n)}(\boldsymbol{\theta})^3}$$

Inserting this into (SA-9.3) concludes the proof. $\qquad\square$