# OpenReview forum: "On the Implicit Bias of Adam"
_ICLR.cc/2024/Conference — Submitted to ICLR 2024_

### Official Review · Reviewer_nNMy · 2023-10-20

**Soundness:** 3 good
**Presentation:** 3 good
**Contribution:** 3 good
**Rating:** 6
**Confidence:** 3

**Summary:**

The authors apply the backward error analysis method to find ODEs that determine continuous trajectories which are close to the discrete trajectories of the popular Adam and RMSProp adaptive gradient-based algorithms.  They succeed in doing this up to a discrepancy which is second-order in the step size, for variants of the two algorithms depending on whether the numerical stability hyperparameter $\varepsilon$ is inside or outside the square root in the step equation, and for both mini-batch and full-batch cases.  The main result for Adam uncovers three different regimes that penalise the positive one-norm of the gradient, or the negative one-norm of the gradient, or the squared two-norm of the gradient, depending on whether the squared gradient momentum hyperparameter $\rho$ is greater than the gradient momentum hyperparameter $\beta$ or not, and whether $\varepsilon$ is small or large compared with the components of the gradient (the latter two cases correspond to early or late phases of the training, respectively).  Some of the results in the literature are derived as special cases of the theorems in this work.  The paper also reports some numerical experiments that seem to confirm the theoretical results as well as suggest that the one-norm of the gradient is inversely correlated with generalisation.

**Strengths:**

The introduction conveys clearly the place of this work in relation to the literature.

The summary of the main result in the full-batch case is helpful, and so are the discussion, the illustration using the bilinear model, and the suggestions of future directions.

The proofs are provided in the supplementary appendix, together with details of the numerical experiments.

**Weaknesses:**

The introduction is a little dry.

The statement of the main result that follows its summary in the full-batch case is difficult to parse, and its connection with the summary that precedes it is not obvious.  A minor point is that equation 10 should not end with a full stop.

**Questions:**

Can you say more about the graphs in Figure 1?  Why are we plotting the integral, and what can we conclude from the shapes of the various curves?

What can you say about the situation when the numerical stability hyperparameter $\varepsilon$ (or rather its square root) is neither small nor large in relation to the components of the gradient?  Can that be the case for a long period of the training?

---

> ### Author Response · Authors · 2023-11-15
>
> Thank you so much for your comments. Each point has been addressed in our response below.
>
> ## Weakness 1
>
> Thank you for pointing this out. We will add more intuition in the introduction.
>
> ## Weakness 2
>
> Thank you for the great suggestions. We have incorporated a brief outline of the derivation and included a comment discussing the relationship between the informal summary and Theorem 3.1 (see Remark 3.2). Thank you for identifying the typo; it has been corrected in the revised version.
>
> ## Question 1
>
> Figure 1 illustrates what kind of function of the gradient is getting "penalized" by taking dimension $p = 1$. The bias term in Eq. (4), ignoring the $h / 2$ coefficient, is the derivative with respect to (one-dimensional) $\theta$ of the function $F(E'(\theta))$, where $F(x) := \int_0^x \bigl[ \frac{1 + \beta}{1 - \beta} - \frac{1 + \rho}{1 - \rho} + \frac{1 + \rho}{1 - \rho} \cdot \frac{\varepsilon}{y^2 + \varepsilon}\bigr] d \sqrt{\varepsilon + y^2}$ and $E'(\theta)$ is the derivative (= gradient) of the loss. Specifically, $ 2/h \cdot \mathrm{bias} = \frac{d}{d \theta} F(E'(\theta)) $ represents what is being penalized, akin to how gradient descent penalizes the loss. For small  $ \varepsilon $, the penalty is effectively plus or minus the 1-norm of the gradient, as shown in the left picture of Figure 1, adjusted by a positive coefficient. As $ \varepsilon $ increases, this term gradually shifts to resemble more closely the 2-norm of the gradient, again modified by a positive coefficient, reflecting Adam's convergence towards gradient descent.
>
> ## Question 2
>
> We believe that in practice $\varepsilon$ is small and, during most of training, the first extreme case describes the situation better. We do, however, also have an understanding of what happens if $\varepsilon$ is not small. In such cases, where $\rho > \beta $, the anti-regularization effect is less pronounced compared to regular Adam. This leads to more stable training, potentially improving generalization, particularly in situations where SGD is superior. Of course, this is contingent on adjusting the effective learning rate and $ h $, as detailed in Example 2.3. Our hypothesis is that increasing $ \varepsilon$ and modifying the learning rate can bring training dynamics closer to momentum-GD, in line with Choi et al. (2019).
>
> Conversely, in language model training, where researchers sometimes fit the data almost exactly and continue training (a phase associated with "grokking"), the situation may hover between the two extremes, leaning towards the second.  It is an interesting regime to study in the future, and perhaps the qualitative behavior of our bias term can contribute some insights.
>
> Reference: Choi, D., et al. (2019). "On Empirical Comparisons of Optimizers for Deep Learning." ArXiv:1910.05446.

---

> > ### Comment · Reviewer_nNMy · 2023-11-16
> >
> > Thank you for these responses.

---

### Official Review · Reviewer_71dm · 2023-10-25

**Soundness:** 2 fair
**Presentation:** 2 fair
**Contribution:** 2 fair
**Rating:** 6
**Confidence:** 3

**Summary:**

This paper studies the implicit bias of Adaptive Gradient Methods (especially RMSProp and ADAM) based on the modified equation of those methods derived by backward error analysis (Sections 2 and 3). As pointed out by the authors (in Related Work), given that research has primarily been conducted on Batch GD, SGD, and SGD with momentum, it is timely to explore the Adaptive Gradient Method. The authors demonstrated that ADAM implicitly penalizes (perturbed) one-norm of gradient depending on $\varepsilon$ scale (Section 2) and empirically verified it (Sections 5 and 6).

**Strengths:**

* The overall contents are easily understandable.
* A timely issue, ADAM's modified loss, is addressed in the paper.
* Though I did not review all the content in the Supplementary material, it appears technically correct.
* The authors validated their findings using a Toy problem (Section 5) and practical neural networks (Section 6).

**Weaknesses:**

* While the overall content is easily comprehensible, specific details are inaccessible unless one reviews the supplementary material. Specifically, many theoretical works provide a sketch of the proof in the main paper to explain the techniques newly considered/developed by the authors and clarify how they differ from existing techniques. This paper lacks such details.

* In addition, the experimental setup in Section 6 is not self-contained and has elements that seem arbitrary, making cherry-picking possible.
  * In Section 6, the authors refer to Ma et al. (2022) to conduct experiments for the stable oscillation regime when $\rho$ and $\beta$ are "sufficiently close". How do you define "sufficiently close"? Most ML practitioners are familiar with situations where $\rho$ (0.99~0.999) is larger than $\beta$ (0.9). As the authors suggest, is this a spike regime or a stable oscillation regime? There should be a discussion for such questions within Section 6, but the authors merely refer to Ma et al. (2022) without providing specific details.
  * In Figures 4 and 5, the authors fixed $h$, $\beta$, and $\rho$ at certain values and ran experiments by varying other parameters. What is the basis for these fixed values? While $h$ is fixed at different values (7.5e-5 and 1e-4) without specific mention, proper empirical evidence should warrant experimentation across various values. Moreover, for $\beta$ and $\rho$, they shouldn't just experiment with a single value. Instead, they should test multiple values and demonstrate consistent backing of their theoretical results.

* The concept of ADAM's modified loss is timely. However, it seems that the resulting modified loss doesn't explain the differences between traditional ADAM and SGD. For example, as the authors mentioned, ADAM often provides worse generalization performance and sharper solutions than SGD. Yet, in NLP tasks using Transformers, ADAM significantly outperforms SGD [1,2]. Such observations lead to two natural questions regarding the authors' study:
  * If one tunes the hyperparameters of ADAM based on the discovered implicit bias, can they reproduce results where SGD performs better?
  * Can the authors explain scenarios where ADAM outperforms SGD using their discovered implicit bias (e.g., can they argue that the proposed perturbed one-norm regularization is more suitable for Self-Attention in Transformers than for Convolutions in ResNet?)

[1] Zhang, Jingzhao, et al. "Why are adaptive methods good for attention models?." Advances in Neural Information Processing Systems 33 (2020): 15383-15393.

[2] Kumar, Ananya, et al. "How to fine-tune vision models with sgd." arXiv preprint arXiv:2211.09359 (2022).

**Questions:**

The questions needed to improve the paper are included in the Weakness section.
If the questions are addressed appropriately, I am willing to raise the score.

---

> ### Author Response · Authors · 2023-11-15
>
> Thank you so much for your comprehensive response. We address each of the points you raised below.
>
> ## Weakness 1
>
> We wholeheartedly agree that including more details of the proof would be beneficial to the reader. To address this, we have introduced two additional levels of detail. Apart from the full proof in the Appendix, we have incorporated a 3-page derivation in Section SA-9, also in the Appendix. This derivation essentially forms the basis of the full proof, with the key difference being the characterization of all big-O bounds in precise terms. Additionally, following Theorem 3.1 in the main paper, we have added a concise sketch of this derivation.
>
> ## Weakness 2
>
> This is a very valid concern that we thank you for raising. We have therefore launched additional hyperparameter sweeps, and pictures pertaining to more learning rates have been added in the latest revision.
>
> While excluding the spike regime might seem questionable, we believe it's justifiable in this case. Without explicit regularization and stochastic batching, the spike magnitudes are significant, often reverting the loss to values seen at the beginning of training. This can hardly go unrecognized by any researcher or engineer. We believe studying the spike regime itself is an important direction for research, but Adam in this regime is too unstable to be reasonably approximated by smooth ODE solutions.
>
> To answer your question about the meaning of "sufficiently close," the threshold for entering the spike regime varies with the task and, crucially, the learning rate. A sufficiently small learning rate prevents spikes altogether. Of course, we do not recommend making $ \beta $ and $ \rho $ equal. For instance, with Resnet-50 on CIFAR-10, using full-batch Adam without explicit regularization and a learning rate of $10^{-4}$, $(\beta, \rho) = (0.9, 0.999)$ enters the spike regime, whereas $ (0.94, 0.999) $ does not. We opt for higher learning rates to accelerate training and make the bias term (linear in $h$) more prounounced.
>
> Following your suggestion, we have added more discussion of this topic in the "Numerical experiments" section.
>
> ## Weakness 3
>
> Thank you very much for pointing out our overstatement regarding Adam's generalization compared to SGD. We acknowledge that Adam does not universally underperform and have revised such assertions. Additionally, we've included a brief mention in the "Future directions" section, noting that Adam often excels in training transformers, as supported by referenced studies.
>
> The paper does not focus on whether the minima targeted by Adam's implicit bias term are advantageous or detrimental for specific tasks. For instance, Andriushchenko et al. (2023) observed that, in training transformers, sharper minima often correlate with lower test errors. As our bias term is novel, further research is necessary to understand its impact on transformer training dynamics.
>
> Reference: Andriushchenko, M., Croce, F., Müller, M., Hein, M., \& Flammarion, N. (2023). A modern look at the relationship between sharpness and generalization. Proceedings of the 40th International Conference on Machine Learning (ICML'23), 202, Article 36, 840-902. JMLR.org.

---

> > ### Comment · Reviewer_71dm · 2023-11-22
> >
> > Thank you for your detailed response! I updated my score to 6.

---

### Official Review · Reviewer_z9rC · 2023-10-31

**Soundness:** 2 fair
**Presentation:** 2 fair
**Contribution:** 2 fair
**Rating:** 6
**Confidence:** 3

**Summary:**

The paper is about the implicit bias of Adam.
It does so by studying ODEs such as the gradient flow and the properties of their discretizations.
The approach is based on backward error analysis (Barret & Dherin 2021), which consists in considering a modified version of the ODE, where the modification is done so that the iterates of gradient descent lie closer to the curve traced by the continuous flow solution.

**Strengths:**

- The topic of the paper, implicit bias of first order algorithms, is an active field of research with many recent results. So far, characterizing the implicit bias of Adam and other preconditioned methods has not been easy.
- The paper, to my understanding, seems to present a novel result corroborated by some empirical evidence in dimension 2.

**Weaknesses:**

- The writing of the paper seems subpar to me, and would benefit from being thoroughly proofread. In some locations it sounded very informal/colloquial, eg "which is ``eaten'' by the gradient".
- The analysis, though interesting, is also handwavy: see questions below.

**Questions:**

In many places, statements are made informally that are to translate into rigorous mathematical terms:
- how do the authors characterize/show/test if "$\epsilon$ is very large compared to all squared gradient components"? What if it's smaller at the beginning, they becomes larger as the algorithm converges to an interpolating solution?
- How does penalizing the norm of the gradient lead to flat minima (Sec 4 discussion)? Since the gradient is 0 at optimum, don't $f$ and $f  + ||\nabla f||^2$ have the same set of minimizers? and doesn't this still hold when the 2 norm is replaced by any other norm?
- similarly, in the experiment, why is the perturbed 1 norm close to 0 at convergence? It seems the authors are performing early stopping, but that precisely means that implicit regularization is not happening, and that the model overfits.
- In the numerical illustrations, is it possible to display more than 3 curves/ values of $h$ and $\beta$? In particular, for limiting values, why isn't the red cross attained if there is implicit bias?
- The figure are not averaged across multiple runs to account for randomness

- (contribution summary, third bullet point)?  Why do the authors consider that Adam does not have an implicit bias, despite having a bias term in the backward analysis ODE. It seems to me that the meaning of "implicit regularization", eg in eq 1.is the same as "bias term" mentioned page 2, but then the statement "Adam has no implicit regularization" is unclear.

---

> ### Author Response · Authors · 2023-11-15
>
> We highly appreciate your feedback. We have carefully considered and responded to each specific point below.
>
> ## Weakness 1
>
> Thank you for your feedback on the style of our presentation. We have revised the sentence that ended with "which is `eaten' by the gradient" to ensure a more formal tone. We will also conduct a thorough proofreading of the entire document to identify instances where the writing is too informal.
>
> ## Weakness 2
>
> Thank you for your comment on the analysis. We included an "Informal summary" section at the beginning of our paper to make it more accessible and prevent readers from being overwhelmed by detailed mathematics. We would like to emphasize that the theorem and its proof are rigorous: the assumptions are precisely defined (as in Theorem 3.1 of the main paper), and the arguments are formal. Note that we do not use a common convention in pure mathematics to allow the value of constant $C$ to change from line to line (the statement and proof of Lemma SA-6.15 may illustrate this). In our latest revision, we have improved the connection between the theorem statement and the informal summary (see Remark 3.2).
>
> ## Question 1
>
> You make a great point. In reality, for the majority of training, the situation tends to align more closely with the first extreme case, which we have now clarified in the revised version of our document. The purpose of outlining these extreme cases is to provide an intuitive understanding of the qualitative behavior. Indeed, at the onset of training, the value of $\varepsilon$ is likely to be smaller than the components. However, towards the end, particularly near an interpolating solution, $\varepsilon$ may become comparable to or even exceed these components. It is in this phase that Adam can essentially become gradient descent, regardless of other hyperparameter settings.
>
> ## Question 2
>
> You are absolutely right in noting that the set of stationary points, where the gradient is zero, remains unchanged regardless of whether a norm is added or subtracted. However, this set can be broad, and varying parameter settings can lead different algorithms to converge to distinct points within this set. Figures 2 and 3 in our document illustrate this point, showing how Adam, with varying parameter settings, converges to different global minimizers. While adding a norm does not alter the location of the minima, it does impact the Hessian matrix. The Hessian plays a crucial role in determining the flatness of the landscape; changes to it affect this flatness. For instance, incorporating the 2-norm tends to penalize sharper regions in the optimization landscape, as highlighted in the works of Barrett \& Dherin (2021) and others.
>
> Reference: Barrett, D., and Dherin, B. (2021). "Implicit Gradient Regularization." In International Conference on Learning Representations.
>
> ## Question 3
>
> Thank you for the great point on the unusual nature of implicit regularization in discrete steps, such as in gradient descent. Contrary to typical cases, the implicit bias we observe here acts more like anti-regularization. You also make a great point about the perturbed 1-norm not appearing to be near zero in the figures. This discrepancy arises because, as noted in our "Notation" section, the perturbed norm is not technically a norm: at or near a stationary point, the square root of $ \varepsilon $ is multiplied by the dimension, which is in the tens of millions. In the final iterations, the gradient components themselves are indeed very small, making the perturbed norm close to its lower bound $ p \sqrt{\varepsilon} $, where $ p $ is the dimension. This might have given the impression of early stopping, but in reality, we are nearly fitting the training data exactly towards the end (with a training accuracy of 99.98%). We have made these points clearer in the latest revision. We also agree that one could interpret the model as overfitting due to the lack of both explicit and, as we argue, this particular form of implicit regularization, aligning with our analysis.
>
> (continued below)

---

> ### Author Response · Authors · 2023-11-15
>
> ## Question 4
>
> Thank you for the great question. In the latest revision, we have included more curves in numerical illustrations. In the toy problem section, attempting to widen the gap further at this learning rate causes instability in Adam, leading to oscillatory behavior around the global minima curve, a pattern similar to RMSProp's periodic solution for minimizing $x^2 / 2$ in Ma et al. (2022). Increasing the learning rate in Figure 3 causes lack of convergence. Even substantial explicit regularization doesn't lead Adam to the red cross without disrupting its trajectory. In scenarios without explicit regularization, akin to Barrett \& Dherin (2021) for GD and Ghosh et al. (2023) for momentum-GD (refer to Figure 1 in both), the red cross is not attained because of the relatively small magnitude of the implicit correction term.
>
> Reference: Ghosh, A., Lyu, H., Zhang, X., and Wang, R. (2023). "Implicit Regularization in Heavy-Ball Momentum Accelerated Stochastic Gradient Descent." In The Eleventh International Conference on Learning Representations.
>
> ## Question 5
>
> We haven't rerun the same experiments to smooth the curves, mainly because the sole source of randomness in our setup is the initialization seed. Our preliminary tests with various initialization seeds yielded visually indistinguishable results, suggesting that additional smoothing might not be effective. Moreover, adopting a Monte-Carlo approach for smoothing would significantly increase training time, likely by an order of magnitude. We have, however, added more experiments to demonstrate the variability in outcomes across different training sessions.
>
> ## Question 6
>
> Thank you very much for pointing out the need for clearer wording here. We've revised the relevant section to convey that the implicit bias we observe typically functions as an "anti-regularization" for standard hyperparameters and during most of training. This bias essentially "penalizes" the negative norm, rather than the norm itself, as shown in Table 1. This clarification has been included in the latest revision.

---

> > ### Comment · Reviewer_z9rC · 2023-12-04
> >
> > I thank the authors for their response, which helped clarify some misunderstandings I had about the paper. I have raised my score to 6.

---

### Official Review · Reviewer_Mieg · 2023-11-01

**Soundness:** 3 good
**Presentation:** 3 good
**Contribution:** 3 good
**Rating:** 6
**Confidence:** 3

**Summary:**

Backward error analysis is used to find ODEs approximating convergence trajectories by optimization algorithms. Previous works have been done on Gradient Descent to show that it has implicit regularization properties since the terms in the ODEs penalize the Euclidean norm of the gradients. This paper studies a similar problem but for adaptive algorithms such as Adam and RMSProp. It shows that Adam and RMSProp also have similar implicit regularization properties.

**Strengths:**

- The paper provides detailed backward error analysis for both Adam and RMSProp. The author is able to show that Adam has bias terms that penalize $1-$ norm, $2-$ norm, or $-1-$ norm depending on the settings of $\beta_1$ and $\beta_2$ in Adam.

- The paper's result in the implicit bias might help explain the difference in the generalization ability of Adaptive Algorithms and GD algorithms.

- The numerical experiments confirm the theoretical results.

- The paper is well-written overall.

**Weaknesses:**

- Some of the graphs are a bit confusing since the $x$ and $y$ axes are not labeled carefully. More explanation and discussion on these graphs would be appreciated.

- Some transformer tasks might be helpful to see if we can see consistent behaviors in the $1-norm$ across different domains. If I'm not mistaken, Adam generalizes better than SGD in transformer related tasks which slightly contradicts the first conclusion in the discussion section.

**Questions:**

- Can the authors explain more about Figure 2 and Figure 3? I'm a bit confused about what these graphs are about and how we can see the change of the $1-norm$ from them.

- Are the norms plotted in section 6 the norms in the final iterate of training?

- Is full batch required to observe the same behaviors of the norm and $\rho, \beta$ as in section 6? Can we do mini-batches instead?

---

> ### Author Response · Authors · 2023-11-15
>
> We greatly appreciate your response. Below, we have addressed each point in detail.
>
> ## Weakness 1 and question 1
>
> Thank you very much for pointing out the absence of axis labels in Figures 2 and 3. We have included them in the latest revision. To clarify, the horizontal axis represents $ \theta_1 $, while the vertical axis corresponds to $ \theta_2 $. The loss function, optimized by Adam, is defined as $ E(\theta_1, \theta_2) = (1.5 - 2\theta_1\theta_2)^2/2 $. The blue line in the figures marks the hyperbola of global minima of this loss, where $ \theta_1\theta_2 = 3/4 $. The gradient is expressed as $ (4\theta_1\theta_2 - 3) \times (\theta_2, \theta_1)^{T} $, which means its 1-norm is simply a factor of $
>  4\theta_1\theta_2 - 3 $ times the 1-norm of the two-dimensional vector $ (\theta_2, \theta_1)^T $.
>
> On the level sets of the form $4 \theta_1\theta_2 - 3 = c$, "lower 1-norm of the gradient" is equivalent to "lower 1-norm of $(\theta_2, \theta_1)^T$". The red cross in our illustrations marks the global minimum that corresponds to the lowest 1-norm of the parameter. Of course, on the entire hyperbola, the 1-norm of the gradient is zero. This is why we describe this point as a limit point of minimizers on the level sets (we clarify this in the latest revision). Thus, the more Adam drifts towards the red cross, the more it penalizes the 1-norm of the gradient.
>
> ## Question 2
>
> In the "Numerical experiments" section, we stopped training at near-perfect train accuracy. This decision was based on the clarity of the images at that stage, rendering further training unnecessary. Consequently, at the final iteration, the norms are close to their lower bound $p \sqrt{\varepsilon}$, limiting their informativeness. Therefore, this section primarily features plots of norms and accuracies mid-training. A more complete picture is offered by the loss and norm curves in Figure 6. We intend to include additional examples of these curves in the Appendix.
>
> ## Question 3
>
> At present, our training utilizes full-batch Adam, due to our better understanding of its qualitative bias. Although our theorem introduces loss correction terms applicable to mini-batch Adam, we do not yet fully understand their implications. Gaining a deeper understanding of the qualitative behavior of these mini-batch correction terms represents a promising direction for future research.
>
> ## Weakness 2
>
> Thank you for highlighting our overly assertive statement regarding the generalization capabilities of adaptive gradient algorithms. A more accurate statement would be: "In the absence of carefully calibrated regularization, it has been observed that adaptive gradient methods may exhibit poorer generalization compared to non-adaptive optimizers." This amended statement aligns with the findings reported by Cohen et al., 2021, who also reference three other studies supporting this observation. We have amended our text accordingly.
>
> We have launched additional experiments and added more evidence in the latest revision. We also plan to explore the training dynamics of transformers as a future research direction. Efforts are underway to incorporate a language translation task, although time constraints may limit our ability to do so.
>
> Reference: Cohen, J. M., Ghorbani, B., Krishnan, S., Agarwal, N., Medapati, S., Badura, M., Suo, D., Cardoze, D., Nado, Z., Dahl, G. E., et al. (2022). "Adaptive Gradient Methods at the Edge of Stability." ArXiv:2207.14484.

---

> > ### Comment · Reviewer_Mieg · 2023-11-18
> >
> > Thank you for the author's responses! I'll keep my score and recommend the paper for acceptance.

---

### Meta-Review · Area_Chair_28u2 · 2023-12-04

**Metareview:**

The paper analyzes the implicit bias of ADAM using a continuous-time analysis. ADAM is the defacto optimizer in machine learning and understanding its bias is an important topic. More specifically, the main result of the paper is to derive an ODE for ADAM using the backward error analysis framework. While prior has shown that the ODE (or SDE for the stochastic case) of GD (gradient descent) penalizes the two-norm of the loss gradients, the authors here demonstrate that the norm penalized by ADAM is different from the one of GD.

Originally, the reviewers were slightly negative about the paper, criticizing the quality of the writing ("The writing of the paper seems subpar") and the arbitrary choice of the experiments ("elements that seem arbitrary, making cherry-picking possible."). The discussion with the authors led the reviewers to slightly increase their scores. However, there is still not a lot of enthusiasm for the paper. I also note that not all comments have been properly addressed. For instance, the curves shown in the paper are not averaged over the choice of seeds (the explanation given by the authors does not seem completely satisfying to me, I would advise to rerun these results).

In addition to the problems raised by the reviewers, one significant additional problem is that the discussion of prior work is severely lacking in several ways:
### Prior work on Implicit bias
For instance, the following papers are not discussed: Qian, Qian, and Xiaoyuan Qian. "The implicit bias of adagrad on separable data." Advances in Neural Information Processing Systems 32 (2019). Wang, Bohan, et al. "Does Momentum Change the Implicit Regularization on Separable Data?." Advances in Neural Information Processing Systems 35 (2022): 26764-26776.

### ODE/SDE for ADAM
Another very important missing reference is https://arxiv.org/pdf/2205.10287.pdf which provides an SDE for the stochastic version of ADAM. They use the framework of Li et al. (which is cited in the paper) which is different from the backward analysis.
The following reference also derives an ODE for ADAM:
Barakat, Anas, and Pascal Bianchi. "Convergence and dynamical behavior of the ADAM algorithm for nonconvex stochastic optimization." SIAM Journal on Optimization 31.1 (2021): 244-274.

### SignGD and choice of norm
There is also a connection between ADAM and signGD (which is obtained by setting some hyper-parameters to zero). SignGD is known to optimize a different norm, which is one of the key results discussed in this paper. Can the insight derived in this paper be seen as sufficiently novel? The following paper also discusses the fact that adaptive algorithm optimizes different norms:
https://arxiv.org/pdf/2002.08056.pdf.
The latter paper is especially relevant. I note that this submission does not contain any detailed discussion regarding the implications for optimizing a different norm.

None of these references are cited in this ICLR submission. I think the paper should include a detailed discussion of these references in a revision and compare their ODE to the differential equations obtained in prior work. One more minor aspect: it would be interesting to plot the distance between the ODE and the discrete-time process for different choices of step sizes.

Overall, the direction pursued by the authors is interesting and I believe this work holds potential for publication, but it is not quite ready from my perspective and needs additional work. I'm looking forward to seeing a revised version of the paper that incorporates a thorough discussion of previous research, validates the presented ODE, better discusses the choice of hyper-parameters used in the experimental results, and enhances the overall writing quality.

**Justification For Why Not Higher Score:**

1) The paper does not discuss some important prior work.
2) The overall quality of the writing of the paper is not good.

**Justification For Why Not Lower Score:**

N/A

---

### Decision · Program_Chairs · 2024-01-16

Reject